# Tuning Sequential Monte Carlo Samplers via Greedy Incremental Divergence Minimization

**Kyurae Kim** [* 1]   **Zuheng Xu** [* 2]   **Jacob R. Gardner** [1]   **Trevor Campbell** [2]

## Abstract

The performance of sequential Monte Carlo (SMC) samplers heavily depends on the tuning of the Markov kernels used in the path proposal. For SMC samplers with unadjusted Markov kernels, standard tuning objectives, such as the Metropolis-Hastings acceptance rate or the expected-squared jump distance, are no longer applicable. While stochastic gradient-based end-to-end optimization has been explored for tuning SMC samplers, they often incur excessive training costs, even for tuning just the kernel step sizes. In this work, we propose a general adaptation framework for tuning the Markov kernels in SMC samplers by minimizing the incremental Kullback-Leibler (KL) divergence between the proposal and target paths. For step size tuning, we provide a gradient- and tuning-free algorithm that is generally applicable for kernels such as Langevin Monte Carlo (LMC). We further demonstrate the utility of our approach by providing a tailored scheme for tuning *kinetic* LMC used in SMC samplers. Our implementations are able to obtain a full *schedule* of tuned parameters at the cost of a few vanilla SMC runs, which is a fraction of gradient-based approaches.

## 1. Introduction

Sequential Monte Carlo (SMC; Dai et al., 2022; Del Moral et al., 2006; Chopin & Papaspiliopoulos, 2020) is a general methodology for simulating Feynman-Kac models (Del Moral, 2004; 2016), which describe the evolution of distributions through sequential changes of mea-

sure. When tuned well, SMC provides state-of-the-art performance in a wide range of modern problem settings, from inference in both state-space models and static models (Dai et al., 2022; Chopin & Papaspiliopoulos, 2020; Doucet & Johansen, 2011; Cappé et al., 2007), to training deep generative models (Arbel et al., 2021; Matthews et al., 2022; Doucet et al., 2023; Maddison et al., 2017), steering large language models (Zhao et al., 2024; Lew et al., 2023), conditional generation from diffusion models (Trippe et al., 2023; Wu et al., 2023), and solving inverse problems with diffusion model priors (Cardoso et al., 2024; Dou & Song, 2024; Achituve et al., 2025).

In practice, however, tuning SMC samplers is often a significant challenge. For example, for static models (Chopin, 2002; Del Moral et al., 2006), one must tune the number of steps, number of particles, target distribution, and Markov kernel at each step, as well as criteria for triggering particle resampling. Since the asymptotic variance of SMC samplers is additive over the steps (Del Moral et al., 2006; Gerber et al., 2019; Chopin, 2004; Webber, 2019; Bernton et al., 2019), all of the above must be tuned adequately *at all times*; an SMC run will not be able to recover from a single mistuned step. While multiple methods for adapting the path of intermediate targets have been proposed (Zhou et al., 2016; Syed et al., 2024), especially in the AIS context (Kiwaki, 2015; Goshtasbpour et al., 2023; Masrani et al., 2021; Jasra et al., 2011), methods and criteria for tuning the path proposal kernels are relatively scarce.

Markov kernels commonly used in SMC can be divided into two categories: those of the Metropolis-Hastings (Metropolis et al., 1953; Hastings, 1970) type, commonly referred to as *adjusted* kernels, and *unadjusted* kernels. For tuning adjusted kernels, one can leverage ideas from the adaptive Markov chain Monte Carlo (MCMC; Robert & Casella, 2004) literature, such as controlling the acceptance probability (Andrieu & Robert, 2001; Atchadé & Rosenthal, 2005) or maximizing the expected-squared jump distance (Pasarica & Gelman, 2010). Both have previously been incorporated into adaptive SMC methods (Fearnhead & Taylor, 2013; Buchholz et al., 2021). On the other hand, tuning unadjusted kernels, which have favorable high-dimensional convergence properties compared to their adjusted coun-

---

[*]Equal contribution [1]Dept. Computer and Information Science, University of Pennsylvania, Philadelphia, U.S. [2]Dept. Statistics, University of British Columbia, Vancouver, Canada. Correspondence to: Kyurae Kim <kyrkim@seas.upenn.edu>, Zuheng Xu <zuheng.xu@stat.ubc.ca>, Trevor Campbell <trevor@stat.ubc.ca>, Jacob R. Gardner <jacobrg@seas.upenn.edu>.

*Proceedings of the $42^{nd}$ International Conference on Machine Learning*, Vancouver, Canada. PMLR 267, 2025. Copyright 2025 by the author(s).

terparts (Lee et al., 2021; Chewi et al., 2021; Roberts & Rosenthal, 1998; Wu et al., 2022; Biswas et al., 2019) and enable fully differentiable samplers (Geffner & Domke, 2021; Zhang et al., 2021; Doucet et al., 2022), is not as straightforward as most techniques from adaptive MCMC cannot be used.

Instead, the typical approach to tuning unadjusted kernels is to minimize a variational objective via stochastic gradient descent (SGD; Robbins & Monro, 1951; Bottou et al., 2018) in an end-to-end fashion (Doucet et al., 2022; Goshtasbpour & Perez-Cruz, 2023; Salimans et al., 2015; Caterini et al., 2018; Gu et al., 2015; Arbel et al., 2021; Matthews et al., 2022; Maddison et al., 2017; Geffner & Domke, 2021; Heng et al., 2020; Chehab et al., 2023; Geffner & Domke, 2023; Naesseth et al., 2018; Le et al., 2018; Zenn & Bamler, 2023). End-to-end optimization approaches are costly: SGD typically requires at least thousands of iterations to converge (*e.g*, Geffner & Domke 2021 use $1.5 \times 10^5$ SGD steps for tuning AIS), where each iteration itself involves an entire run of SMC/AIS. Moreover, SGD is sensitive to several tuning parameters, such as the step size, batch size, and initialization (Sivaprasad et al., 2020). But many of the unadjusted kernels, *e.g.*, random walk MH (Metropolis et al., 1953; Hastings, 1970), Metropolis-adjusted Langevin (Rossky et al., 1978; Besag, 1994), Hamiltonian Monte Carlo (Duane et al., 1987; Neal, 2011), have only a few scalar parameters (*e.g.*, step size) subject to tuning. In this setting, the full generality (and cost) of SGD is not required; it is possible to design a simpler and more efficient method for tuning each transition kernel sequentially in a single SMC/AIS run.

In this work, we propose a novel strategy for tuning path proposal kernels of SMC samplers. Our approach is based on greedily minimizing the incremental Kullback-Leibler (KL; Kullback & Leibler, 1951) divergence between the target and the proposal path measures at each SMC step (§ 3.1). This is reminiscent of annealed flow transport (AFT; Arbel et al., 2021; Matthews et al., 2022), where a normalizing flow (Papamakarios et al., 2021) proposal is trained at each step by minimizing the incremental KL. Instead of training a whole normalizing flow, which requires expensive gradient-based optimization, we tune the parameters of off-the-shelf kernels at each step. This simplifies the optimization process, leading to a gradient- and tuning-free step size adaptation algorithm with quantitative convergence guarantees (§ 3.3).

Using our tuning scheme, we provide complete implementations of tuning-free adaptive SMC samplers for static models: (i) SMC-LMC, which is based on Langevin Monte Carlo (LMC; Rossky et al., 1978; Parisi, 1981; Grenander & Miller, 1994), also commonly known as the unadjusted Langevin algorithm, and (ii) SMC-KLMC, which uses kinetic Langevin Monte Carlo with the "OABAO" discretiza-

tion (Duane et al., 1987; Horowitz, 1991; Monmarché, 2021), also known as unadjusted generalized Hamiltonian Monte Carlo (Neal, 2011). Our method achieves lower variance in normalizing constant estimates compared to the best fixed step sizes obtained through grid search or SGD-based tuning methods. Additionally, the step size schedules found by our method achieve lower or comparable variance than those found by end-to-end optimization approaches without involving any manual tuning (§ 5).

## 2. Background

**Notation.** Let $\mathcal{B}(\mathcal{X})$ be the set of Borel-measurable subsets of some set $\mathcal{X} \subseteq \mathbb{R}^d$. With some abuse of notation, we use the same symbol to denote both a distribution and its density. Also, $\log_+(x) \triangleq \log \max(x, 1)$, $[\cdot]_+ \triangleq \max(\cdot, 0)$, and $[T] \triangleq \{1, \ldots, T\}$.

### 2.1. SMC sampler and Feynman Kac Models

*Sequential Monte Carlo* (SMC; Dai et al., 2022; Del Moral et al., 2006; Chopin & Papaspiliopoulos, 2020) is a general framework for sampling from Feynman-Kac models (Del Moral, 2004; 2016). Consider a space $\mathcal{X}$ with a $\sigma$-finite base measure. Feynman-Kac models describe a change of measure between the *target path distribution*

$$P_{0:T}^\theta(\mathrm{d}x_{0:T}) \triangleq \frac{1}{Z_T^\theta} \left\{ G_0(x_0) \prod_{t=1}^{T} G_t^\theta(x_{t-1}, x_t) \right\} Q_{0:T}^\theta(\mathrm{d}x_{0:T})$$

and the *proposal path distribution*

$$Q_{0:T}^\theta(\mathrm{d}x_{0:T}) \triangleq q(\mathrm{d}x_0) \prod_{t=1}^{T} K_t^\theta(x_{t-1}, \mathrm{d}x_t) , \text{ where}$$

$q$ is the *reference* or initial proposal distribution, $(K_t^\theta)_{t \in [T]}$ are Markov kernels parameterized with $\theta$, and $(G_t^\theta)_{t \in [T]}$ are non-negative $Q$-measurable functions referred to as *potentials*.
The (intermediate) normalizing constant at time $t \in [T]$ is

$$Z_t^\theta = \int_{\mathcal{X}^{t+1}} G_0(x_0) \prod_{s=1}^{t} G_s^\theta(x_{s-1}, x_s) Q_{0:t}^\theta(\mathrm{d}x_{0:t}) .$$

The goal is often to draw samples from $P_{0:T}^\theta$ or to estimate the normalizing constant $Z_T^\theta$.

At time $t = 0$, SMC draws $N$ particles $x_0^{1:N}$ from the initial proposal $q$, each assigned with equal weights $w_0^n = 1$ for $n \in [N]$. At each subsequent time $t \in [T]$, particles $x_{t-1}^{1:N}$ are transported via the transition kernel $K_t^\theta$, reweighted using the potentials $G_t^\theta$, and optionally resampled to discard particles with low weights. See the textbook by Chopin & Papaspiliopoulos (2020) for more details.

At each time $t \in [T]$, the SMC sampler outputs a set of weighted particles $(\bar{w}_t^{1:N}, x_t^{1:N})$, where $\bar{w}_t^n \triangleq w_t^n / \sum_{m \in [N]} w_t^m$, along with an estimate of the normalizing constant $\widehat{Z}_{t,N}$. Under suitable conditions, SMC sam-

plers return consistent estimates of the expectation $P_t(\varphi)$ of a measurable function $\varphi : \mathcal{X} \to \mathbb{R}$ over the marginal $P_t$ and the normalizing constant $Z_t$ (Del Moral, 2004; 2016):

$$\sum_{n \in [N]} \bar{w}_t^n \varphi(x_t^n) \xrightarrow{N \to \infty} P_t(\varphi) \quad \text{and} \quad \widehat{Z}_{t,N} \xrightarrow{N \to \infty} Z_t.$$

Different choices of $G_{0:T}$, $q$, and $K_{0:T}^\theta$ can describe the same target path distribution $P_{0:T}^\theta$ but result in vastly different SMC algorithm performance. Proper tuning is thus essential for achieving high efficiency and accuracy.

## 2.2. Sequential Monte Carlo for Static Models

In this work, we focus on SMC samplers for *static models* where we target a "static" distribution $\pi$, whose density $\pi : \mathcal{X} \to \mathbb{R}_{>0}$ is known up to a normalizing constant $Z$ through the unnormalized density function $\gamma : \mathcal{X} \to \mathbb{R}_{>0}$:

$$\pi(x) \triangleq \frac{\gamma(x)}{Z}, \quad \text{where} \quad Z = \int_{\mathcal{X}} \gamma(x)\,dx.$$

This can be embedded into a sequential inference targeting a "path" of distributions $(\pi_0, \dots, \pi_T)$, where the endpoints are constrained as $\pi_0 = q$ and $\pi_T = \pi$. It is common to choose the *geometric annealing path* by setting the density of $\pi_t$, for $t \in \{0, \dots, T\}$, as

$$\pi_t(x) \propto \gamma_t(x) \triangleq q(x)^{1-\lambda_t} \gamma(x)^{\lambda_t}, \tag{1}$$

where the "temperature schedule" $(\lambda_t)_{t \in \{0,\dots,T\}}$ is monotonically increasing as $0 = \lambda_0 < \dots < \lambda_T = 1$.

To implement an SMC sampler that simulates the path $(\pi_t)_{t \in [T]}$, we introduce a sequence of *backward* Markov kernels $(L_{t-1}^\theta)_{t \in [T]}$ (and refer to the $(K_t^\theta)_{t \in [T]}$ as *forward* kernels). We can then form a Feynman-Kac model by setting the potential for $t \geq 1$ as

$$G_t^\theta(x_{t-1}, x_t) = \frac{Z_{t-1}}{Z_t} \frac{d\left(\pi_t \otimes L_{t-1}^\theta\right)}{d\left(\pi_{t-1} \otimes K_t^\theta\right)}(x_{t-1}, x_t). \tag{2}$$

As long as the condition

$$\pi_t \otimes L_{t-1}^\theta \ll \pi_{t-1} \otimes K_t^\theta \tag{3}$$

holds for all $t \geq 0$ and the Radon-Nikodym derivative can be evaluated pointwise, Eq. (2) is equivalent to

$$G_t^\theta(x_{t-1}, x_t) = \frac{\gamma_t(x_t)\, L_{t-1}^\theta(x_t, x_{t-1})}{\gamma_{t-1}(x_{t-1})\, K_t^\theta(x_{t-1}, x_t)}. \tag{4}$$

Other than the constraint Eq. (3), the choice of forward and backward kernels is a matter of design. Typically, the forward kernel $K_t^\theta$ is selected as a $\pi_t$-invariant (*i.e.*, adjusted) MCMC kernel (Del Moral et al., 2006), such that the particles following $P_{t-1}$ are transported to approximately follow $\pi_t$. This Feynman-Kac model targets the path measure

$$P_{0:T}^\theta(dx_{0:T}) = \pi(dx_T) \prod_{t=1}^{T} L_{t-1}^\theta(x_t, dx_{t-1}).$$

Then, the marginal of $x_T$ is $\pi$, and for $t \in [T]$, the intermediate normalizing constant $Z_t^\theta$ is precisely $Z_t = \int_{\mathcal{X}} \gamma_t(x)\,dx$.

# 3. Adaptation Methodology

## 3.1. Adaptation Objective

The variance of sequential Monte Carlo is minimized when the target path measure $P$ and the proposal path measure $Q$ are close together (Del Moral et al., 2006; Gerber et al., 2019; Chopin, 2004; Webber, 2019; Bernton et al., 2019). A common practice has been to make them close by solving

$$\underset{\theta}{\text{minimize}} \ \ D_{\mathrm{KL}}(Q_{0:T}^\theta, P_{0:T}^\theta).$$

In this work, we are interested in a scheme enabling efficient online adaptation within SMC samplers. One could appeal to the chain rule of the KL divergence:

$$D_{\mathrm{KL}}(Q_{0:T}^\theta, P_{0:T}^\theta)$$
$$= D_{\mathrm{KL}}(Q_0, P_0) + \sum_{t \in [T]} \mathbb{E}_{Q_{t-1}}\{D_{\mathrm{KL}}(Q_{t|t-1}, P_{t|t-1})\},$$

and attempt to minimize the incremental KL terms. Unfortunately, at each step of SMC, we have access to a particle approximation $P_{t-1}$ but not the marginal path proposal $Q_{t-1}$ due to resampling. Instead, we can consider the *forward/inclusive* KL divergence

$$D_{\mathrm{KL}}(P_{0:T}^\theta, Q_{0:T}^\theta)$$
$$= D_{\mathrm{KL}}(P_0, Q_0) + \sum_{t \in [T]} \mathbb{E}_{P_{t-1}}\{D_{\mathrm{KL}}(P_{t|t-1}, Q_{t|t-1})\}.$$

Estimating the incremental forward KL divergence $D_{\mathrm{KL}}(P_{t|t-1}, Q_{t|t-1})$, however, is difficult due to the expectation taken over $P_{t|t-1}$, often resulting in high variance. Therefore, we would like to have a proper divergence measure between the joint paths that (i) decomposes into $T$ incremental terms like the chain rule of the KL divergence, (ii) is easy to estimate, just like the exclusive KL divergence.

Notice that naively summing the incremental exclusive KL divergences as

$$D_{\mathsf{path}}(P_{0:T}, Q_{0:T})$$
$$\triangleq D_{\mathrm{KL}}(Q_0, P_0) + \sum_{t \in [T]} \mathbb{E}_{P_{0:t-1}}\left\{D_{\mathrm{KL}}\left(Q_{t|0:t-1}, P_{t|0:t-1}\right)\right\},$$

satisfies both requirements and turns out to be a valid divergence between path measures:

**Proposition 1.** *Consider joint distributions $Q_{0:T}, P_{0:T}$. Then $D_{\mathsf{path}}$ satisfies the following:*

*(i) $D_{\mathsf{path}}(P_{0:T}, Q_{0:T}) \geq 0$ for any $Q_{0:T}, P_{0:T}$.*

*(ii) $D_{\mathsf{path}}(P_{0:T}, Q_{0:T}) = 0$ if and only if $P_{0:T} = Q_{0:T}$.*

*Proof.* See the *full proof* in page 25.

**Ideal Adaptation Scheme.** Therefore, we propose to adapt SMC samplers by minimizing $D_{path}$.

$$\underset{\theta}{\text{minimize}} \ \ D_{\mathsf{path}}\left(P_{0:T}^\theta, Q_{0:T}^\theta\right). \tag{5}$$

The key convenience of this objective is that for most cases that we will consider, the tunable parameters $\theta$ decompose into a sequence of subsets $\theta = (\theta_1, \dots, \theta_T)$, where at any

$t \in [T]$, $K_t$ and $G_t$ depend on only $\theta_{1:t}$ while $\theta_t$ dominates their variance contribution. This suggests a greedy scheme where we solve for a subset of parameters at a time. By fixing $\theta_{1:t-1}$ from previous iterations, we solve for

$$\theta_t = \arg\min_{\theta_t} \mathbb{E}_{P_{t-1}^{\theta_{1:t-1}}} \left\{ D_{KL}(Q_{t|t-1}^{\theta_{1:t}}, P_{t|t-1}^{\theta_{1:t}}) \right\} . \quad (6)$$

This greedy strategy does not guarantee a solution to the joint optimization in Eq. (5). However, as long as greedily setting $\theta_t$ does not negatively influence future and past steps, which is reasonable for the kernels we consider, this strategy should yield a good approximate solution.

**Relation with Annealed Flow Transport.** For the static model case, Arbel et al. (2021) noted that the objective in Eq. (6) approximates

$$\mathbb{E}_{x_{t-1} \sim \pi_{t-1}} \left\{ D_{KL}(\pi_{t-1} \otimes K_t^{\theta_{1:t}}, \pi_t \otimes L_{t-1}^{\theta_{1:t}} | x_{t-1}) \right\} . \quad (7)$$

Furthermore, Matthews et al. (2022, §3) showed that, when $K_t$ is taken to be a normalizing flow $\mathcal{F}_t$ (Papamakarios et al., 2021) and $L_{t-1} = \mathcal{F}_t^{-1}$, there exists a joint objective associated with Eq. (7),

$$D_{KL}\left( \prod_{t=1}^{T} \mathcal{F}_t^{\#} \pi_{t-1}, \ \prod_{t=1}^{T} \pi_t \right) , \quad (8)$$

where $\mathcal{F}_t^{\#} \pi_{t-1}$ is the pushforward measure of $\pi_{t-1}$ pushed through $\mathcal{F}_t$. Our derivation of Eq. (6) shows that it is not just minimizing an approximation to some joint objective as Eq. (8), but a proper divergence between the joint target $P$ and joint path $Q$. This general principle applies to all Feynman-Kac models, not just those for static models.

**Incremental KL Objective for Feynman-Kac Models.** For Feynman-Kac models, Eq. (6) takes the form

$$\mathbb{E}_{P_{1:t-1}^{\theta_{1:t-1}}} \left\{ D_{KL}(Q_{t|1:t-1}^{\theta_{1:t}}, P_{t|1:t-1}^{\theta_{1:t}}) \right\}$$

$$= \int \int \frac{dQ_{t|t-1}^{\theta_{1:t}}}{dP_{t|t-1}^{\theta_{1:t}}} dQ_{t|t-1}^{\theta_{1:t}} dP_{t-1}^{\theta_{1:t-1}}$$

$$= \int \int -\log G_t^{\theta_{1:t}}(x_{t-1}, x_t) K_t^{\theta_{1:t}}(x_{t-1}, dx_t) \, dP_{t-1}^{\theta_{1:t-1}}$$

$$\quad - \log\left( Z_t^{\theta_{1:t}} / Z_{t-1}^{\theta_{1:t-1}} \right) .$$

The normalizing constant ratio forms a telescoping sum such that the path divergence becomes

$$D_{path}\left( P_{0:T}^{\theta}, Q_{0:T}^{\theta} \right) = D_{KL}(Q_0, P_0) - \log\frac{Z_T}{Z_0}$$

$$+ \sum_{t \in [T]} \mathbb{E}_{(x_{t-1}, x_t) \sim P_{t-1}^{\theta_{1:t-1}} \otimes K_t^{\theta_{1:t}}} \left\{ -\log G_t^{\theta_{1:t}}(x_{t-1}, x_t) \right\} .$$

In practice, Feynman-Kac models are designed such that both $Z_T$ and $Z_0$ are fixed regardless of $\theta$: $Z_0$ is the normalizing constant of $q$, which is usually 1, and $Z_T$ is the normalizing constant of the target $P_{0:T}^{\theta}$. Therefore, for such Feynman-Kac models, solving Eq. (6) is equivalent to

$$\theta_t = \arg\min_{\theta_t} \mathbb{E}_{(x_{t-1}, x_t) \sim P_{t-1}^{\theta_{1:t-1}} \otimes K_t^{\theta_{1:t}}} \left\{ -\log G_t^{\theta_{1:t}}(x_{t-1}, x_t) \right\}. \quad (9)$$

---

**Algorithm 1:** Adaptive Sequential Monte Carlo

$x_0^n \sim q, \quad w_0^n = 1, \quad \widehat{Z}_{0,N} \leftarrow 1$
**for** $t = 1, \ldots, T$ **do**
$\quad \epsilon_t^b \sim \psi$
$\quad \widetilde{a}_{t-1}^{1:B} = \texttt{resample}_B\left( w_{t-1}^{1:N} \right)$
$\quad \widetilde{x}_{t-1}^b = x_{t-1}^{\widetilde{a}_{t-1}^b}, \quad \widetilde{w}_{t-1}^b = 1$
$\quad \theta_t = \arg\min_{\theta_t} \widehat{\mathcal{L}}_t\left( \theta_t; \widetilde{x}_{t-1}^{1:B}, \widetilde{w}_{t-1}^{1:B}, \epsilon_t^{1:B} \right) + \tau\texttt{reg}\left(\theta_t\right)$
$\quad x_t^n \sim K_t^{\theta_{1:t}}\left( x_{t-1}^n, \cdot \right)$
$\quad w_t^n \leftarrow w_{t-1}^n G_t^{\theta_{1:t}}\left( x_{t-1}^n, x_t^n \right)$
$\quad$**if** *resampling is triggered or* $t = T$ **then**
$\quad\quad \widehat{Z}_{t,N} = \widehat{Z}_{t-1,N} \frac{1}{N} \sum_{n \in [N]} w_t^n$
$\quad\quad a_t^{1:N} = \texttt{resample}_N\left( w_t^{1:N} \right)$
$\quad\quad x_t^n \leftarrow x_t^{a_t^n}, \quad w_t^n \leftarrow 1$
$\quad$**end**
**end**

---

### 3.2. General Adaptation Scheme

**Estimating the Incremental KL Objective.** Now that we have discussed our ideal objective for adaptation in Eq. (9), we turn to estimating this objective in practice. At each iteration $t \in [T]$, we have access to a collection of weighted particles

$$\sum_{n \in [N]} \frac{1}{Z_{t-1}^{\theta_{1:t-1}}} w_{t-1}^n \delta_{x_{t-1}^n} \sim P_{t-1}^{\theta_{1:t-1}}$$

up to a constant with respect to $\theta_t$, $Z_{t-1}^{\theta_{1:t-1}}$, where $\delta_{x_{t-1}^n}$ is a Dirac measure centered on $x_{t-1}^n$. Consider the case where sampling from $K_t^{\theta_{1:t}}$ can be represented by a map $M_t^{\theta_{1:t}} : \mathcal{X} \times \mathcal{E} \to \mathcal{X}$, where the randomness over the space $\mathcal{E}$ following $\psi : \mathcal{B}(\mathcal{E}) \to \mathbb{R}_{\geq 0}$ is captured by $\epsilon_t^n \sim \psi$:

$$x_t^n = M_t^{\theta_{1:t}}\left( x_{t-1}^n; \epsilon_t^n \right) .$$

Then, up to a constant, we obtain a conditionally unbiased estimate of the expectation in Eq. (9) as a function of $\theta$:

$$\widehat{\mathcal{L}}_t(\theta_t; x_{t-1}^{1:N}, w_{t-1}^{1:N}, \epsilon_t^{1:N})$$

$$\triangleq - \sum_{n \in [N]} \bar{w}_{t-1}^n \log G_t^{\theta_{1:t}}\left( x_{t-1}^n, M_t^{\theta_{1:t}}\left( x_{t-1}^n; \epsilon_t^n \right) \right). \quad (10)$$

**Efficiently Optimizing the Objective.** Directly optimizing $\widehat{\mathcal{L}}_t$, however, is challenging: (i) Evaluating $\widehat{\mathcal{L}}_t$ takes $O(N)$ evaluations of the potential, which can be expensive. (ii) The expectation over the kernel $K_t$ or, equivalently, over $\epsilon_t^n \sim \psi$, is intractable. We address these issues as follows:

1. **Subsampling of Particles.** To reduce the $O(N)$ cost of evaluating $\widehat{\mathcal{L}}_t$, we apply resampling over the particles according to the weights $w_{t-1}^{1:N}$ such that we end up with a smaller subset of particles of size $B \ll N$, which remains a valid approximation of $P_{t-1}^{\theta_{1:t-1}}$. Then, evaluating $\widehat{\mathcal{L}}_t$ takes $O(B)$ evaluations of the potential.

2. **Sample Average Approximation.** Properly minimizing the expectation over $K_t$ requires stochastic optimization algorithms, which introduce numerous challenges related to convergence determination, step size tuning,

---

**Algorithm 2:** AdaptStepsize $(\mathcal{L}, t, h_{\text{guess}}, \delta, c, r, \epsilon)$

---

**Input:** Adaptation objective $\mathcal{L} : (0, \infty) \to \mathbb{R} \cup \{+\infty\}$,
      SMC iteration $t \in [T]$,
      initial guess $h_{\text{guess}} > 0$,
      backing-off step size $\delta < 0$,
      exponential search coefficient $c > 0$,
      exponential search exponent $r > 1$,
      absolute tolerance $\epsilon > 0$.
**Output:** Adapted step size $h$.

1   $\mathcal{L}^{\log}(\ell) \triangleq \mathcal{L}(\exp(\ell))$
2   $\ell \leftarrow \log h_{\text{guess}}$
3   **if** $t = 1$ **then**
4     $\ell \leftarrow$ FindFeasible $(\mathcal{L}^{\log}, \ell, \delta)$
5   **end**
6   $\ell \leftarrow$ Minimize $(\mathcal{L}^{\log}, \ell, c, r, \epsilon)$
7   Return $\exp(\ell)$

---

handling instabilities, and such. Instead, we draw a single batch of randomness $(\epsilon_t^b)_{b \in [B]}$, and fix it throughout the optimization procedure. This sample average approximation (SAA; Kim et al., 2015) introduces bias in the optimized solution but enables the use of more reliable deterministic techniques.

3. **Regularization.** Subsampling the particles results in a higher variance for estimating the objective. We counteract this by adding a weighted regularization term $\tau \operatorname{reg}(\theta_t)$ to the objective. For example, for the case of step sizes at $t > 1$ such that $\theta_t$ contains $h_t$, we will set $\tau \operatorname{reg}(h_t) = \tau |\log h_t - \log h_{t-1}|^2$, which has a smoothing effect over the tuned step size schedule. This also makes the objective "more convex," easing optimization. For time $t = 1$, where we don't have $h_{t-1}$, we use a guess $h_0$ instead. Effective values of $\tau$ depend on the type of kernel in question, but not much on the target problem. We thus used a fixed value (App. B) throughout all our experiments.

The high-level workflow of the proposed adaptive SMC scheme is shown in Alg. 1. The notable change is the addition of the adaptation step in Line 3 (colored region), where the tunable parameters to be used at time $t$ are tuned to perform best at the $t$th SMC step, which follows the "pretuning" principle of Buchholz et al. (2021). In contrast, retrospective tuning (Fearnhead & Taylor, 2013), which uses parameters that performed well in the previous step, forces SMC to run with suboptimal parameters at all times.

### 3.3. Algorithm for Step Size Tuning

Recall that for SMC samplers applied to static models (§ 2.2), the path proposal kernel is typically chosen to be an MCMC kernel. For most popular MCMC kernels such as random walk MH (Metropolis et al., 1953; Hastings, 1970) or Metropolis-adjusted Langevin (MALA; Besag, 1994; Rossky et al., 1978), the crucial tunable parameter is a scalar-valued parameter called the *step size* denoted as $h_t > 0$ for $t \in [T]$. In this section, we will describe a general procedure for tuning such step sizes.

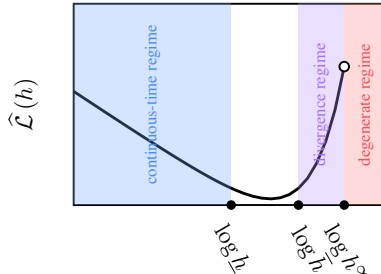

*Figure 1.* **Illustration of Assumption 1.** The solid line is the empirical objective $\widehat{\mathcal{L}}_t$ for the LMC kernel computed using the Bones model from PosteriorDB at time $t = 1$.

**AdaptStepsize.** The adaptation routine is shown in Alg. 2. First, in Line 1 and 2, we convert the optimization space to log-space; from $(0, \infty)$ to $(-\infty, \infty)$. At the SMC iteration $t = 1$, $h_{\text{guess}}$ is provided by the user. Here, it is unsafe to immediately trust $h_{\text{guess}}$ to be non-degenerate ($\mathcal{L}(h_{\text{guess}}) < \infty$). Therefore, FindFeasible in Line 4 ensures that $\mathcal{L}(\exp(\ell)) < \infty$. At time $t > 1$, we set $h_{\text{guess}} = h_{t-1}$, which should be non-degenerate as long as adaptation at time $t - 1$ went successfully. Then we proceed to optimization in Minimize (Alg. 8), which mostly relies on the *golden section search* algorithm (GSS; Avriel & Wilde, 1968; Kiefer, 1953), a gradient-free 1-dimensional optimization method. GSS deterministically achieves an absolute tolerance of $\epsilon > 0$. Since we optimize in log-space, this translates to a natural *relative* tolerance $e^{\pm \epsilon / 2}$ with respect to the minimizer of $\mathcal{L}$. In our implementation and choice of $r, c, \epsilon$ (described in App. B), this procedure terminates after around 10 objective evaluations for $t > 1$ and few tens of iterations for $t = 1$. For an in-depth discussion on the algorithm, please refer to App. C.

### 3.4. Analysis of the Algorithm for Step Size Tuning

We provide quantitative performance guarantees of the presented step size adaptation procedures. To theoretically model various degeneracies that can happen in the large step size regime, we will assume that the objective function $\mathcal{L}$ takes the value of $+\infty$ beyond some threshold. In practice, whenever a numerical degeneracy is detected when evaluating $\log \gamma$ (NaN or $-\infty$), we ensure that the objective value is accordingly set as $\infty$. Our algorithm can deal with such cases by design, as reflected in the following assumptions:

**Assumption 1.** For the objective $\mathcal{L} : (0, \infty) \to \mathbb{R} \cup \{+\infty\}$, we assume the following:

(a) There exists some $h^\infty \in (0, \infty]$ such that $\mathcal{L}$ is finite and continuous on $(0, h^\infty)$ and $+\infty$ on $[h^\infty, \infty)$.

(b) There exists some $\underline{h} \in (0, h^\infty)$ such that $\mathcal{L}$ is strictly monotonically decreasing on $(0, \underline{h}]$.

(c) There exists some $\overline{h} \in [\underline{h}, h^\infty)$ such that $\mathcal{L}$ is strictly monotonically increasing on $[\overline{h}, h^\infty)$

Assumption (a) stipulates that degenerate regions are never disconnected and only exist in the direction of large step sizes. Assumptions (b) and (c) represent the intuition that when the step size is too small or too large, the MCMC kernels degenerate predictably. Most of the MCMC kernels used in practice are based on time-discretized diffusions. In these cases, (b) is satisfied as they approach the continuous-time regime, while (c) will be satisfied as the discretization becomes unstable (divergence). Fig. 1 validates this intuition on one of the examples.

**Theorem 1.** *Suppose Assumption 1 holds. Then* `AdaptStepsize`$(\mathcal{L}, t, h_{\text{guess}}, \delta, c, r, \epsilon)$ *returns a step size* $h \in (0, h^{\infty})$ *that is $\epsilon$-close to a local minimum of $\mathcal{L}$ in log-scale after* $\mathcal{C}_{\text{feas}} + \mathcal{C}_{\text{bm}} + \mathcal{C}_{\text{gss}}$ *objective evaluations, where, defining* $\Delta \triangleq \log_+(\overline{h}/h_0) + \log_+(h_0/\underline{h})$ *and* $h_0 \triangleq \min(h_{\text{guess}}, h^{\infty})$,

$$\mathcal{C}_{\text{feas}} = \mathrm{O}\left\{\delta^{-1} \log_+\left(h_{\text{guess}}/h^{\infty}\right)\right\}$$
$$\mathcal{C}_{\text{bm}} = \mathrm{O}\left\{(\log r)^{-1} \log_+\left(\Delta r c^{-1}\right)\right\}$$
$$\mathcal{C}_{\text{gss}} = \mathrm{O}\left\{\log_+\left(\left(r^3\Delta + r^2 c\right)\epsilon^{-1}\right)\right\} .$$

*Proof.* See the *full proof* in page 28. $\qquad\square$

This suggests that, ignoring the dependence on $r, c$, the objective query complexity of our optimization procedure is $\mathrm{O}\left(\log\left(\Delta/\epsilon\right)\right)$. Here, $\Delta$ represents the difficulty of the problem, where $\Delta \geq \left|\log\overline{h} - \log\underline{h}\right|$. In essence, $\left|\log\overline{h} - \log\underline{h}\right|$ represents how "multimodal" the problem is. In practice, however, many problems result in less pessimistic objective surfaces such as follows:

**Assumption 2.** $\mathcal{L}$ *is unimodal on* $(0, h^{\infty})$.

This is equivalent to assuming (b) and (c) in Assumption 1 with $\overline{h} = \underline{h}$ and implies there is a unique global minimum. Then Theorem 1 can be strengthened into the following:

**Corollary 1.** *Suppose Assumption 2 and 3 hold. Then Theorem 1 holds, where* `AdaptStepsize`$(\mathcal{L}, t, h_{\text{guess}}, \delta, c, r, \epsilon)$ *returns* $h \in (0, h^{\infty})$ *that is $\epsilon$-close to the global optimum* $h^*$ *and* $\Delta = |\log h^* - \log h_0|$.

Note that, at $t > 1$, it is sensible to set $h_{\text{guess}} \leftarrow h_{t-1}$ since $\pi_{t-1} \approx \pi_t$ by design. Therefore, after $t = 1$, `AdaptStepsize` will run in a "warm start" regime where $\Delta \approx 0$. For instance, assume the initial guess is warm such that $|\log h_{\text{guess}} - \log h^*| \leq \epsilon$ and $h_{\text{guess}} \in (0, h^{\infty})$. Then Corollary 1 states that the number of objective evaluations will be $\mathrm{O}\left\{\log_+\left(r^2 c \epsilon^{-1} + r^3\right)\right\}$.

The parameters of `AdaptStepsize`, $r$ and $c$, must balance the performance of both the warm and cold start cases. For a warm start, $cr^{-1} = \mathrm{O}(\epsilon)$ optimizes performance. For a cold start, $r$ needs to be large enough to keep the $(\log r)^{-1}$ term in $\mathcal{C}_{\text{bm}}$ small. Thus, leaning towards making $c$ small and $r$ moderately large balances both cases. The values we use in the experiments are shown in App. B.1.

# 4. Implementations

Based on the procedure in § 3.3, we now describe complete implementations of adaptive SMC samplers. Here, we will focus on the static model setting (§ 2.2), where the main objective is tuning of the MCMC kernels $(K_t^{\theta})_{t\in[T]}$.

## 4.1. SMC with Langevin Monte Carlo

First, we consider SMC with Langevin Monte Carlo (LMC; Grenander & Miller, 1994; Rossky et al., 1978; Parisi, 1981), also known as the unadjusted Langevin algorithm. LMC forms a kernel $K_t : \mathbb{R}^d \times \mathcal{B}\left(\mathbb{R}^d\right) \to \mathbb{R}_{>0}$ on the state space $\mathcal{X} = \mathbb{R}^d$, which, for $s \geq 0$, simulates the Langevin stochastic differential equation (SDE)

$$\mathrm{d}x_s = \nabla \log \pi_t\left(x_s\right)\mathrm{d}s + \sqrt{2}\,\mathrm{d}B_s, \qquad (11)$$

where $(B_s)_{s\geq 0}$ is Brownian motion. Under appropriate conditions on the target $\pi_t$, it is well known that the stationary distribution of the process $(x_s)_{s\geq 0}$ is $\pi_t$, where converges exponentially fast in total variation (Roberts & Tweedie, 1996, Thm 2.1). The Euler-Maruyama discretization of Eq. (11) yields a Markov kernel

$$K_t^h\left(x, \mathrm{d}x'\right) = \mathcal{N}\left(\mathrm{d}x';\, x + h\nabla\log\pi_t\left(x\right), 2h\,\mathrm{I}_d\right),$$

where $h > 0$ is the step size, which conveniently has a tractable density with respect to the Lebesgue measure.

Note that LMC is an *approximate* MCMC algorithm; for any $h > 0$, the stationary distribution of $K_t^h$ is only *approximately* $\pi_t$. This contrasts with its MH-adjusted counterpart MALA (Besag, 1994; Roberts & Tweedie, 1996; Rossky et al., 1978), which is stationary on $\pi_t$.

**Backward Kernel.** For the sequence of backward kernels $(L_{t-1}^{\theta})_{t=2,...,T}$, multiple choices are possible. For instance, in the literature, a typical choice is $L_{t-1}^{h_t} = K_t^{h_t}$. In this work, we instead take the choice of

$$L_{t-1}^{h_{t-1}}\left(x_t, x_{t-1}\right) \triangleq K_{t-1}^{h_{t-1}}\left(x_t, x_{t-1}\right),$$

which we call the "time-correct forward kernel." Compared to more popular alternatives, this choice results in significantly lower variance. (An in-depth discussion can be found in App. E.) The resulting potentials are

$$G_1^{h_1}\left(x_0, x_1\right) = \frac{\gamma_1\left(x_1\right)}{K_1^{h_1}\left(x_0, x_1\right)}$$

$$G_t^{h_{t-1}, h_t}\left(x_{t-1}, x_t\right) = \frac{\gamma_t\left(x_t\right)L_{t-1}^{h_{t-1}}\left(x_t, x_{t-1}\right)}{\gamma_{t-1}\left(x_{t-1}\right)K_t^{h_t}\left(x_{t-1}, x_t\right)},$$

where, at each step $t \in [T]$, we optimize for $h_t$ using the general step size tuning procedure described in § 3.3 while the backward kernel re-uses the tuned parameter $h_{t-1}$ from the previous iteration.

## 4.2. SMC with Kinetic Langevin Monte Carlo

Next, we consider a variant of the LMC that operates on the augmented state space $\mathcal{Z} = \mathcal{X} \times \mathcal{X}$, where, for $t \geq 0$,

**Algorithm 3:** $\mathtt{AdaptKLMC}\left(\mathcal{L}, h_{\mathtt{guess}}, \rho_{\mathtt{guess}}, \delta, \Xi, c, r, \epsilon\right)$

**Input:** Adaptation objective $\mathcal{L} : \mathbb{R}_{>0} \times (0,1) \to \mathbb{R} \cup \{\infty\}$,
     initial guess $(h_{\mathtt{guess}}, \rho_{\mathtt{guess}}) \in \mathbb{R}_{>0} \times (0,1)$,
     backing-off step size $\delta < 0$,
     grid of refreshment parameters $\Xi \in (0,1)^k$,
     exponential search coefficient $c > 0$,
     exponential search exponent $r > 1$,
     absolute tolerance $\epsilon > 0$.
**Output:** Adapted step size and refreshment rate $(h, \rho)$.

1   $\mathcal{L}^{\log}(\ell, \rho) \triangleq \mathcal{L}(\exp(\ell), \rho)$
2   $\ell \leftarrow \log h_{\mathtt{guess}}, \quad \rho \leftarrow \rho_{\mathtt{guess}}$
3   **if** $t = 1$ **then**
4     |   $\ell \leftarrow \mathtt{FindFeasible}\left(\ell \mapsto \mathcal{L}^{\log}(\ell, \rho), \ell, \delta\right)$
5   **end**
6   **while** *not converged* **do**
7     |   $\ell' \leftarrow \mathtt{Minimize}\left(\ell \mapsto \mathcal{L}^{\log}(\ell, \rho), \ell, c, r, \epsilon\right)$
       |   $\rho' \leftarrow \arg\min_{\rho \in \Xi} \mathcal{L}^{\log}(\ell', \rho)$.
8     |   **if** $\max\left(|\ell - \ell'|, |\rho - \rho'|\right) \leq \epsilon$ **then**
9        |   Return $(\exp(\ell'), \rho')$
10    |   **end**
11    |   $\ell \leftarrow \ell', \quad \rho \leftarrow \rho'$
12   **end**
13 Return $(\exp(\ell'), \rho')$

each state of the Feynman-Kac model is denoted as $z_t = (x_t, v_t) \in \mathcal{Z}$, $x_t, v_t \in \mathcal{X}$, $\mathcal{X} = \mathbb{R}^d$, and the target is

$$\pi_t^{\mathtt{klmc}}(x, v) \triangleq \pi_t(x) \mathcal{N}(v; 0_d, I_d).$$

Evidently, the $x$-marginal of the augmented target is $\pi$. Therefore, a Feynman-Kac model targeting $\pi_t^{\mathtt{klmc}}$ is also targeting $\pi$ by design. Kinetic Langevin Monte Carlo (KLMC; Horowitz, 1991; Duane et al., 1987), also referred to as underdamped Langevin, for $s \geq 0$, is given by

$$dx_s = v_s ds$$

$$dv_s = \nabla \log \pi(v_s) ds - \eta v_s ds + \sqrt{2\eta} dB_s,$$

where $\eta > 0$ is a tunable parameter called the *damping coefficient*. The stationary distribution of the joint process $(x_s, v_s)_{s \geq 0}$ is then $\pi^{\mathtt{klmc}}$. This continuous time process corresponds to the "Nesterov acceleration (Nesterov, 1983; Su et al., 2016)" of Eq. (11) (Ma et al., 2021), meaning that the process should converge faster. We thus expect KLMC to reduce the required number of steps $T$ compared to LMC.

To simulate this, we consider the OBABO discretization (Leimkuhler & Matthews, 2013), which operates in a Gibbs scheme (Geman & Geman, 1984): its kernel

$$K_t(z_{t-1}, dz_t)$$
$$= R^\rho\left(v_{t-1}, dv_{t-1/2}\right) S_t^{h,L}\left((x_{t-1}, v_{t-1/2}), (dx_t, dv_t)\right)$$

is a composition of the *momentum refreshment kernel*

$$R^\rho\left(v_{t-1}, dv_{t-1/2}\right) \triangleq \mathcal{N}\left(dv_{t-1/2}; \sqrt{1 - \rho^2}\, v_{t-1}, \rho^2 I_d\right),$$

where $\rho \triangleq 1 - \exp(-\eta h) \in (0,1)$ is the "momentum refreshment rate" for some step size $h > 0$, and the *Leapfrog integrator kernel*

$$S_t^h\left((x_{t-1}, v_{t-1/2}), \cdot\right) \triangleq \delta_{\Phi_t^h(x_{t-1}, v_{t-1/2})}(\cdot),$$

where $\Phi_t^h$ is a single step of leapfrog integration with step

size $h$ preserving the "Hamiltonian energy" $-\log \pi_t^{\mathtt{klmc}}$. This discretization also coincides with the unadjusted version of the generalized Hamiltonian Monte Carlo (Duane et al., 1987; Neal, 2011) with a single leapfrog step.

**Backward Kernel.** Since the kernel $S^{h_t}$ is a deterministic mapping, $K_t^{\theta_t}$ does not admit a density with respect to the Lebesgue measure. Therefore, we are restricted to a specific backward kernel that satisfies the condition in Eq. (3): Since the leapfrog integrator $\Phi_t^h$ is a diffeomorphism, its inverse map $(\Phi_t^h)^{-1}$ exists. Therefore, we can choose

$$L_{t-1}^{h,\rho}(z_t, \cdot) = \delta_{(\Phi_t^h)^{-1}(x_t, v_t)}\left((dx_t, dv_{t-1/2})\right) R^\rho\left(v_{t-1/2}, dv_{t-1}\right)$$

This results in the deterministic component of $K_t$ and $L_{t-1}$ being supported on the same pair of points, ensuring absolute continuity (Doucet et al., 2022; Geffner & Domke, 2023). Then the potential is given by the Radon-Nikodym derivative between the momentum refreshments $R^\rho$ as

$$G_t^{h_t, \rho_t}(z_{t-1}, z_t)$$
$$= \frac{\gamma_t(x_t) \mathcal{N}(v_t; 0_d, I_d) \mathcal{N}\left(v_{t-1}; \sqrt{1 - \rho_t^2}\, v_{t-1/2}, \rho_t^2 I_d\right)}{\gamma_{t-1}(x_{t-1}) \mathcal{N}(v_{t-1}; 0_d, I_d) \mathcal{N}\left(v_{t-1/2}; \sqrt{1 - \rho_t^2}\, v_{t-1}, \rho_t^2 I_d\right)}$$

with two tunable parameters: $(h_t, \rho_t) \in \mathbb{R}_{>0} \times (0,1)$.

**Adaptation Algorithm.** As KLMC has two parameters, we cannot immediately apply the tuning procedure offered in § 3.3. Thus, we will tailor it to KLMC. At $t \in [T]$, we will minimize the incremental KL objective $\widehat{\mathcal{L}}_t(h, \rho)$ through coordinate descent. That is, we alternate between minimizing over $h$ and $\rho$. This is shown in Alg. 3. In particular, $h_t$ is updated using the procedure used in § 3.3, while $\rho_t$ is directly minimized over a grid $\Xi \in (0,1)^k$ of $k$ grid points. As shown in Fig. 4, empirically, the minimizers of $\widehat{\mathcal{L}}_t$ with respect to $\rho_t$ tend to concentrate on the boundary, as if the adaptation problem is determining to "fully refresh" or "not refresh at all." Therefore, the grid $\Xi$ can be made as coarse as $\Xi = \{0.1, 0.9\}$, which is what we use in the experiments.

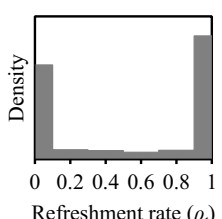

*Figure 4.* **Distribution of tuned refreshment rates** $\rho_t$. The results were obtained by running adaptive SMC on the Sonar problem with $T = 256$ and $N = 1024$

## 5. Experiments

### 5.1. Implementation and General Setup

We implemented our SMC sampler[1] using the Julia language (Bezanson et al., 2017). For resampling, we use the Srinivasan sampling process (SSP) by Gerber et al. (2019), which performs similarly to the popular systematic resam-

---

[1]Link to GITHUB repository: https://github.com/Red-Portal/ControlledSMC.jl/tree/v0.0.4.

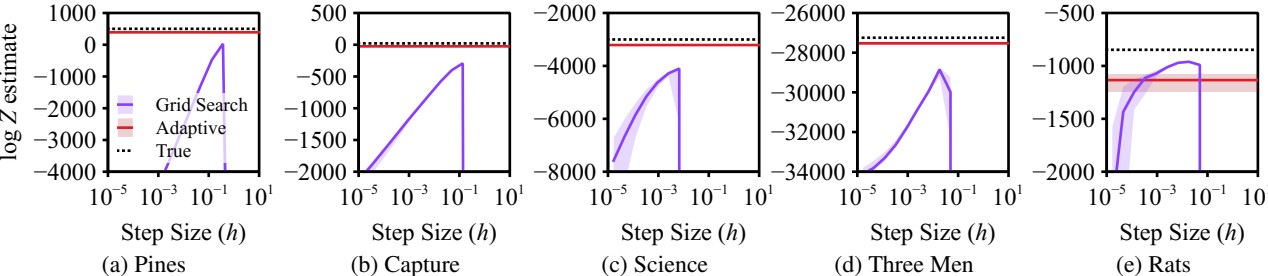

*Figure 2.* **SMC-LMC with adaptive tuning v.s. fixed step sizes.** The solid lines are the median of the estimates of $\log Z$, while the colored regions are the $80\%$ empirical quantiles computed over 32 replications.

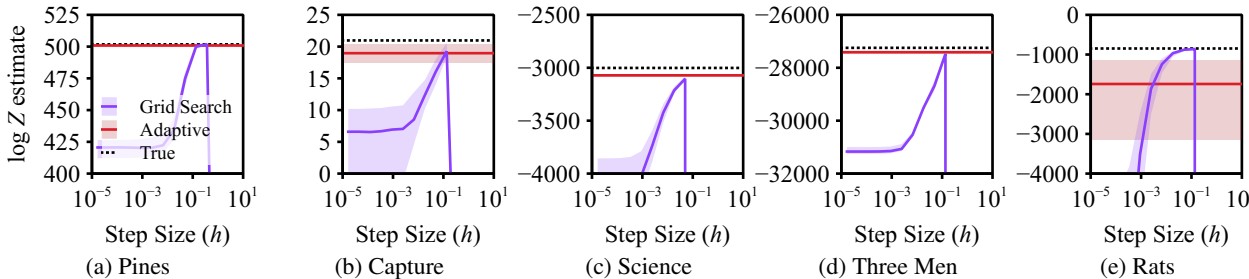

*Figure 3.* **SMC-KLMC with adaptive tuning v.s. fixed step sizes and refreshment rates.** For SMC-KLMC with fixed parameters $h, \rho$, we show the result of the best-performing refreshment rate. The solid lines are the median of the estimates of $\log Z$, while the colored regions are the $80\%$ empirical quantiles computed over 32 replications.

pling strategy (Carpenter et al., 1999; Kitagawa, 1996), while having stronger theoretical guarantees. Resampling is triggered adaptively, which has been theoretically shown to work well (Syed et al., 2024) under the typical rule of resampling as soon as the effective sample size (Kong, 1992; Elvira et al., 2022) goes below $N/2$. In all cases, the reference distribution is a standard Gaussian $q = \mathcal{N}(0_d, I_d)$, while we use a quadratic annealing schedule $\lambda_t = (t/T)^2$.

**Evaluation Metric.** We will compare the estimate $\log \widehat{Z}_{T,N}$, where, for unbiased estimates of $Z$ against a ground truth estimate obtained by running a large budget run with $N = 2^{14}$ and $T = 2^9$. Due to adaptivity, our method only yields *biased* estimates of $Z$. Therefore, after adaptation, we run vanilla SMC with the tuned parameters, which yields unbiased estimates.

**Benchmark Problems.** For the benchmarks, we ported some problems from the Inference Gym (Sountsov et al., 2020) to Julia, where the rest of the problems are taken from PosteriorDB (Magnusson et al., 2025). Details on the problems considered in this work are in App. A, while the configuration of our adaptive method is specified in App. B.

## 5.2. Comparison Against Fixed Step Sizes

**Setup.** First, we evaluate the quality of the parameters tuned through our method. For this, we compare the performance of SMC-LMC and SMC-KLMC against hand tuning a fixed step size $h$, such that $h_t = h$, over a grid of step sizes. For KLMC, we also perform a grid search of the refreshment rate over $\{0.1, 0.5, 0.9\}$. The computational budgets are set as $N = 1024$, $B = 128$, and $T = 64$.

**Results.** A representative subset of the results is shown in Figs. 2 and 3, while the full set of results is shown in App. F.1. First, we can see that SMC with fixed step sizes is strongly affected by tuning. On the other hand, our adaptive sampler obtains estimates that are closer or comparable to the best fixed step size on 20 out of 22 benchmark problems. Our method performed poorly on the Rats problem, which is shown in the right-most panes in Figs. 2 and 3. Overall, our method results in estimates that are better or comparable to those obtained with the best fixed step size.

## 5.3. Comparison Against End-to-End Optimization

**Setup.** Now, we compare our adaptive tuning strategy against end-to-end optimization strategies. In particular, we compare against differentiable AIS (Geffner & Domke, 2023; 2021; Zhang et al., 2021) instead of SMC, as differentiating through resampling does not necessarily improve the results (Zenn & Bamler, 2023). To focus on the tuning capabilities, we do not optimize the reference $q$. However, results with variational reference tuning can be found in App. F.2. Furthermore, we performed a grid search over the SGD step sizes $\{10^{-4}, 10^{-3}, 10^{-2}\}$ and show the best results. Additional implementation details can be found in App. B.2. Since end-to-end methods need to differentiate through the models, we only ran them on the problems with JAX (Bradbury et al., 2018) implementations (Funnel, Sonar, Brownian, and Pine). We use $T = 32$ SMC iterations for all methods. For adaptation, our method uses $B = 128$ particles out of $N = 1024$ particles, while end-to-end optimization uses an SMC sampler with 32 particles during

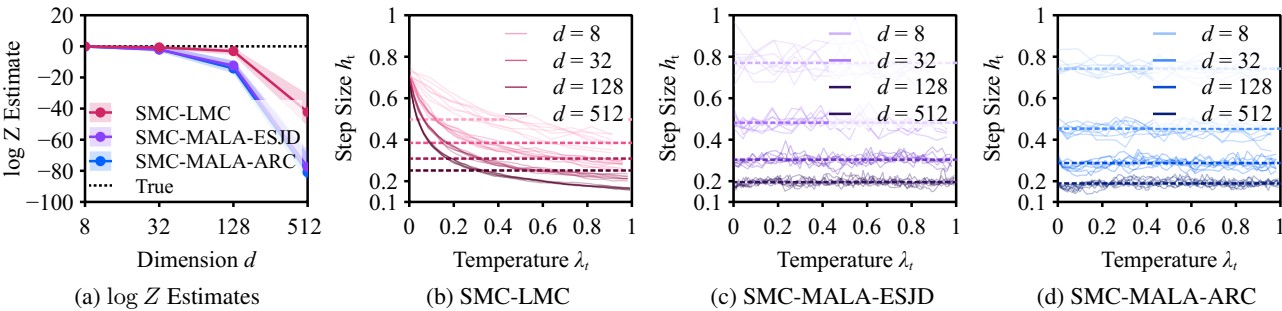

*Figure 5.* **Dimensional scaling of adaptive SMC with Langevin-based kernels with (MALA) and without (LMC) MH adjustment.** (a) Comparison of the $\log Z$ estimates under growing dimensionality. The solid lines are the median, while the shaded regions are the 80% quantiles obtained from 32 replications. (b-d) Tuned step size schedules obtained under each sampler. SMC-MALA-ESJD uses ESJD maximization for adaptation, while SMC-MALA-ARC uses acceptance rate control (ARC). Each solid line is a step size schedule obtained from a single run (eight examples are shown), while the dotted lines are the average over $t$.

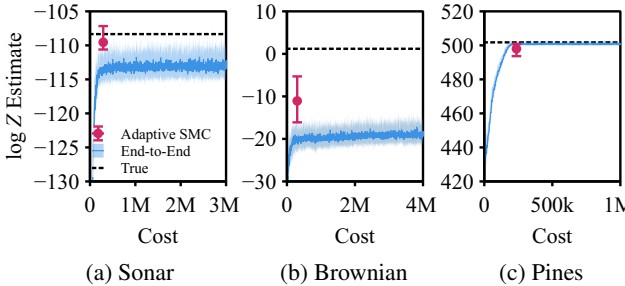

*Figure 6.* **Comparison against end-to-end optimization.** The "cost" is the cumulative number of gradient evaluations of the target. 32 independent runs for end-to-end optimization are shown. The error bars/bands are 80% empirical quantiles of the cost and the estimates of $\log Z$ computed from 32 replications.

optimization, and $N = 1024$ particles when actually estimating $\widehat{Z}_{T,N}$. For both methods, the cost of estimating the unbiased normalizing constant is excluded.

**Results.** The results with the KLMC kernel are shown in Fig. 6, while additional results can be found in App. F.2. Our Adaptive SMC sampler achieves more accurate estimates than the best-tuned end-to-end tuning results on Sonar and Brownian, while the estimate on Pines is comparable. This demonstrates that our SMC tuning approach achieves estimates that are better or on par with those obtained through end-to-end optimization.

### 5.4. Dimensional Scaling with and without Metropolis-Hastings Adjustment

We will now compare the tuned performance of unadjusted versus adjusted kernels, in particular, LMC versus MALA. To maintain a non-zero acceptance rate, MH-adjusted methods generally require $h$ to decrease with dimensionality $d$. Theoretical results suggest that, for MALA, the step size has to decrease as $O(d^{-1/3})$ (Chewi et al., 2021; Roberts & Tweedie, 1996) for Gaussian targets and as $O(d^{-1/2})$ in general (Chewi et al., 2021; Wu et al., 2022). In contrast, LMC only needs to reduce $h$ to counteract the *asymptotic*

bias in the stationary distribution, which grows as $O(d)$ in squared Wasserstein distance (Dalalyan, 2017; Durmus & Eberle, 2024; Durmus & Moulines, 2019). However, since SMC never operates in the stationary regime (except for the waste-free variant by Dau & Chopin 2022), we expect SMC-LMC to scale better than SMC-MALA with dimensionality $d$. Here, we will empirically verify this intuition.

**Setup.** We set $\pi = \mathcal{N}(3 \cdot 1_d, I_d)$ and $q = \mathcal{N}(0_d, I_d)$ under varying dimensionality $d$. The computational budgets are set as $N = 1024$, $T = 4\lceil \sqrt{d} \rceil$, where the latter is suggested by Syed et al. (2024, §4.7). For MALA, we will consider two common adaptation strategies: controlling the acceptance rate (Buchholz et al., 2021) such that it is 0.575 (Roberts & Tweedie, 1996) and maximizing the ESJD (Pasarica & Gelman, 2010; Buchholz et al., 2021; Fearnhead & Taylor, 2013). For both, we use the tricks stated in § 3.2, such as subsampling and SAA.

**Results.** The results are shown in Fig. 5. For SMC-LMC, the step size schedule is shown to decrease with $t$ (Fig. 5b). Since the smoothness constant of the target density does not change with $t$, this means our adaptation scheme is automatically performing a trade-off between convergence speed and asymptotic bias. Also, the average step sizes decrease with $d$, which is expected since the bias grows with $d$. However, for $d \geq 128$, the step sizes of SMC-LMC tend to be larger than those of SMC-MALA (Figs. 5c and 5d). Consequently, SMC-ULA obtains more accurate estimates in higher dimensions (Fig. 5a).

## 6. Conclusions

In this work, we established a methodology for tuning path proposal kernels in SMC samplers, which involves greedily minimizing an incremental KL divergence at each SMC step. We also developed a specific instantiation of the method for tuning scalar-valued step sizes of MCMC kernels used in SMC samplers. A potential future direction would be to investigate the consistency of the proposed scheme, possibly through the framework of Beskos et al. (2016).

## Acknowledgements

The authors sincerely thank Nicolas Chopin for pointing out relevant theoretical results, Alexandre Bouchard-Côté for discussions throughout the project, and the anonymous reviewers for helpful suggestions.

K. Kim and J. R. Gardner were supported through the NSF award [IIS2145644]; T. Campbell and D. Xu were supported by the NSERC Discovery Grant RGPIN-2025-04208. We also gratefully acknowledge the use of the ARC Sockeye computing platform at the University of British Columbia.

## Impact Statement

This paper presents a method for automatically tuning sequential Monte Carlo (SMC) samplers. Due to the technical nature of the work, we do not expect to have direct societal consequences. Downstream applications of SMC samplers, however, include uncertainty quantification (Dai et al., 2022), statistical model comparison (Zhou et al., 2016), and more recently, conditional generation from diffusion models (Wu et al., 2023; Trippe et al., 2023) and steering large language models to ensure their output conforms to ethical constraints (Zhao et al., 2024). Our work may improve the efficiency and efficacy of such tasks, indirectly affecting their societal consequences.

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

# Contents

| Name | Description | $d$ | Source | Reference |
|------|-------------|-----|--------|-----------|
| Funnel | Neal's funnel distribution. | 10 | Inference Gym | Sountsov et al. 2020 Neal 2003 |
| Brownian | Latent Brownian motion with missing observations. | 32 | Inference Gym | Sountsov et al. 2020 |
| Sonar | Bayesian logistic regression with the sonar classification dataset. | 61 | Inference Gym | Sountsov et al. 2020 Gorman & Sejnowski 1988 |
| Pines | Log-Gaussian Cox process model of the concentration of Scotch pine saplings in Finland over a $40 \times 40$ grid. | 1600 | Inference Gym | Sountsov et al. 2020 Møller et al. 1998 |

*Table 1.* Overview of Benchmark Problems

## A. Benchmark Problems

In this section, we provide additional details about the benchmark problems. A full list of the problems is shown in Tables 1 to 3. For the problems we ported from the Inference Gym, we provide additional details for clarity:

**Funnel.** This is the classic benchmark problem by Neal (2003). We use the formulation:

$$y \sim \mathcal{N}\left(0, 3^2\right)$$
$$x \sim \mathcal{N}\left(0_{d-1}, e^y \mathrm{I}_{d-1}\right),$$

where $d = 10$.

**Sonar.** This is a logistic regression problem with a standard normal prior on the coefficients. That is, for $d = 61$, given a dataset $(X, y)$, where $X \in \mathbb{R}^{n,d-1}$ and $y \in \mathbb{R}^n$, the design matrix is augmented with a column containing 1s denoted with $\tilde{X}$ to include an intercept. The data-generating process is

$$\beta \sim \mathcal{N}\left(0_d, \mathrm{I}_d\right)$$
$$y \sim \mathrm{Bernoulli}\big(\sigma(\tilde{X}\beta)\big),$$

where $\sigma(x) \triangleq 1/\left(1 + \mathrm{e}^{-x}\right)$ is the logistic function. Here, we use the sonar classification dataset by Gorman & Sejnowski (1988). The features are pre-processed with $z$-standardization following Phillips et al. (2024).

**Pines.** This is a log-Gaussian Cox process (LGCP; Møller et al., 1998) model applied to a dataset of Scotch pine saplings in Finland (Møller et al., 1998). A LGCP is a non-parametric model of intensity fields, where the observations are assumed to follow a Poisson point process (PPP). Consider a 2-dimensional grid of $n$ cells indexed by $i \in [n]$, each denoted by $S_i \in \mathcal{S}$ and centered on the location $x_i \in \mathbb{R}^d$. The dataset is the number of points contained in the $i$th cell, $y_i \in \mathbb{N}_{\geq 0}$, for all $i \in [n]$, which is assumed to follow a PPP such that

$$y_i \sim \mathrm{Poisson}\left(\int_{S_i} \lambda(x) \, \mathrm{d}x\right)$$

with the intensity field

$$\log \lambda \sim \mathcal{GP}(\mu, k),$$

where $\mathcal{GP}(\mu, k)$ is a Gaussian process prior (GP; Rasmussen & Williams, 2005) with mean $\mu$ and covariance kernel $k : \mathbb{R}^2 \times \mathbb{R}^2 \to \mathbb{R}_{>0}$. We use the grid approximation

$$\int_{S_i} \lambda(x) \, \mathrm{d}x \approx A_i \exp\left(\log \lambda(x_i)\right),$$

where $A_i$ is the area of $S_i$. The likelihood is then

$$\ell(y_i, x_i, \lambda) = \exp\left\{\lambda(x_i) y_i - A_i \exp\left(\lambda(x_i)\right)\right\}.$$

Following Møller et al. (1998), the hyperparameters of the GP are set as

$$\mu = \log(126) - \frac{\sigma^2}{2}$$
$$k(x_i, x_j) = \sigma^2 \exp\left(-\frac{\|x_i - x_j\|_2}{\sqrt{|\mathcal{S}|\beta^2}}\right),$$

where

$$\sigma^2 = 1.91 \quad \text{and} \quad \beta = \frac{1}{33}.$$

The field $[0,1]^2$ is discretized into a $40 \times 40$ grid such that $|\mathcal{S}| = 40^2$ and $A_i = 1/|\mathcal{S}|$. Furthermore, to improve the conditioning of the posterior, we whiten the GP prior (Murray & Adams, 2010, §2.1).

| Name | Description | $d$ | Source | References |
|------|-------------|-----|--------|------------|
| Bones | Latent trait model for multiple ordered categorical responses for quantifying skeletal maturity from radiograph maturity ratings with missing entries. (model: `bones_model`; dataset: `bones_data`) | 13 | PosteriorDB | Magnusson et al. 2025 Spiegelhalter et al. 1996 |
| Surgical | Binomial regression model for estimating the mortality rate of pediatric cardiac surgery. (model: `surgical_model`; dataset: `surgical_data`) | 14 | PosteriorDB | Magnusson et al. 2025 Spiegelhalter et al. 1996 |
| HMM | Hidden Markov model with a Gaussian emission applied to a simulated dataset. (model: `hmm_gaussian`; dataset: `hmm_gaussian_simulated`) | 14 | PosteriorDB | Magnusson et al. 2025 Cappé et al. 2005 |
| Loss Curves | Loss model of insurance claims. The model is the single line-of-business, single insurer (SISLOB) variant, where the dataset is the "ppauto" line of business, part of the "Schedule P loss data" provided by the Casualty Actuarial Society. (model: `losscurve_sislob`; dataset: `loss_curves`) | 15 | PosteriorDB | Magnusson et al. 2025 Cooney 2017 |
| Pilots | Linear mixed effects model with varying intercepts for estimating the psychological effect of pilots when performing flight simulations on various airports. (model: `pilots`; dataset: `pilots`) | 18 | PosteriorDB | Magnusson et al. 2025 Gelman & Hill 2007 |
| Diamonds | Log-log regression model for the price of diamonds with highly correlated predictors. (model: `diamonds`; dataset: `diamonds`) | 26 | PosteriorDB | Magnusson et al. 2025 Wickham 2016 |
| Seeds | Random effect logistic regression model of the seed germination proportion of seeds from different root extracts. We use the variant with a half-Cauchy prior on the scale. (model: `seeds_stanified_model`; dataset: `seeds_data`) | 26 | PosteriorDB | Magnusson et al. 2025 Crowder 1978 Spiegelhalter et al. 1996 |
| Downloads | Prophet time series model applied to the download count of `rstan` over time. The model is an additive combination of (i) a trend model, (ii) a model of seasonality, and (iii) a model for events such as holidays. (model: `prophet`; dataset: `rstan_downloads`) | 62 | PosteriorDB | Magnusson et al. 2025 Taylor & Letham 2018 Bales et al. 2019 |
| Rats | Linear mixed effects model with varying slopes and intercepts for modeling the weight of young rats over five weeks. (model: `rats_model`; data: `rats_data`) | 65 | PosteriorDB | Magnusson et al. 2025 Spiegelhalter et al. 1996 Gelfand et al. 1990 |
| Radon | Multilevel mixed effects model with log-normal likelihood and varying intercepts for modeling the radon level measured in U.S. households. We use the Minnesota state subset. (model: `radon_hierarchical_intercept_centered`; dataset: `radon_mn`) | 90 | PosteriorDB | Magnusson et al. 2025 Gelman et al. 2014 |

*Table 2.* Overview of Benchmark Problems

| Name | Description | $d$ | Source | Reference |
|------|-------------|-----|--------|-----------|
| Election88 | Generalized linear mixed effects model of the voting outcome of individuals at the 1988 U.S. presidential election. (model: `election88_full`; dataset: `election88`) | 90 | PosteriorDB | Magnusson et al. 2025 Gelman & Hill 2007 |
| Butterfly | Multispecies occupancy model with correlation between sites. The dataset contains counts of butterflies from twenty grassland sites in south-central Sweden (model: `butterfly`; dataset: `multi_occupancy`) | 106 | PosteriorDB | Magnusson et al. 2025 Dorazio et al. 2006 |
| Birds | Mixed effects model with a Poisson likelihood and varying intercepts for modeling the occupancy of the Coal tit (*Parus ater*) bird species during the breeding season in Switzerland. (model: `GLMM1_model`; dataset: `GLMM_data`) | 237 | PosteriorDB | Magnusson et al. 2025 Kéry & Schaub 2012 |
| Drivers | Time series model with seasonal effects of driving-related fatalities and serious injuries in the U.K. from Jan. 1969 to Dec. 1984. (model: `state_space_stochastic_level_stochastic_seasonal`; dataset: `uk_drivers`) | 389 | PosteriorDB | Magnusson et al. 2025 Commandeur & Koopman 2007 |
| Capture | Model of capture-recapture data for estimating the population size. This is the "heterogeneity model," where the detection probability is assumed to be heterogeneous across the individuals. The data is simulated. (model: `Mh_model`; dataset: `Mh_data`) | 388 | PosteriorDB | Magnusson et al. 2025 Kéry & Schaub 2012 |
| Science | Item response model with generalized rating scale. The dataset was taken from the Consumer Protection and Perceptions of Science and Technology section of the 1992 Euro-Barometer Survey. (model: `grsm_latent_reg_irt`; dataset: `science_irt`) | 408 | PosteriorDB | Magnusson et al. 2025 Reif & Melich 1993 Furr 2017 |
| Three Men | Latent Dirichlet allocation for topic modeling. The number of topics is set as $K = 2$, while the dataset is corpus 3 among pre-processed multilingual corpora of the book "Three Men and a Boat." (model: `ldaK2`; dataset: `three_men3`) | 505 | PosteriorDB | Magnusson et al. 2025 Farkas 2014 Blei et al. 2003 |
| TIMSS | Item response model with generalized partial credit. The dataset is from the TIMSS 2011 mathematics assessment of Australian and Taiwanese students. (model: `gpcm_latent_reg_irt`; dataset: `timssAusTwn_irt`) | 530 | PosteriorDB | Magnusson et al. 2025 Muraki 1997 Mullis et al. 2012 |

*Table 3.* Overview of Benchmark Problems (continued)

# B. Details on the Experimental Setup

## B.1. Setup of Adaptive SMC Samplers

**Configuration of the Adaptation Procedure.** Here, we collected the specifications of the tunable parameters in our adaptive SMC samplers. The parameters of SMC-LMC are set as in Table 4:

| Name | Source | Value |
|:---:|:---:|:---:|
| $\tau$ | § 3.2 | 0.1 |
| $\epsilon$ | Alg. 6 | 0.01 |
| $c$ | Alg. 7 | 0.1 |
| $r$ | Alg. 7 | 2 |
| $\delta$ | Alg. 5 | $-1$ |
| $h_{\text{guess}}$ | Alg. 5 | $\exp(-10) \approx 4.54 \times 10^{-5}$ |

*Table 4.* Configuration of SMC-LMC

The parameters of SMC-KLMC are set as in Table 5:

| Name | Source | Value |
|:---:|:---:|:---:|
| $\tau$ | § 3.2 | 5 |
| $\epsilon$ | Alg. 6 | 0.01 |
| $c$ | Alg. 7 | 0.01 |
| $r$ | Alg. 7 | 3 |
| $\delta$ | Alg. 5 | $-1$ |
| $\Xi$ | Alg. 3 | $\{0.1, 0.9\}$ |
| $\rho_{\text{guess}}$ | Alg. 3 | 0.1 |
| $h_{\text{guess}}$ | Alg. 5 | $\exp(-7.5) \approx 5.53 \times 10^{-4}$ |

*Table 5.* Configuration of SMC-KLMC

**Schedule Adaptation.** In some of the experimental results in the appendix, we evaluate the performance of our step size adaptation procedure when combined with an annealing temperature schedule $((\lambda_t)_{t=0,\ldots,T})$ adaptation scheme. In particular, we use the recently proposed method of Syed et al. (2024), which is able to tune both the schedule $(\lambda_t)_{t=0,\ldots,T}$ and the number of SMC steps $T$. Under regularity assumptions, the resulting adaptation schedule asymptotically ($N \to \infty$ and $T \to \infty$) approximates the optimal geometric annealing path that minimizes the variance of the normalizing constant estimator. For a detailed description, see Syed et al. (2024, Sec. 5). Below, we provide a concise description of the schedule adaptation process.

Under suitable regularity assumptions, the asymptotically optimal schedule is the one that makes the "local communication barrier"

$$\text{LCB}(\lambda_{t-1}, \lambda_t) \approx \sqrt{\text{R}(\pi_{t-1} \otimes K_t^\theta \| \pi_t \otimes L_{t-1}^\theta)}$$

uniform across all adjacent steps $\lambda_t, \lambda_{t-1}$ for all $t \in [T]$ (Syed et al., 2024, §4.3). As such, the corresponding adaptation scheme estimates the local communication barrier and uses it to obtain a temperature schedule that makes it uniform. Intuitively, $\text{LCB}(\lambda_{t-1}, \lambda_t)$ quantifies the "difficulty" of approximating $\pi_t \otimes L_{t-1}^\theta$ using weighted particles drawn from $\pi_{t-1} \otimes K_t^\theta$.

In addition, let us denote the local communication barrier accumulated up to time step $t \in \{0, \ldots, T\}$,

$$\Lambda(\lambda_t) \triangleq \sum_{s=1}^{t} \text{LCB}(\lambda_{s-1}, \lambda_s) \, .$$

This serves as a divergence measure for the "length" of the annealing path from $\lambda_0 = 0$ to $\lambda_t$. Furthermore, the *total accumulated local barrier*

$$\Lambda \triangleq \Lambda(\lambda_T) \, ,$$

which is referred to as the *global communication barrier*, quantifies the total difficulty of simulating the annealing path $(\pi_t)_{t \in \{0,\ldots,T\}}$. For the normalizing constant to be accurate, SMC needs to operate in what they call the "stable discretization regime," which occurs at $T = \text{O}(\Lambda)$ (Syed et al., 2024). Therefore, for tuning the number of SMC steps, a good heuristic is to set $T$ to be a constant multiple of the estimated global communication barrier.

The corresponding schedule adaptation scheme is as follows: From the estimates of the communication barrier $(\widehat{\Lambda}(\lambda_t))_{t \in [T]}$ obtained from a previous run, the updated schedule for the *next* run of length $T'$ is set via mapping

$$\lambda_t^\star = \widehat{\Lambda}_{\text{inv}}\left(\widehat{\Lambda} \times \frac{t}{T'}\right), \quad t' = 0, \ldots, T', \quad (12)$$

where the inverse mapping $\widehat{\Lambda}_{\text{inv}}$ is approximated using a monotonic spline with knots $\{(\widehat{\Lambda}(\lambda_t), \lambda_t)\}_{t=0}^{T}$. In our case, the length of the new schedule is set as $T' = 2\widehat{\Lambda}$.

Below, we summarize the general steps for adaptive SMC with round-based schedule adaptation:

---

**Algorithm 4:** Round-Based Annealing Schedule Adaptation

---

**Input:** Number of rounds $r_{\max}$,
       Initial number of SMC iterations $T_1$.
       Initial schedule $(\lambda_t^1)_{t=0,\ldots,T_1}$.

1  **for** $r = 1 \ldots, r_{\max}$ **do**
2      Run adaptive SMC with the schedule $(\lambda_t^r)_{t=0,\ldots,T_r}$.
3      Estimate $\widehat{\Lambda}$ and compute $\widehat{\Lambda}_{\text{inv}}$.
4      Set $T_{r+1} = 2\widehat{\Lambda}$.
5      Obtain $(\lambda_t^{r+1})_{t=0,\ldots,T_{r+1}}$ using Eq. (12).
6  **end**

---

## B.2. Setup of End-to-End Optimization

We provide additional implementation details for end-to-end optimization[2]. We implemented two differentiable AIS methods: one based on LMC (Thin et al., 2021) and another based on KLMC (Geffner & Domke, 2021; Doucet et al., 2022). Both methods are implemented in JAX (Bradbury et al., 2018), modified from the code provided by Geffner & Domke (2023).

For optimization, we used the Adam optimizer (Kingma & Ba, 2015) with three different learning rates $\{10^{-4}, 10^{-3}, 10^{-2}\}$ for 5,000 iterations, with a batch size of 32. We evaluated two different annealing step sizes (32 and 64), keeping the number of steps fixed during training while optimizing the annealing schedule (detailed in App. F.2). Each setting was repeated 32 times, and we report results from the best-performing configurations.

Following the setup of Doucet et al. (2022), the vector-valued step sizes are amortized through a function $\epsilon_\theta(t) : [0, 1] \to \mathbb{R}^d$. This function is parametrized through a 2-layer fully connected neural network with 32 hidden units and ReLU activation, followed by a scaled sigmoid function which enforces $\epsilon_\theta(t) < 0.1$ for the ULA variant and $\epsilon_\theta(t) < 0.25$ for the KLMC variant. Enforcing these step size constraints is necessary to prevent numerical issues during training, which was acknowledged in prior works (Doucet et al., 2022; Geffner & Domke, 2021).

For schedule adaptation, Doucet et al. (2022); Geffner & Domke (2021), parametrize the temperature schedule as

$$\lambda_t = \frac{\sum_{t' \leq t} \sigma\left(b_{t'}\right)}{\sum_{t'=1}^{T} \sigma\left(b_{t'}\right)},$$

where $\sigma$ is the sigmoid function, $\lambda_0$ is fixed to be 0, and $b_{0:T-1}$ is subject to optimization. Following Doucet et al. (2022), we additionally learn the momentum refreshment rate $\rho$ (shared across $t \in [T]$) for SMC-KLMC. That is, we parametrize $\rho$ with a parameter $u$ as $\rho = .98\sigma(u) + .01$, which ensures $\rho \in (0.01, 0.99)$,

---

[2]Link to GITHUB repository: https://github.com/zuhengxu/dais-py/releases/tag/v1.1

---

**Algorithm 5:** FindFeasible $(f, x_0, \Delta)$

---

**Input:** Objective $f : \mathbb{R} \to \mathbb{R}$,
   initial guess $x_0 \in \mathbb{R}$,
   backing off step size $\delta \in \mathbb{R} \setminus \{0\}$.
**Output:** Feasible initial point $x_0$

1   $x \leftarrow x_0$
2   **while** $f(x) = \infty$ **do**
3      $x \leftarrow x + \delta$
4   **end**
5   Return $x$

---

## C. Algorithms

In this section, we will provide a detailed description of our proposed adaptation algorithms and their components.

### C.1. FindFeasible (Algorithm 5)

Our adaptation schemes receive a guess from the user. For robustness, it is safe to assume that this guess may not be non-degenerate. As such, we must first check that it is non-degenerate, and if it is not, move it to somewhere that is. This is done by FindFeasible $(f, x_0, \delta)$ shown in Alg. 5. If $x_0$ is already non-degenerate, it immediately returns the initial point $x_0$. Otherwise, if $x_0$ is degenerate, it increases or decreases $x_0$ with a step size of $\delta$ until the objective function becomes finite.

### C.2. GoldenSectionSearch (Alg. 6)

The workhorse of our step size adaptation scheme is the golden section search (GSS) algorithm (Avriel & Wilde, 1968), which is a variation of the Fibonacci search algorithm (Kiefer, 1953). In particular, we are using the implementation of Press et al. (1992, §10.1), shown in Alg. 6, which uses a *triplet*, $(a, b, c) \in \mathbb{R}^3$, for initialization. This triplet requires the condition

$$a < b < c, \quad f(b) < f(a), \text{ and } f(b) < f(c) \quad (13)$$

to hold. Then, by Lemma 1, this implies that the open interval $(a, c)$ contains a local minimum. Then GSS is guaranteed to find the contained local minimum at a "linear rate" of $(1 - \sqrt{5})/2 \approx 1.62$, the golden ratio. For finding a point $\epsilon$-close to a local minimum, this translates into an objective query complexity of $\mathcal{O}(\log |c - a|/\epsilon)$. Furthermore, if $f$ is unimodal (Assumption 2), then the solution will be $\epsilon$-close to the global minimum (Luenberger & Ye, 2008, §7.1). The key is to find a triplet $(a, b, c)$ satisfying the condition in Eq. (13), which is done in Alg. 7 in the next section.

---

**Algorithm 6:** GoldenSectionSearch $(f, a, b, c, \epsilon)$

---

**Input:** Objective $f : \mathbb{R} \to \mathbb{R} \cup \{+\infty\}$,
   initial triplet $(a, b, c) \in \mathbb{R}^3$ satisfying Eq. (13),
   absolute tolerance $\epsilon > 0$.

1   $\phi^{-1} \triangleq (\sqrt{5} - 1)/2$
2   $x_0 \leftarrow a$
3   $x_3 \leftarrow c$
4   **if** $|c - b| > |b - a|$ **then**
5      $x_1 \leftarrow b$
6      $x_2 \leftarrow b + (1 - \phi^{-1})(c - b)$
7   **else**
8      $x_2 \leftarrow b$
9      $x_1 \leftarrow b - (1 - \phi^{-1})(b - a)$
10   **end**
11   $f_1 \leftarrow f(x_1)$
12   $f_2 \leftarrow f(x_2)$
13   **while** $|x_1 - x_2| > \epsilon/2$ **do**
14      **if** $f_2 < f_1$ **then**
15          $x_0 \leftarrow x_1$
16          $x_1 \leftarrow x_2$
17          $x_2 \leftarrow \phi^{-1} x_2 + (1 - \phi^{-1}) x_3$
18          $f_1 \leftarrow f_2$
19          $f_2 \leftarrow f(x_2)$
20      **else**
21          $x_3 \leftarrow x_2$
22          $x_2 \leftarrow x_1$
23          $x_1 \leftarrow \phi^{-1} x_1 + (1 - \phi^{-1}) x_0$
24          $f_2 \leftarrow f_1$
25          $f_1 \leftarrow f(x_1)$
26      **end**
27   **end**
28   **if** $f_1 \leq f_2$ **then**
29      Return $x_1$
30   **else**
31      Return $x_2$
32   **end**

---

---

**Algorithm 7:** BracketMinimum $(f, x_0, c, r)$

---

**Input:** Objective $f : \mathbb{R} \to \mathbb{R} \cup \{+\infty\}$,
  initial point $x_0 \in (-\infty, x^\infty)$,
  exponential search coefficient $c > 0$,
  exponential search base $r > 1$.
**Output:** Triplet $(x^-, x_{\mathsf{mid}}, x^+)$

---

1 $x \leftarrow x_0$
2 $y \leftarrow f(x)$
3 $k \leftarrow 0$
4 **while** *true* **do**
5     $x' \leftarrow x_0 + cr^k$
6     $y' \leftarrow f(x')$
7     **if** $y < y'$ **then**
8        $x^+ \leftarrow x'$
9        $x_0 \leftarrow x$
10        break
11     **end**
12     $x \leftarrow x'$
13     $y \leftarrow y'$
14     $k \leftarrow k + 1$
15 **end**
16 $k \leftarrow 0$
17 **while** *true* **do**
18     $x' \leftarrow x_0 - cr^k$
19     $y' \leftarrow f(x')$
20     **if** $y < y'$ **then**
21        $x^- \leftarrow x'$
22        $x_{\mathsf{mid}} \leftarrow x$
23        break
24     **end**
25     $x \leftarrow x'$
26     $y \leftarrow y'$
27     $k \leftarrow k + 1$
28 **end**
29 Return $(x^-, x_{\mathsf{mid}}, x^+)$

---

### C.3. BracketMinimum (Algorithm 7)

The main difficulty of applying GSS in practice is setting the initial bracketing interval. If the bracketing interval does not contain a local minimum, nothing can be said about what GSS is converging towards. In our case, we require a triplet $(a, b, c)$ that satisfies the sufficient conditions in Lemma 2. Therefore, an algorithm for finding such an interval is necessary. Naturally, this algorithm should have a computational cost that is better or at least comparable to GSS. Otherwise, a more naive way of setting the intervals would make more sense. Furthermore, the width of the interval found by the algorithm should be as narrow as possible so that GSS can be run more efficiently.

While Press et al. (1992, §10.1) presents an algorithm for finding such a bracket using parabolic interpolation, the

---

**Algorithm 8:** Minimize $(f, x_0, c, r, \epsilon)$

---

**Input:** objective $f : \mathbb{R} \to \mathbb{R} \cup \{+\infty\}$,
  initial point $x_0 \in \mathbb{R}$ such that $f(x_0) < \infty$,
  exponential search coefficient $c > 0$,
  exponential search exponent $r > 1$,
  absolute tolerance $\epsilon > 0$.

---

1 $(x^-, x_{\mathsf{int}}, x^+) \leftarrow$ BracketMinimum $(f, x_0, c, r)$
2 $x^* \leftarrow$ GoldenSectionSearch $(f, x^-, x_{\mathsf{int}}, x^+, \epsilon)$

---

efficiency and quality of the output of this algorithm are not analyzed. Furthermore, the presence of discontinuities in our objective function (Assumption 3) warrants a simpler algorithm that is provably robust. Therefore, we use a specialized routine, BracketMinimum, shown in Alg. 7.

BracketMinimum works in two stages: Given an initial point $x_0$, it expands the search interval to the right (towards $+\infty$; Line 4-15) and then to the left (towards $-\infty$; Line 17-28). During this, it generates a sequence of exponentially increasing intervals (Lines 5 and 18) and, in the second stage, stops when it detects points that satisfy the condition in Equation (13). This algorithm was inspired by a Stack Exchange post by Lavrov (2017), which was in turn inspired by the exponential search algorithm (Bentley & Yao, 1976).

Most tunable parameters of the adaptation method in § 3.3 come from BracketMinimum. In fact, the parameters of BracketMinimum most crucially affect the overall computational performance of our schemes. Recall that the convergence speed of GSS depends on the width of the provided triplet, $|a - c|$. Given, this $c$ and $r$ affect performance as follows: (a) The width of the resulting triplet, $|a - c|$, increases with $r$ and $c$. (b) Smaller $r$ and $c$ requires more time to find a valid triplet. For a discussion on how to set these parameters, see § 3.4.

### C.4. Minimize (Algorithm 8)

We finally discuss our complete optimization routine, which is shown in Alg. 8. Given an initial point $x_0$ and suitable assumptions, Minimize $(f, x_0, c, r, \epsilon)$ finds a point that is $\epsilon$-close to a local minimum. This is done by first finding an interval that contains the minimum (Line 1) by calling BracketMinimum, which is then used by GoldenSectionSearch for proper optimization (Line 2). As such, the computation cost of the routine is the sum of the two stages.

There are four parameters: $x_0, c, r, \epsilon$. Admissible values of $\epsilon$ will depend on the requirements of the downstream task. On the other hand, $c$ and $r$ can be optimized. The effect of these parameters on the execution time is analyzed in Theorem 2, while a discussion on how to interpret the theoretical analysis is in § 3.4.

# D. Theoretical Analysis

In this section, we will provide a formal theoretical analysis of the algorithms presented in App. C as well as the omitted proof of the theorems in the main text.

## D.1. Definitions and Assumptions

Formally, when we say "local minimum," we follow the following definition:

**Definition 1** (Definition 7; Rudin, 1976)**.** Consider some continuous function $f : \mathcal{X} \to \mathbb{R}$ on a metric space $\mathcal{X} \subseteq \mathbb{R}$. We say $f$ has a local minimum at $x^* \in \mathcal{X}$ if there exists some $\epsilon > 0$ such that $f(x^*) \leq f(x)$ for all $x \in \mathcal{X}$ with $|x^* - x| < \epsilon$.

Also, unimodal functions are defined as follows:

**Definition 2.** We say $f : [a,b] \to \mathbb{R}$ is unimodal if there exists some point $x^*$ such that $f$ is monotonically strictly decreasing on $[a, x^*]$ and strictly increasing on $[x^*, b]$.

Now, recall that our adaptation objective in § 3.2 operates on $\mathbb{R}_{>0}$. During adaptation, however, the objectives are log-transformed so that optimization is performed on $\mathbb{R}$. Therefore, it is convenient to assume everything happens on $\mathbb{R}$. That is, instead of Assumption 1 and 2, we will work with the following assumptions that are equivalent up to log transformation:

**Assumption 3.** For the objective $f : \mathbb{R} \to \mathbb{R} \cup \{+\infty\}$, we assume the following:

(a) There exists some $x^\infty \in (-\infty, \infty]$ such that $x$ is finite and continuous on $(-\infty, x^\infty)$ and $\infty$ on $[x^\infty, \infty)$.

(b) There exists some $\underline{x} \in (-\infty, x^\infty)$ such that $f$ is strictly monotonically decreasing on $(-\infty, \underline{x}]$.

(c) There exists some $\overline{x} \in [\underline{x}, x^\infty)$ such that $f$ is strictly monotonically increasing on $[\overline{x}, x^\infty)$

**Assumption 4.** $f$ is unimodal on $(-\infty, x^\infty)$.

Evidently, these assumptions are equivalent to Assumption 1 and 2 by setting

$$f(x) = \mathcal{L}(\exp(x))$$
$$\overline{x} = \log \overline{h}$$
$$\underline{x} = \log \underline{h}$$
$$x^\infty = \log h^\infty .$$

## D.2. Proof of Proposition 1

**Proposition 1.** *Consider joint distributions* $Q_{0:T}, P_{0:T}$. *Then* $\mathrm{D}_{\mathsf{path}}$ *satisfies the following:*

(i) $\mathrm{D}_{\mathsf{path}}(P_{0:T}, Q_{0:T}) \geq 0$ *for any* $Q_{0:T}, P_{0:T}$.

(ii) $\mathrm{D}_{\mathsf{path}}(P_{0:T}, Q_{0:T}) = 0$ *if and only if* $P_{0:T} = Q_{0:T}$.

*Proof.* (i) is trivial. (ii) follows from the fact that if $P_{0:T} = Q_{0:T}$, the incremental KL divergences are all 0, while if $P_{0:T} \neq Q_{0:T}$, $\mathrm{D}_{\mathsf{path}}(Q_{0:T}, P_{0:T}) \geq \mathrm{D}_{\mathsf{path}}(Q_{t|0:t-1}, P_{t|0:t-1}) > 0$ for any $t \in [T]$ by the fact that the conditional KL divergence is 0 if and only if $Q_{t|0:t-1} = P_{t|0:t-1}$. $\square$

## D.3. Sufficient Condition for an Interval to Contain a Local Minimum (Lemma 1)

Our adaptation algorithms primarily rely on GSS (Algorithm 6) to identify a local optimum. To guarantee this, however, GSS needs to be initialized on an interval that contains a local minimum. For this, the following lemma establishes a sufficient condition for identifying such intervals. As such, we will use these conditions as invariants during the execution of GSS, such that it finds narrower and narrower intervals that continue to contain a local minimum.

**Lemma 1.** *Suppose* $f$ *satisfies Assumption 3 and there exist some* $a < b < c$ *such that*

$$f(b) \leq f(a) < \infty \quad and \quad f(b) \leq f(c) .$$

*Then* $(a, c)$ *contains a local minimum of* $f$ *on* $(-\infty, x^\infty)$.

*Proof.* Consider any triplet $(a', b', c')$ consisting of three points $a' < b' < c'$ with $f(b') \leq f(a') < \infty$ and $f(b') \leq f(c') < \infty$. Then $(a', c')$ contains a local minimum of $f$: $f$ attains its minimum on $[a', c']$ by the extreme value theorem, which is either on $(a', c')$—in which case the result holds immediately—or on $\{a', c'\}$—in which case the result holds because $b'$ is a local minimum since $f(b') \leq \min\{f(a'), f(c')\}$.

We now apply this result to triplets contained in $[a, c]$. First, if $c < x^\infty$, use the triplet $(a', b', c') = (a, b, c)$. For the remaining cases, assume $c \geq x^\infty$. If $b \geq \overline{x}$, set the triplet $(a', b', c') = (a, b, d)$ for any $d \in (b, x^\infty)$. If $b < \overline{x}$ and $f(\overline{x}) \geq f(b)$, set the triplet $(a', b', c') = (a, b, \overline{x})$. Otherwise, if $b < \overline{x}$ and $f(\overline{x}) < f(b)$, set the triplet $(a', b', c') = (b, \overline{x}, d)$ for any $d \in (\overline{x}, x^\infty)$. $\square$

### D.4. `GoldenSectionSearch` (Lemma 2)

We first establish that, under suitable initialization, GSS is able to locate a local minimum. Most existing results assume that $f$ is unimodal (Luenberger & Ye, 2008, §7.1) and show that GSS converges to the unique global minimum. Here, we prove a more general result that holds under weaker conditions: GSS can also find a local minimum even when unimodality doesn't hold. For this, we establish that our assumptions in Assumption 3 and initializing at a triplet $(a, b, c)$ satisfying the condition in Lemma 1 are sufficient. Furthermore, while it is well known that GSS achieves a linear convergence rate with coefficient $\phi \triangleq (1 + \sqrt{5})/2$, we could not find a proof that exactly applied to the GSS variant by Press et al. (1992), which is the one we use. Therefore, we also prove linear convergence rate with a proof that precisely applies to Algorithm 6.

**Lemma 2.** *Suppose Assumption 3 holds. Then, for any triplet $(a, b, c)$ satisfying $a < b < c$, $f(b) \leq f(a) < \infty$, and $f(b) \leq f(c)$, $\texttt{GoldenSectionSearch}(f, a, b, c, \epsilon)$ returns a point $x^* \in (-\infty, x^\infty)$ that is $\epsilon$-close to a local minimum after*

$$\Theta\left(\log|c - a|\frac{1}{\epsilon}\right)$$

*objective evaluations, where $\phi = (1 + \sqrt{5})/2$.*

*Proof.* For clarity, let us denote the value of the variables $x_0, x_1, x_2, x_3$ set at iteration $k \geq 1$ of the while loop in Line 13-27 as $x_0^k, x_1^k, x_2^k, x_3^k$. Before the while loop at $k = 0$, they are initialized as follows: If $|c - b| \geq |b - a|$,

$$\left(x_0^0,\ x_1^0,\ x_2^0,\ x_3^0\right) = \left(a,\ b,\ b + \left(1 - \phi^{-1}\right)(c - b),\ c\right)$$

and

$$\left(x_0^0,\ x_1^0,\ x_2^0,\ x_3^0\right) = \left(a,\ b + \left(1 - \phi^{-1}\right)(b - a),\ b,\ c\right)$$

otherwise. For all $k \geq 0$, the following set of variables is set as follows: If $f\left(x_2^k\right) < f\left(x_1^k\right)$, the next set of variables is set as

$$
\begin{aligned}
&\left(x_0^{k+1},\ x_1^{k+1},\ x_2^{k+1},\ x_3^{k+1}\right) \\
&\triangleq \left(x_1^k,\ x_2^k,\ \phi^{-1}x_2^k + \left(1 - \phi^{-1}\right)x_3^k,\ x_3^k\right),
\end{aligned}
\tag{14}
$$

and

$$
\begin{aligned}
&\left(x_0^{k+1},\ x_1^{k+1},\ x_2^{k+1},\ x_3^{k+1}\right) \\
&\triangleq \left(x_0^k,\ \phi^{-1}x_1^k + \left(1 - \phi^{-1}\right)x_0^k,\ x_1^k,\ x_2^k\right)
\end{aligned}
$$

otherwise. We also denote $f_2^k \triangleq f\left(x_2^k\right)$ and $f_1^k \triangleq f\left(x_1^k\right)$.

Assuming the algorithm terminates at some $k^* < \infty$, the algorithm outputs either $x_1^{k^*}$ or $x_2^{k^*}$. Therefore, it suffice to show that $k^* < \infty$, $\left|x_2^{k^*} - x_1^{k^*}\right| \leq \epsilon/2$, and that the interval $\left(x_0^{k^*}, x_3^{k^*}\right)$ contains a local minimum.

First, let's establish that $k^* < \infty$. GSS terminates as soon as $\left|x_3^k - x_0^k\right| \leq \epsilon$ for some $0 \geq k < \infty$. We will establish this by showing that $\left|x_3^k - x_0^k\right|$ satisfies a contraction. For this, however, we first have to show that $x_1^k, x_2^k$ satisfy

$$x_1^k = \phi^{-1}x_0^k + \left(1 - \phi^{-1}\right)x_3^k \tag{15}$$

$$x_2^k = \left(1 - \phi^{-1}\right)x_0^k + \phi^{-1}x_3^k \tag{16}$$

at all $k \geq 0$. We will show this via induction. Before we proceed, notice that the name "golden" section search comes from the fact that $\phi$, the golden ratio, is the solution to the equation

$$\phi^2 = \phi + 1 \quad \Rightarrow \quad 1 - \phi^{-1} = \phi^{-2}. \tag{17}$$

Now, for some $k > 0$, suppose Equations (15) and (16) hold. Then, if $f_2^k < f_1^k$,

$$
\begin{aligned}
x_1^{k+1} &= x_2^k \\
&= \left(1 - \phi^{-1}\right)x_0^k + \phi^{-1}x_3^k, \\
&= \phi^{-2}x_0^k + \left(1 - \phi^{-2}\right)x_3^k && \text{(Eq. (17))} \\
&= \phi^{-2}x_0^k + \left(1 + \phi^{-1}\right)\left(1 - \phi^{-1}\right)x_3^k \\
&= \phi^{-1}\left(\phi^{-1}x_0^k + \left(1 - \phi^{-1}\right)x_3^k\right) \\
&\qquad + \left(1 - \phi^{-1}\right)x_3^k, \\
&= \phi^{-1}x_1^k + \left(1 - \phi^{-1}\right)x_3^k && \text{(Eq. (15))} \\
&= \phi^{-1}x_0^{k+1} + \left(1 - \phi^{-1}\right)x_3^{k+1}. && \text{(Eq. (14))}
\end{aligned}
$$

This establishes Equation (15) for $k + 1$. Similarly,

$$
\begin{aligned}
x_2^{k+1} &= \phi^{-1}x_2^k + \left(1 - \phi^{-1}\right)x_3^k, \\
&= \phi^{-1}\left(\left(1 - \phi^{-1}\right)x_0^k + \phi^{-1}x_3^k\right) \\
&\qquad + \left(1 - \phi^{-1}\right)x_3^k && \text{(Eq. (16))} \\
&= \phi^{-1}\left(1 - \phi^{-1}\right)x_0^k \\
&\qquad + \left(1 - \phi^{-1} + \phi^{-2}\right)x_3^k \\
&= \left(1 - \phi^{-1}\right)\left(\phi^{-1}x_0^k + \left(1 - \phi^{-1}\right)x_3^k\right) \\
&\qquad + \phi^{-1}x_3^k, \\
&= \left(1 - \phi^{-1}\right)x_1^k + \phi^{-1}x_3^{k+1} && \text{(Eq. (15))} \\
&= \left(1 - \phi^{-1}\right)x_0^{k+1} + \phi^{-1}x_3^{k+1}. && \text{(Eq. (14))}
\end{aligned}
$$

This establishes Equation (16) for $k + 1$. The proof for the remaining case of $f_2^k \geq f_1^k$ is identical due to symmetry. Furthermore, the base case for $k = 0$ automatically holds due to the condition on $(a, b, c)$. Therefore, Equations (15) and (16) hold for all $k \geq 0$.

From Equations (15) and (16), we now have a precise rate of decrease for the interval $\left|x_3^k - x_0^k\right|$. That is, for $f_2^k < f_1^k$,

$$
\begin{aligned}
\left|x_3^k - x_0^k\right| &= \left|x_3^{k-1} - x_1^{k-1}\right| \\
&= \left|x_3^{k-1} - \left(\phi^{-1}x_0^{k-1} + \left(1 - \phi^{-1}\right)x_3^{k-1}\right)\right|
\end{aligned}
$$

$$= \phi^{-1}\left|x_3^{k-1} - x_0^{k-1}\right|$$

and for $f_2^k \geq f_1^k$,

$$\begin{aligned}
\left|x_3^k - x_0^k\right| &= \left|x_2^{k-1} - x_0^{k-1}\right| \\
&= \left|\left(\left(1 - \phi^{-1}\right)x_0^{k-1} + \phi^{-1}x_3^{k-1}\right) - x_0^{k-1}\right| \\
&= \phi^{-1}\left|x_3^{k-1} - x_0^{k-1}\right|.
\end{aligned}$$

Furthermore, This implies, for all $k \geq 1$, the interval $[x_3^k, x_0^k]$ shrinks at a geometrical rate

$$\left|x_3^k - x_0^k\right| = \phi^{-k}\left|x_3^0 - x_0^0\right|.$$

Then

$$\left|x_2^k - x_1^k\right| = \left(2\phi^{-1} - 1\right)\left|x_0^k - x_3^k\right| \leq \epsilon/2$$

can be guaranteed by iterating until the smallest iteration count $k \geq 1$ that satisfies

$$\phi^{-k}\left|x_3^0 - x_0^0\right| \leq \frac{1}{2\phi^{-1} - 1}\frac{\epsilon}{2},$$

$$k = \left\lceil \frac{1}{\log \phi}\log \frac{2\left(2\phi^{-1} - 1\right)|c - a|}{\epsilon}\right\rceil,$$

which yields the execution time complexity statement.

We now prove that the interval $(x_0^{k^*}, x_0^{k^*})$ contains a local minima. For this, we will prove a stronger result that $(x_0^k, x_3^k)$ contains a local minimum for all $k \geq 0$ by induction. Suppose, for some $k \geq 1$,

$$\min\left(f_1^k, f_2^k\right) \leq f(x_0^k) < \infty \tag{18}$$
$$\min\left(f_1^k, f_2^k\right) \leq f(x_3^k) \tag{19}$$

hold. If $f_2^k < f_1^k$, the next set of variables is set as

$$\left(x_0^{k+1}, x_1^{k+1}, x_3^{k+1}\right) = \left(x_1^k, x_2^k, x_3^k\right), \tag{20}$$

which guarantees that the inequalities

$$\min\left(f_1^{k+1}, f_2^{k+1}\right) \leq f_1^{k+1} = f_2^k \leq f(x_0^{k+1}) < \infty$$
$$\min\left(f_1^{k+1}, f_2^{k+1}\right) \leq f_1^{k+1} = f_2^k \leq f(x_3^{k+1})$$

hold. Otherwise, if $f_2^k \geq f_1^k$,

$$\left(x_0^{k+1}, x_2^{k+1}, x_3^{k+1}\right) = \left(x_0^k, x_1^k, x_2^k\right),$$

guarantee

$$\min\left(f_1^{k+1}, f_2^{k+1}\right) \leq f_2^{k+1} = f_1^k \leq f(x_0^{k+1}) < \infty$$
$$\min\left(f_1^{k+1}, f_2^{k+1}\right) \leq f_2^{k+1} = f_1^k \leq f(x_3^{k+1}).$$

The base case $k = 0$ trivially holds by assumption $f(b) < f\left(x_0^0\right) < \infty$, $f(b) < f(x_3^0)$, and the fact that either $x_1^0$ or $x_2^0$ is set as $b$. Therefore, Equations (18) and (19) hold for all $k \geq 0$. Now, Equations (18) and (19) imply that either $(x_0^k, x_1^k, x_3^k)$ or $(x_0^k, x_2^k, x_3^k)$ satisfy the condition in Lemma 1. Therefore, a local minimum is contained in $(x_0^k, x_3^k)$ for all $k \geq 0$. $\qquad\square$

## D.5. `BracketMinimum` (Lemma 3)

We now prove that `BracketMinimum` returns a triplet $(x^-, x_{\mathsf{mid}}, x^+)$ satisfying the condition in Lemma 1. Furthermore, under Assumption 3, we analyze the width of the initial search interval represented by the triplet, $|x^+ - x^-|$. Note that, while `BracketMinimum` is designed to be valid even if $x_0 \geq x^\infty$, accommodating this complicates the analysis. Therefore, in the analysis that will follow, we will assume $x_0 < x^\infty$.

**Lemma 3.** *Suppose* Assumption 3 *holds. Then* `BracketMinimum`$(f, x_0, r, c)$ *for* $x_0 \in (-\infty, x^\infty)$ *returns a triplet* $(x^-, x_{\mathsf{mid}}, x^+)$, *where* $x^- < x_{\mathsf{mid}} < x^+$,

$$f(x_{\mathsf{mid}}) \leq f(x^-) < \infty, \quad f(x_{\mathsf{mid}}) \leq f(x^+), \tag{21}$$

*and*

$$\left|x^+ - x^-\right| \leq r^2\left((r+1)[\overline{x} - x_0]_+ + [x_0 - \underline{x}]_+\right) + 3r^2c$$

*after*

$$\mathrm{O}\left\{(\log r)^{-1}\log_+\left((r[\overline{x} - x_0]_+ + [x_0 - \underline{x}]_+)/c\right)\right\}$$

*objective evaluations.*

*Proof.* `BracketMinimum` has two stages: exponential search to the right (Stage I) and exponential search to the left (Stage II). In the worst case, Stage I must pass $\overline{x}$ moving to the right starting from $x_0$, which takes at most $\mathrm{O}(\bar{k}_r)$ iterations, where

$$\bar{k}_r = \left\lceil (\log r)^{-1}\log_+((\overline{x} - x_0)/c)\right\rceil.$$

Similarly, in the worst case Stage II must pass $\underline{x}$ moving to the left starting from $x_0 + cr^{\bar{k}_r}$, which takes at most $\mathrm{O}(\bar{k}_\ell)$ iterations, where

$$\begin{aligned}
\bar{k}_\ell &= \left\lceil (\log r)^{-1}\log_+\left(\left(x_0 + cr^{\bar{k}_r} - \underline{x}\right)/c\right)\right\rceil \\
&\leq \left\lceil (\log r)^{-1}\log_+\left(x_0 + \left(r[\overline{x} - x_0]_+ - \underline{x}\right)/c\right)\right\rceil.
\end{aligned}$$

Adding these two costs yields the stated result. At the end of Stage I, by inspection, we know that $f(x) \leq f(x^+)$, and that $f(x) < \infty$. Also, Stage II continues until the first increase in objective value, which guarantees that $\infty > f(x^-) \geq f(x_{\mathsf{mid}})$ and $f(x_{\mathsf{mid}}) \leq f(x) \leq f(x^+)$. Finally,

$$\begin{aligned}
\left|x^+ - x^-\right| &\leq \left(x_0 + cr^{\bar{k}_r + 1}\right) - \left(x_0 + cr^{\bar{k}_r} - cr^{\bar{k}_\ell + 1}\right) \\
&\leq rc\left(r^{\bar{k}_r} + r^{\bar{k}_\ell}\right) \\
&\leq r\left(r[\overline{x} - x_0]_+ + rc + r[x_0 + cr^{\bar{k}_r} - \underline{x}]_+ + rc\right) \\
&\leq r^2\left([\overline{x} - x_0]_+ + [x_0 - \underline{x}]_+ + cr^{\bar{k}_r} + 2c\right) \\
&\leq r^2\left([\overline{x} - x_0]_+ + [x_0 - \underline{x}]_+ + r[\overline{x} - x_0]_+ + 3c\right) \\
&= r^2\left((r+1)[\overline{x} - x_0]_+ + [x_0 - \underline{x}]_+\right) + 3r^2c.
\end{aligned}$$
$\qquad\square$

**D.6.** `Minimize` (**Theorem 2**)

We prove that combining `BracketMinimum` and `GoldenSectionSearch`, which we call `Minimize`, results in an optimization algorithm that finds a point $\epsilon$-close to local minimum in $\mathrm{O}\left(\log\left(\Delta/\epsilon\right)\right)$ time.

**Theorem 2.** *Suppose Assumption 3 holds. Then* `Minimize`$(f, x_0, c, r, \epsilon)$ *returns a point that is $\epsilon$-close to a local minimum after $\mathcal{C}_{\mathsf{bm}} + \mathcal{C}_{\mathsf{gss}}$ objective evaluations, where*

$$\mathcal{C}_{\mathsf{bm}} = \mathrm{O}\left\{\frac{1}{\log r}\log_+\left(\Delta\frac{r}{c}\right)\right\}$$

$$\mathcal{C}_{\mathsf{gss}} = \mathrm{O}\left\{\log_+\left(r^3\Delta + r^2c\right)\frac{1}{\epsilon}\right\},$$

*where $\Delta \triangleq [x_0 - \underline{x}]_+ + [\overline{x} - x_0]_+$.*

*Proof.* $\mathcal{C}_{\mathsf{bm}}$ immediately follows from Lemma 3, while $\mathcal{C}_{\mathsf{gss}}$, on the other hand, follows from Lemma 2 as

$$\mathcal{C}_{\mathsf{gss}} = \mathrm{O}\left\{\log_+\left(x^+ - x^-\right)\frac{1}{\epsilon}\right\}$$

$$= \mathrm{O}\left\{\log_+\left(r^3\Delta + r^2c\right)\frac{1}{\epsilon}\right\},$$

where we plugged in the bound on $|x^+ - x^-|$ from Lemma 3. This yields the stated result. Furthermore, since `BracketMinimum` returns a triplet $(x^-, x_{\mathsf{mid}}, x^+)$ that satisfies the requirement of `GoldenSectionSearch` as stated in Lemma 2, the output $x^* \in (-\infty, x^\infty)$ is $\epsilon$-close to a local minimum. $\square$

**Remark 1.** In Theorem 2, the "difficulty" of the problem is represented by $\Delta \geq 0$, where the magnitude of $[x_0 - \underline{x}]_+$ and $[\overline{x} - x_0]_+$ represent the quality of the initialization $x_0$ (how much $x_0$ undershoots or overshoots $\overline{x}$ and $\underline{x}$). Furthermore, we have $\Delta \geq |\overline{x} - \underline{x}|$, where $|\overline{x} - \underline{x}|$ can be thought as the quantitative multimodality of the problem. Therefore, the execution time of `Minimize` becomes longer as the problem becomes more multimodal and the initialization is far from $[\overline{x}, \underline{x}]$.

**Remark 2.** The execution time of `Minimize`$(f, x_0, c, r, \epsilon)$ depends on $r$ and $c$. In general, the best-case performance ($\Delta = 0$) can only become worse as $c$ increases. On the other hand, in the worst-case when $\Delta$ is large, increasing $r$ reduces $\mathcal{C}_{\mathsf{bm}}$, while slowly making $\mathcal{C}_{\mathsf{gss}}$ worse. Therefore, a large $r$ improves the worst-case performance.

**D.7.** `AdaptStepsize` (**Proof of Theorem 1**)

We now present the proof for the theoretical guarantees of Alg. 2 in the main text, Theorem 1. Since most of the heavy lifting in Alg. 2 is done by Alg. 8, Theorem 1 is almost a corollary of Theorem 2. The main difference is that Alg. 2 invokes Alg. 5 at $t = 1$ and operates in log-space. Therefore, the proof incorporates these two modifications into the results of Theorem 2.

**Theorem 1.** *Suppose Assumption 1 holds. Then* `AdaptStepsize`$(\mathcal{L}, t, h_{\mathsf{guess}}, \delta, c, r, \epsilon)$ *returns a step size $h \in (0, h^\infty)$ that is $\epsilon$-close to a local minimum of $\mathcal{L}$ in log-scale after $\mathcal{C}_{\mathsf{feas}} + \mathcal{C}_{\mathsf{bm}} + \mathcal{C}_{\mathsf{gss}}$ objective evaluations, where, defining $\Delta \triangleq \log_+\left(\overline{h}/h_0\right) + \log_+(h_0/\underline{h})$ and $h_0 \triangleq \min(h_{\mathsf{guess}}, h^\infty)$,*

$$\mathcal{C}_{\mathsf{feas}} = \mathrm{O}\left\{\delta^{-1}\log_+\left(h_{\mathsf{guess}}/h^\infty\right)\right\}$$

$$\mathcal{C}_{\mathsf{bm}} = \mathrm{O}\left\{(\log r)^{-1}\log_+\left(\Delta r c^{-1}\right)\right\}$$

$$\mathcal{C}_{\mathsf{gss}} = \mathrm{O}\left\{\log_+\left(\left(r^3\Delta + r^2c\right)\epsilon^{-1}\right)\right\}.$$

*Proof.* Since Assumption 1 implies that the function $\mathcal{L}^{\mathsf{log}}(h)$ satisfies Assumption 3 with

$$f = \mathcal{L}^{\mathsf{log}}(h), \quad \overline{x} = \log\overline{h}, \quad \underline{x} = \log\underline{h}, \quad x^\infty = \log h^\infty.$$

Then the result is a simple application of the lemmas in the previous sections.

First, under Assumption 1, Alg. 5 can find a point $\ell' \in (-\infty, \log h^\infty)$ that guarantees $\mathcal{L}^{\mathsf{log}}(\ell) < \infty$ within

$$\mathcal{C}_{\mathsf{feas}} \leq \mathrm{O}\left(\delta^{-1}\log_+\left(h_{\mathsf{guess}}/h^\infty\right)\right)$$

steps. Furthermore, $\ell' = \log h_{\mathsf{guess}}$ if $h_{\mathsf{guess}} < h^\infty$, and $\ell' < \log h^\infty$ otherwise. Then, Theorem 2 states that Line 6 of Alg. 2 is guaranteed to find a local minimum of $\mathcal{L}$ after $\mathcal{C}_{\mathsf{bm}} + \mathcal{C}_{\mathsf{gss}}$ iterations, while

$$\Delta = [\overline{x} - x_0]_+ + [x_0 - \underline{x}]_+$$

$$= \left[\log\overline{h} - \ell'\right]_+ + \left[\ell' - \log\underline{h}\right]_+$$

$$= \log_+\left(\overline{h}/h_0\right) + \log_+(h_0/\underline{h}).$$

$\square$

# E. Backward Kernels

## E.1. Some backward kernels are not like the others

Here, we will discuss some options for the "backward kernel" used in SMC samplers in the static model setting (Del Moral et al., 2006; Neal, 2001).

**Detailed Balance Formula.** In the literature, the choice

$$L_{t-1}^{\mathsf{dbf}}(x_t, x_{t-1}) \triangleq \frac{\gamma_t(x_t) K_t(x_{t-1}, x_t)}{\gamma_t(x_{t-1})}, \quad (22)$$

which we will refer to as the "detailed balance formula backward kernel," has been the widely used (Dai et al., 2022; Bernton et al., 2019; Heng et al., 2020). Conveniently, Eq. (22) results in a simple expression for the potential

$$G_t(x_{t-1}, x_t) = \frac{\gamma_t(x_{t-1})}{\gamma_{t-1}(x_{t-1})},$$

which does not involve the densities of $K_t$. The origin of this backward kernel is that many $\pi_t$-invariant MCMC kernels used in practice satisfy the detailed balance formula (Robert & Casella, 2004, Def. 6.45) with $\pi_t$,

$$\pi_t(x_{t-1}) K_t(x_{t-1}, x_t) = \pi_t(x_t) K_t(x_t, x_{t-1}),$$

which, given $L_{t-1}^{\mathsf{dbf}}(x_{t-1}, x_t) = K_t(x_{t-1}, x_t)$, yields Eq. (22) after re-aranging.

**The detailed balance formula backward kernel is biased.** Now, let's focus on the fact that $K_t = L_{t-1}^{\mathsf{dbf}}$ only holds under the detailed balance condition. Said differently, $L_{t-1}^{\mathsf{dbf}}$ is a properly *normalized* kernel only when $K_t$ satisfies detailed balance. This implies that, for non-reversible kernels like LMC, using the detailed balance formula kernel with $h > 0$ may result in biased normalized constant estimates. This bias can be substantial, as we will see in App. E.2. Fortunately, this bias does diminish as $h_t \to 0$ and $T \to \infty$ since the continuous Langevin dynamics is reversible under stationarity (Heng et al., 2020). However, the need for smaller step sizes means that a larger number of SMC steps $T$ has to be taken for the Markov process to converge.

**Forward Kernel.** The properly normalized analog of $L_{t-1}^{\mathsf{dbf}}$ at time $t \geq 1$ is the "forward" kernel

$$L_{t-1}^{\mathsf{fwd}}(x_t, x_{t-1}) \triangleq K_t(x_t, x_{t-1}).$$

This has been used, for example, by Thin et al. (2021). Recall that the "optimal" $L_{t-1}$ is a kernel that transports the particles following $P_t$ to follow $P_{t-1}$. The fact that we are using $K_t$ to do this implies that we are assuming $P_t \approx P_{t-1}$, which is only true if $T$ is large. We propose a different option, which should work even when $T$ is moderate or small.

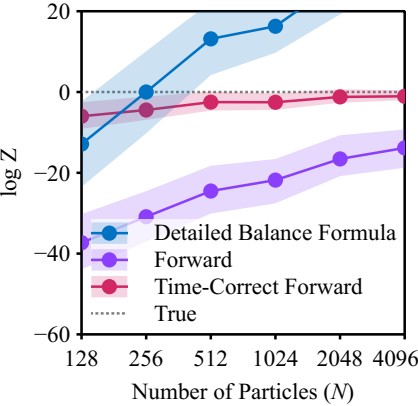

*Figure 7.* **Comparison of backward kernels for SMC-LMC.** The solid lines are the median, while the colored bands mark the 80% empirical quantile over 256 replications.

**Time-Correct Forward Kernel.** In § 4.1, we used the forward kernel at time $t - 1$,

$$L_{t-1}^{\mathsf{tc\text{-}fwd}}(x_t, x_{t-1}) \triangleq K_{t-1}(x_t, x_{t-1}),$$

which we will refer to as the time-correct forward kernel. Unlike $L_{t-1}^{\mathsf{fwd}}$, the stationary distribution of this transport map is properly $\pi_{t-1}$. Informally, the reasoning is that

$$\frac{(\pi_t \otimes K_{t-1})(x_t, x_{t-1})}{(\pi_{t-1} \otimes K_t)(x_{t-1}, x_t)} \approx \frac{(\pi_t \otimes \pi_{t-1})(x_t, x_{t-1})}{(\pi_{t-1} \otimes \pi_t)(x_{t-1}, x_t)} = 1.$$

Therefore, this should result in lower variance.

## E.2. Empirical Evaluation

**Setup.** We compare the three backward kernels on a toy problem with $d = 10$ dimensional Gaussians: $\pi = \mathcal{N}(30 \cdot 1_d, I_d)$, $q_0 = \mathcal{N}(0_d, I_d)$. Since the scale of the target distribution is constant under geometric annealing, a fixed step size $h = h_t = 0.5$ should work well. We use a linear schedule with $T = 64$.

**Results.** The results are shown in Fig. 7. The backward kernel from the detailed balance formula severely overestimates the normalizing constant due to bias, while the forward kernel exhibits significantly higher variance than the time-correct forward kernel.

## E.3. Conclusions

We have demonstrated that caution must be taken when using the popular backward kernel based on the detail-balance formula. Instead, we have proposed the "time-correct forward kernel," which is not only valid but also results in substantially lower variance. Unfortunately, the time-correct forward kernel is only available for MCMC kernels that have a tractable density, which may not be the case; for instance, the KLMC kernel used in § 4.2 does not have this option. However, whenever it is available, it should be preferred.

# F. Additional Experimental Results

## F.1. Comparison Against Fixed Step Sizes

### F.1.1. SMC-LMC

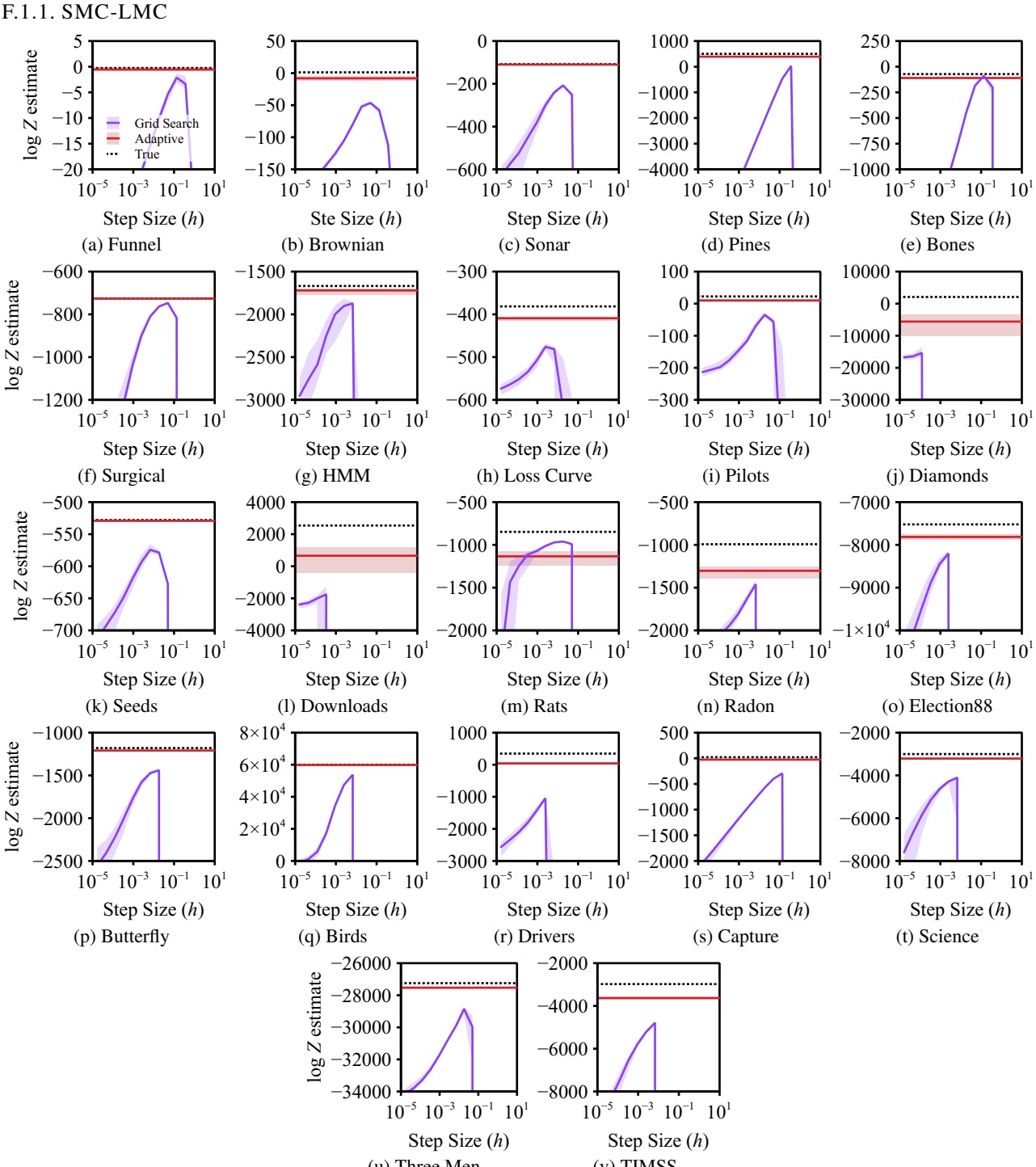

*Figure 8.* **SMC-LMC with adaptive tuning v.s. fixed step sizes.** The solid lines are the median estimate of $\log Z$, while the colored regions are the $80\%$ empirical quantiles computed over $32$ replications.

## F.1.2. SMC-KLMC

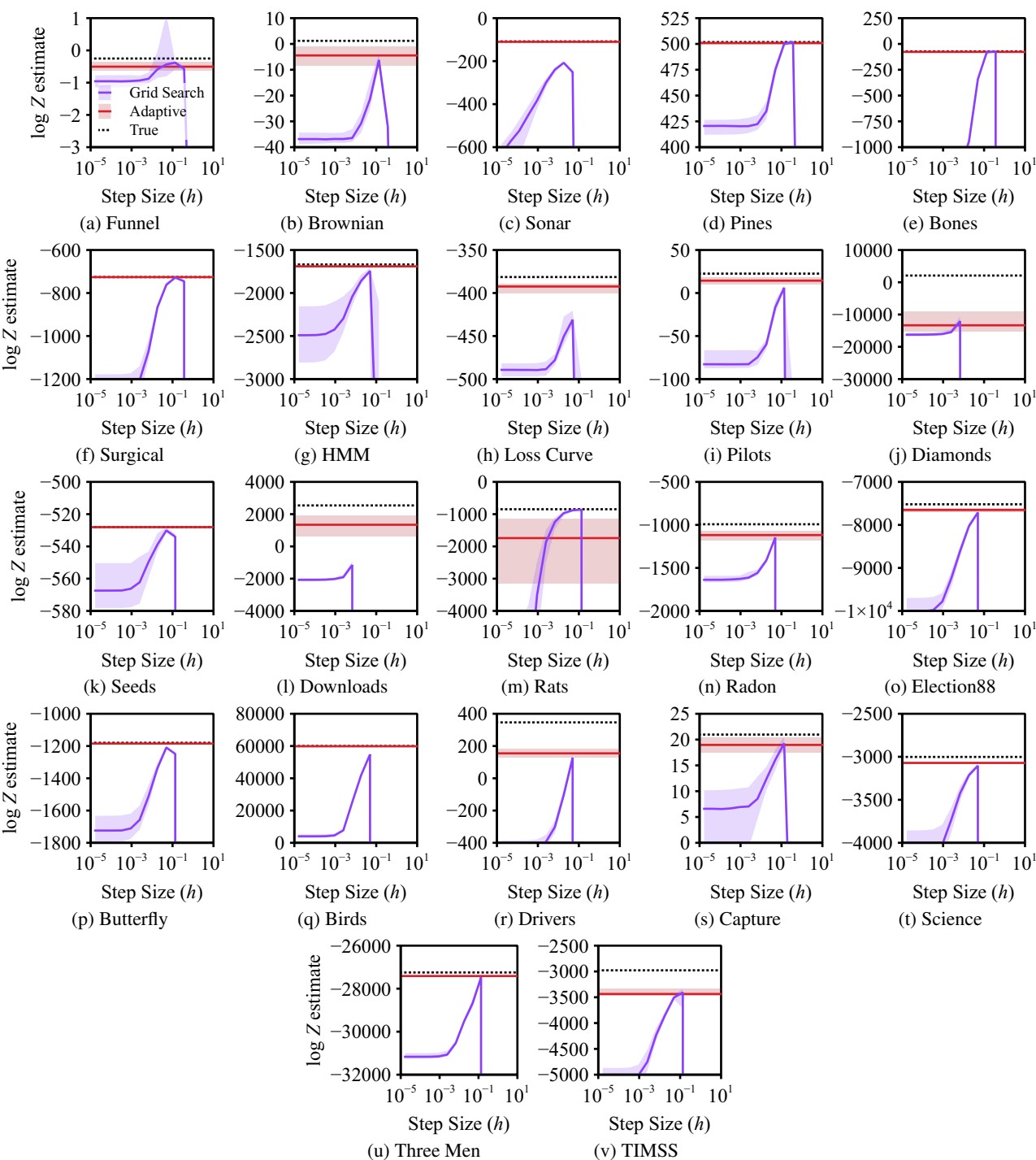

*Figure 9.* **SMC-KLMC with adaptive tuning v.s. fixed step sizes and refreshment rates.** For SMC-KLMC with fixed parameters $h, \rho$, we show the result of the best-performing refreshment rate. The solid lines are the median estimate of $\log Z$, while the colored regions are the 80% empirical quantiles computed over 32 replications.

### F.2. Comparison Against End-to-End Optimization

For all results, the "cost" is calculated as the cumulative number of gradients and Hessian evaluations used by each method. Also, end-to-end optimization with variational tuning of the reference $q$ is referred as "End-to-End + VI." (End-to-end optimization methods require Hessians.) For all figures, the error bars/bands are $80\%$ empirical quantiles computed from 32 replications, while $\gamma$ is the Adam step size used for end-to-end optimization.

### F.2.1. SMC-LMC

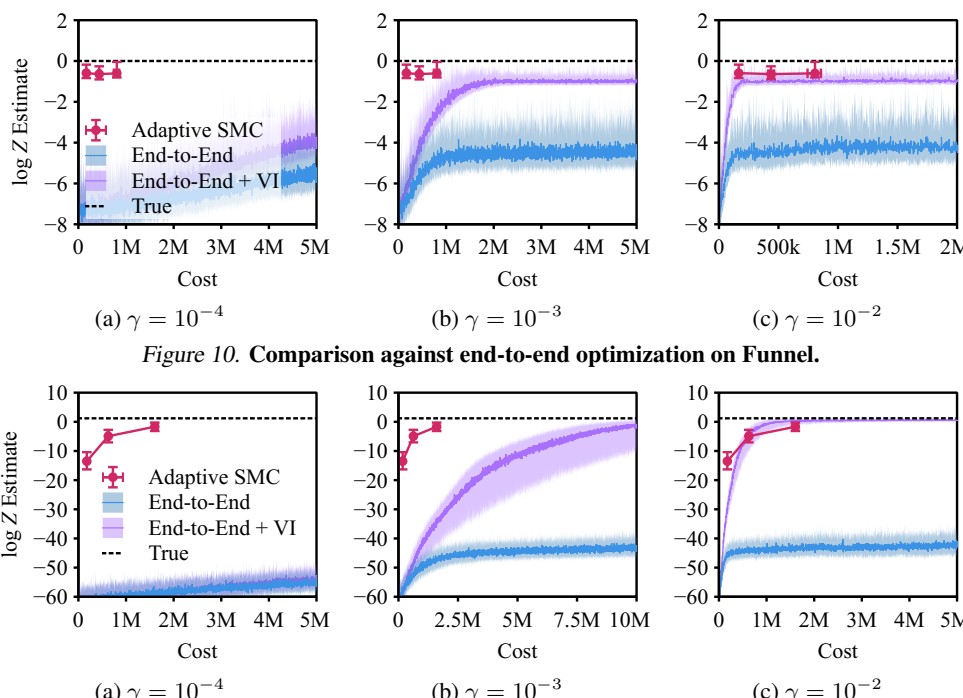

(a) $\gamma = 10^{-4}$      (b) $\gamma = 10^{-3}$      (c) $\gamma = 10^{-2}$

*Figure 10.* **Comparison against end-to-end optimization on Funnel.**

(a) $\gamma = 10^{-4}$      (b) $\gamma = 10^{-3}$      (c) $\gamma = 10^{-2}$

*Figure 11.* **Comparison against end-to-end optimization on Brownian.** The result for variational reference tuning (End-to-End + VI) for $\gamma = 10^{-2}$ is omitted as most runs diverged.

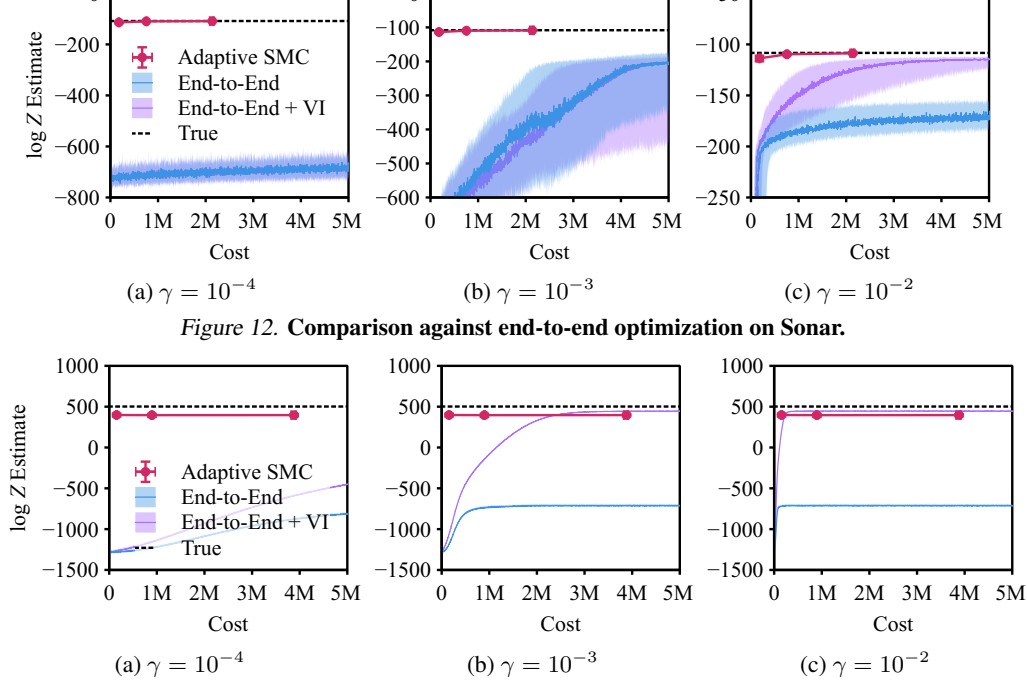

(a) $\gamma = 10^{-4}$      (b) $\gamma = 10^{-3}$      (c) $\gamma = 10^{-2}$

*Figure 12.* **Comparison against end-to-end optimization on Sonar.**

(a) $\gamma = 10^{-4}$      (b) $\gamma = 10^{-3}$      (c) $\gamma = 10^{-2}$

*Figure 13.* **Comparison against end-to-end optimization on Pines.**

F.2.2. SMC-KLMC

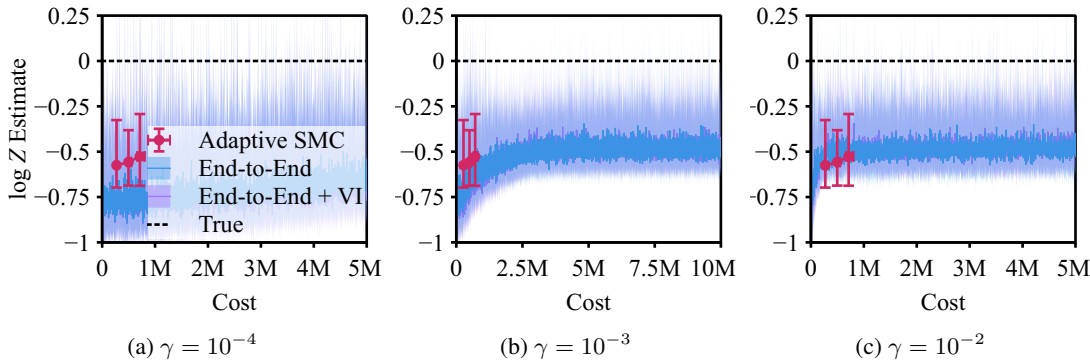

(a) $\gamma = 10^{-4}$  (b) $\gamma = 10^{-3}$  (c) $\gamma = 10^{-2}$

*Figure 14.* **Comparison against end-to-end optimization on Funnel.**

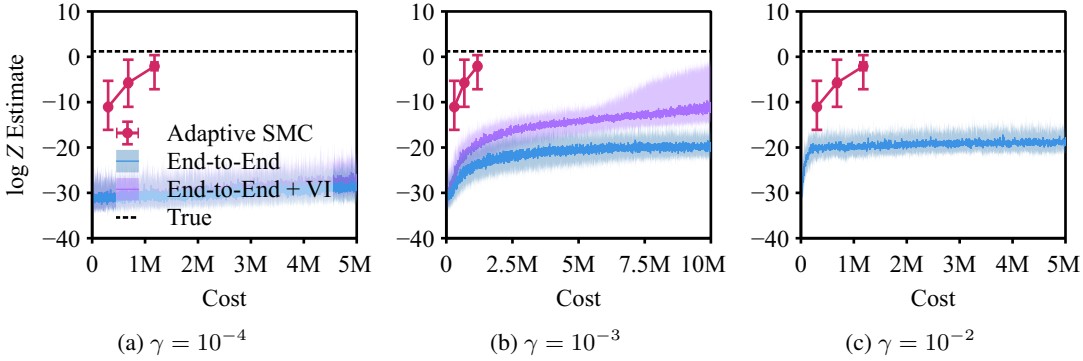

(a) $\gamma = 10^{-4}$  (b) $\gamma = 10^{-3}$  (c) $\gamma = 10^{-2}$

*Figure 15.* **Comparison against end-to-end optimization on Brownian.**

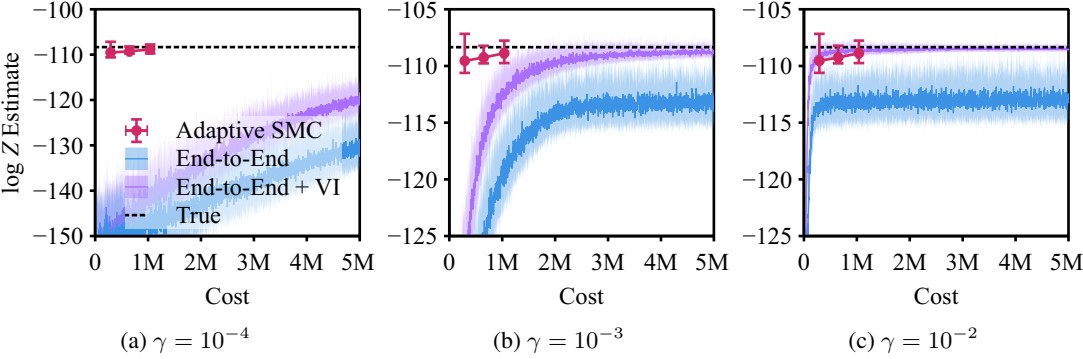

(a) $\gamma = 10^{-4}$  (b) $\gamma = 10^{-3}$  (c) $\gamma = 10^{-2}$

*Figure 16.* **Comparison against end-to-end optimization on Sonar.**

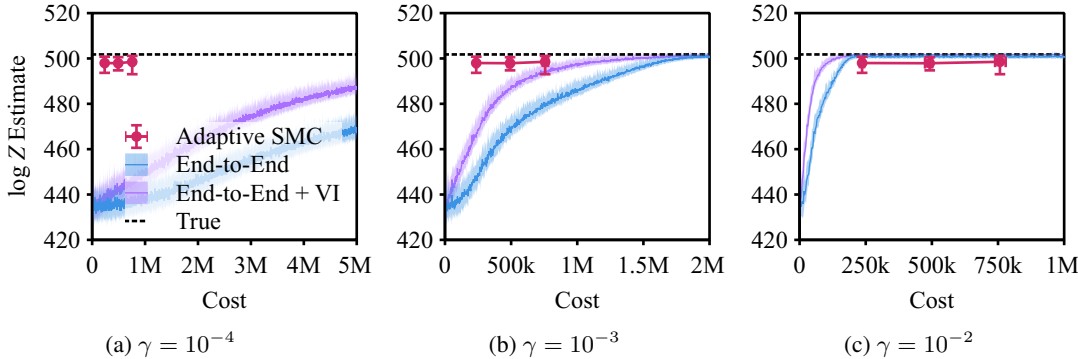

(a) $\gamma = 10^{-4}$  (b) $\gamma = 10^{-3}$  (c) $\gamma = 10^{-2}$

*Figure 17.* **Comparison against end-to-end optimization on Pines.**

## F.3. Adaptation Cost

In this section, we will visualize the cost of adaptation of our algorithm. In particular, we show the number of objective evaluations used at each SMC iteration during adaptation. Recall that the cost of evaluating our objective is in the order of $\mathcal{O}(B)$ unnormalized log-density evaluations ($\log \gamma$) and its gradients ($\nabla \log \gamma$), where $B$ is the number of subsampled particles (§ 3.2). Therefore, the cost of $N/B$ adaptation objective evaluations at every SMC step roughly amounts to the cost of a single vanilla SMC run with $N$ particles. That is, for $N = 1024$ and $B = 128$, the cost of running our adaptive SMC sampler is comparable to two to three times that of a vanilla SMC sampler. The exact number of objective evaluations spent at each SMC iteration is shown in the figures that will follow. All experiments used $N = 1024$, $B = 128$, and $T = 64$

### F.3.1. SMC-LMC

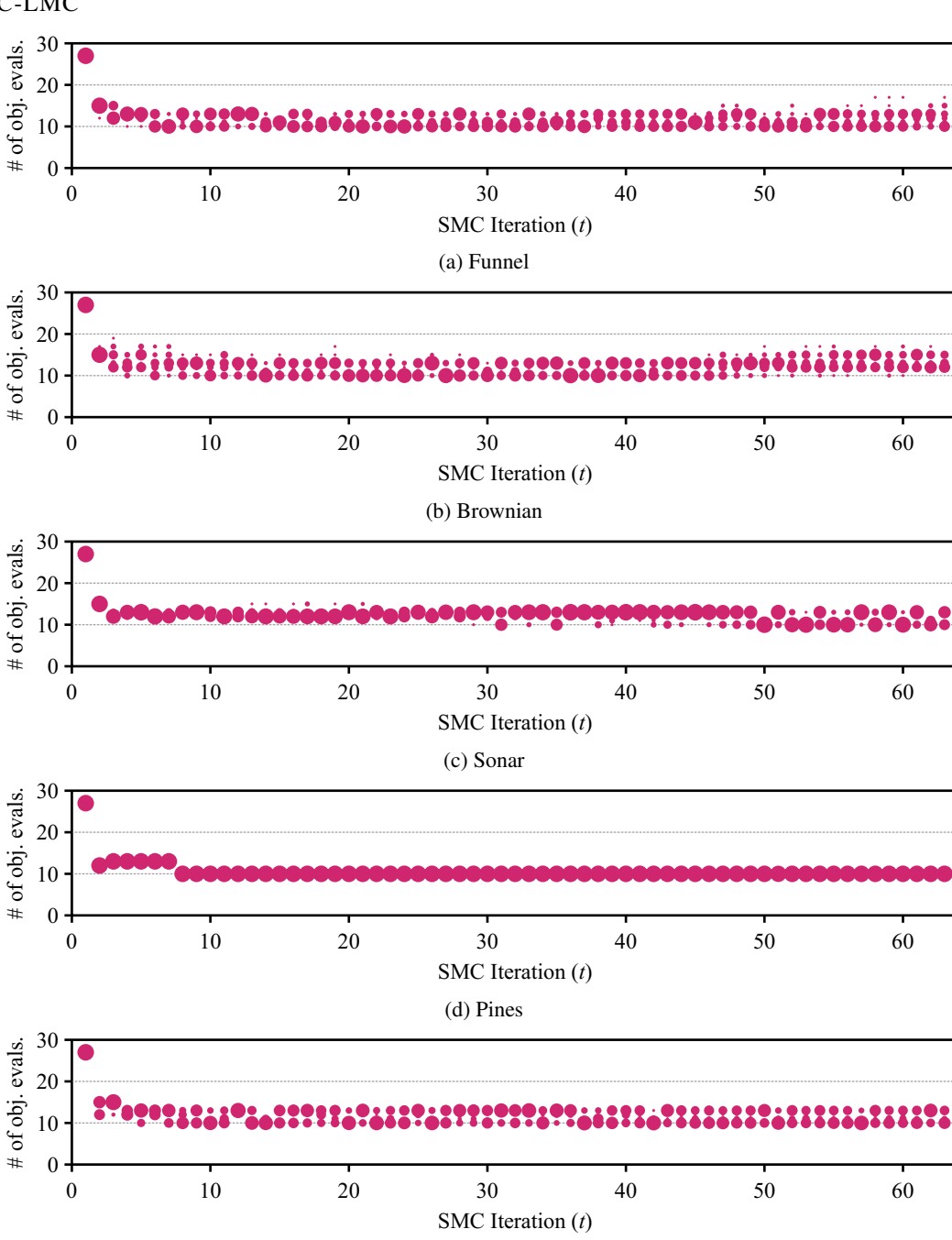

(a) Funnel

(b) Brownian

(c) Sonar

(d) Pines

(e) Bones

*Figure 18.* **Number of objective evaluations spent during adaptation at each SMC iteration.** The size of the markers represents the proportion of runs that spent each respective number of evaluations among 32 independent runs.

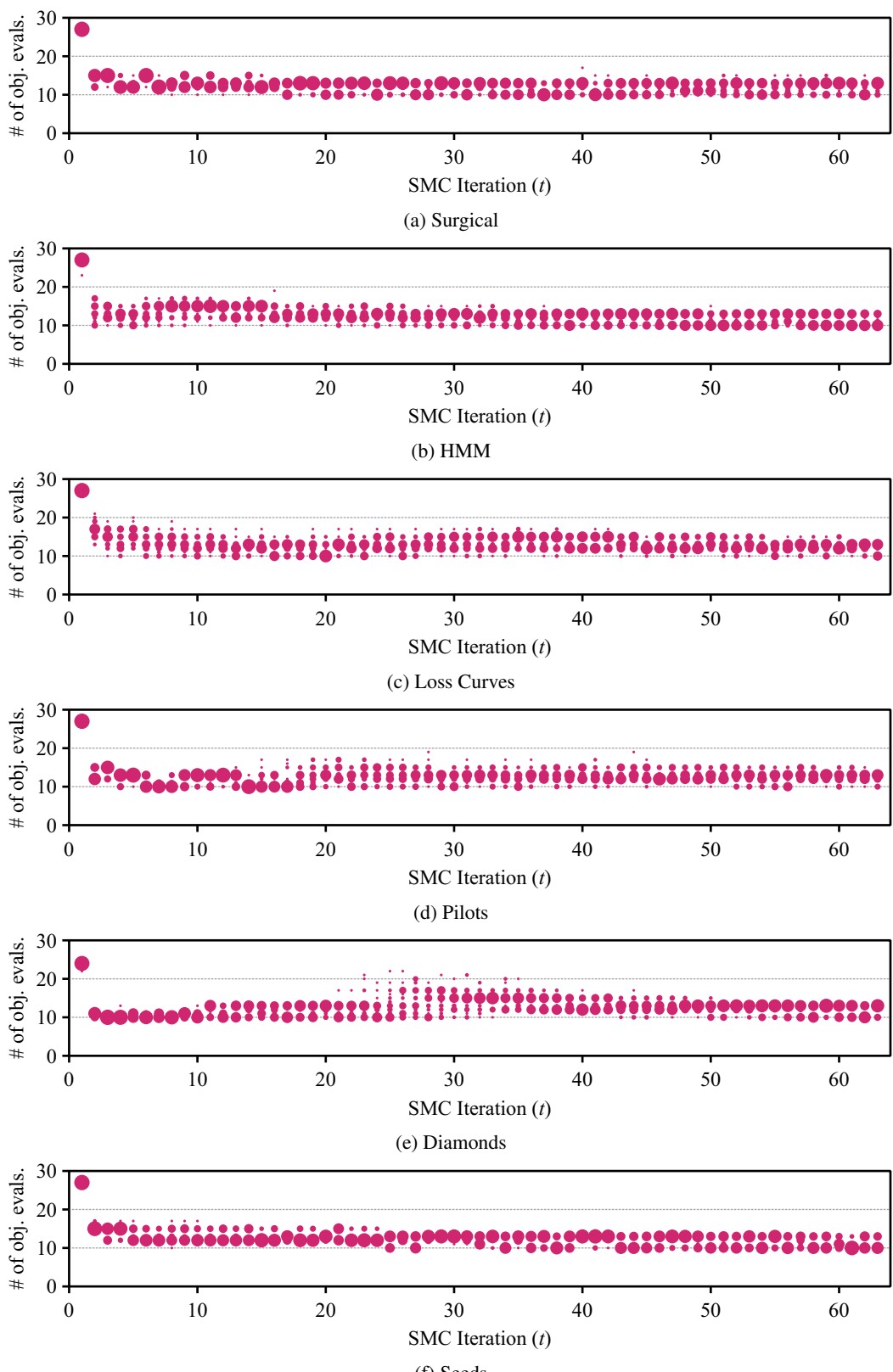

*Figure 19.* **Number of objective evaluations spent during adaptation at each SMC iteration.** The size of the markers represents the proportion of runs that spent each respective number of evaluations among 32 independent runs.

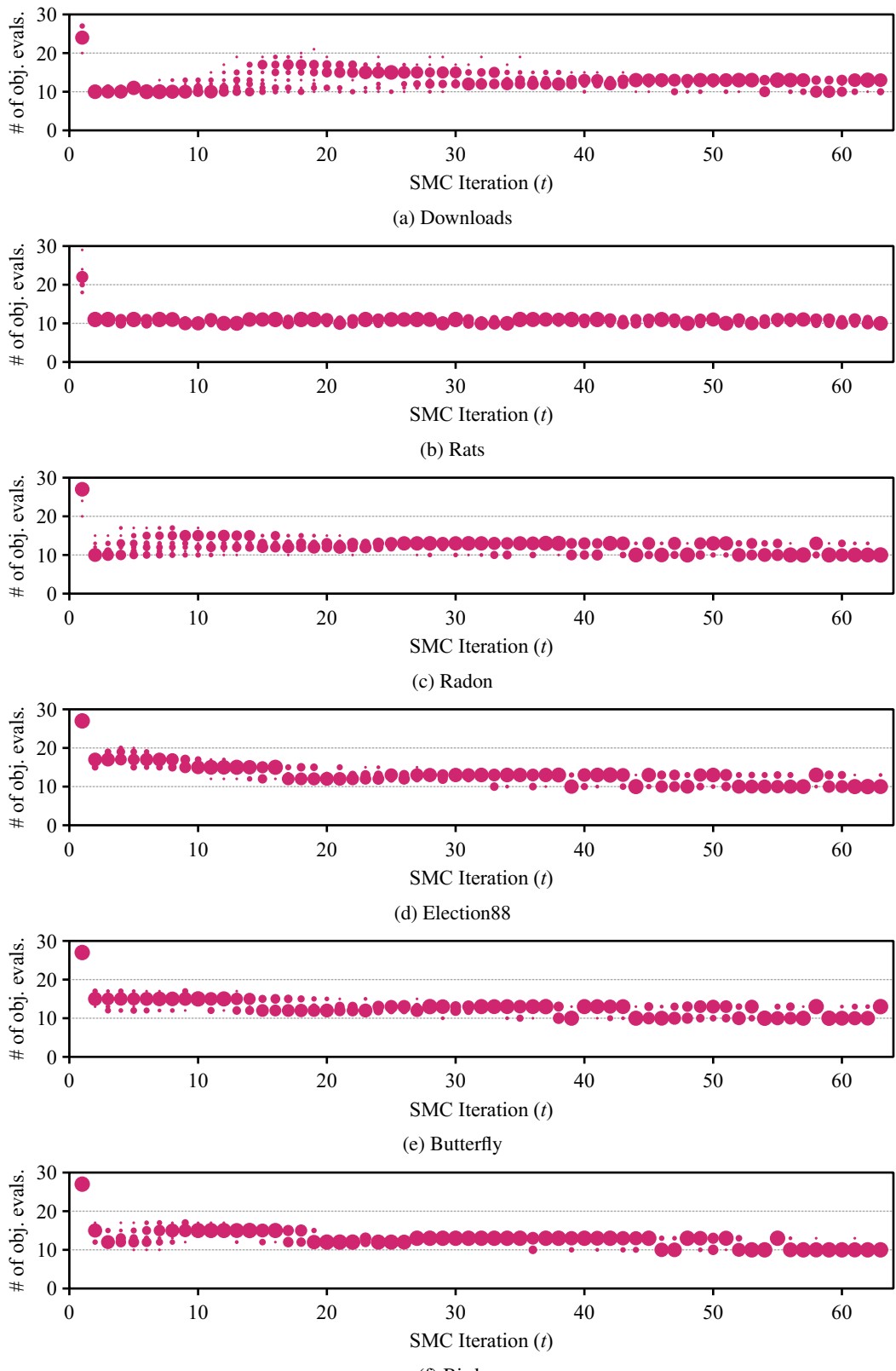

*Figure 20.* **Number of objective evaluations spent during adaptation at each SMC iteration (continued).** The size of the markers represents the proportion of runs that spent each respective number of evaluations among 32 independent runs.

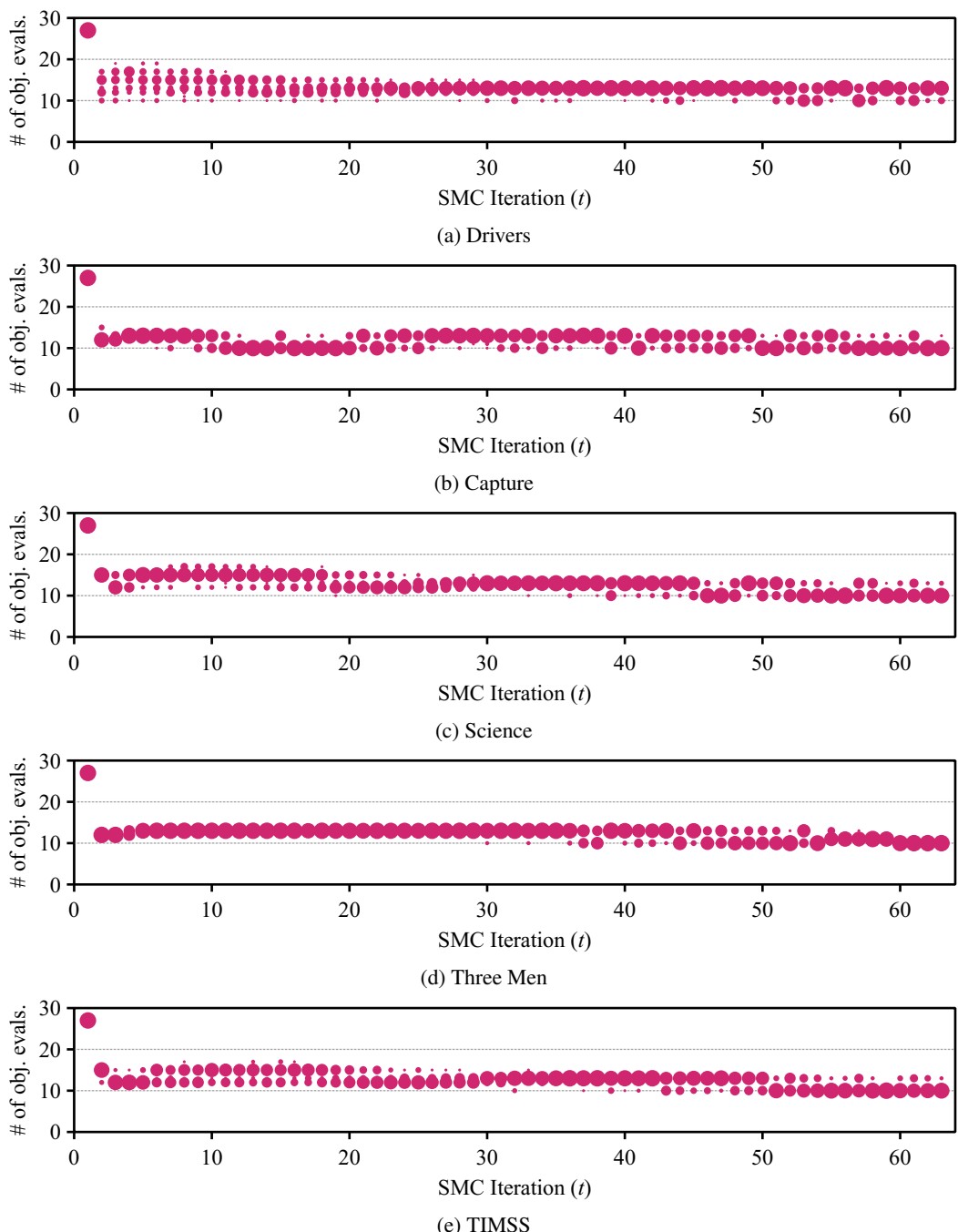

*Figure 21.* **Number of objective evaluations spent during adaptation at each SMC iteration (continued).** The size of the markers represents the proportion of runs that spent each respective number of evaluations among 32 independent runs.

F.3.2. SMC-KLMC

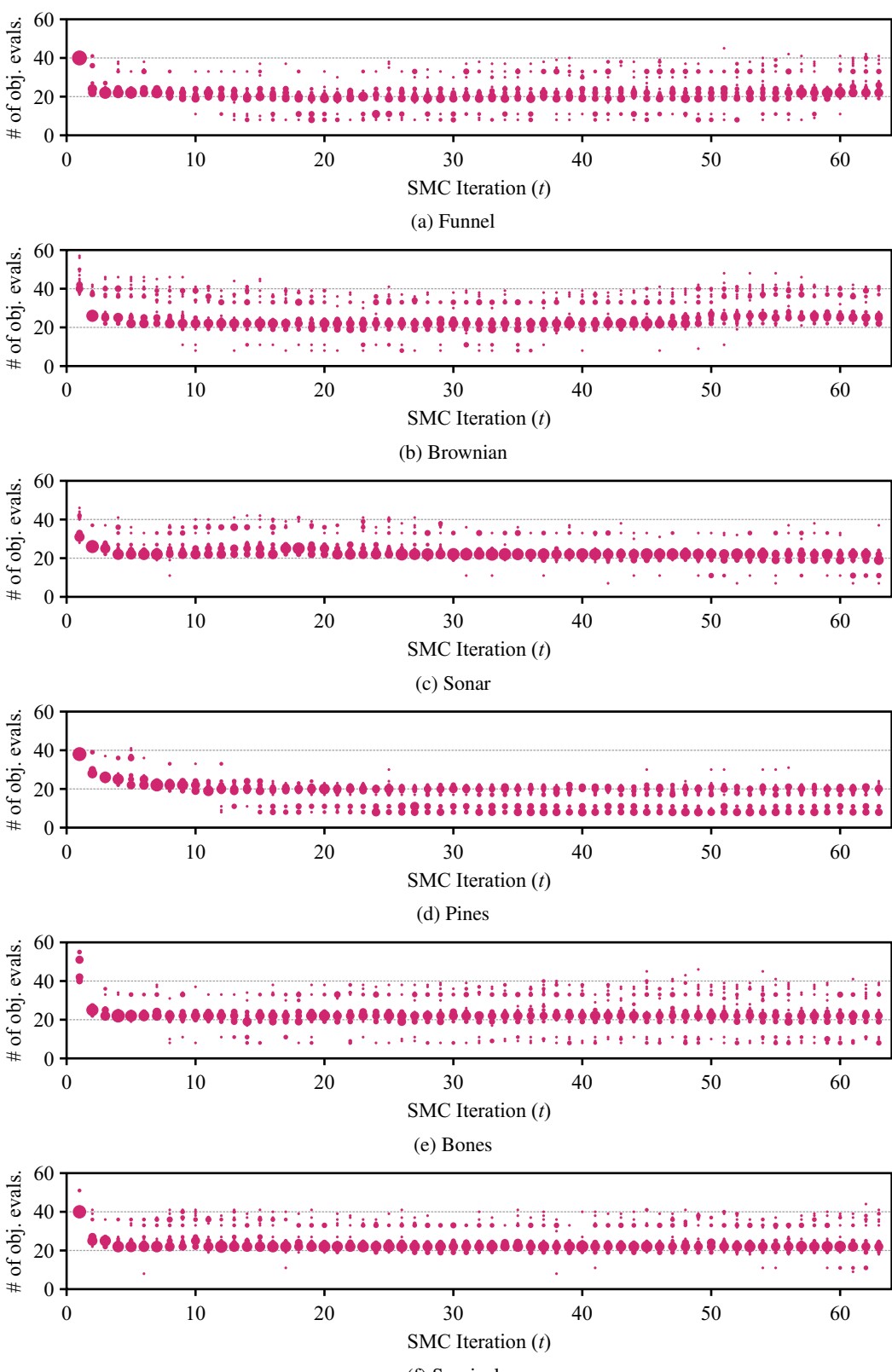

*Figure 22.* **Number of objective evaluations spent during adaptation at each SMC iteration.** The size of the markers represents the proportion of runs that spent each respective number of evaluations among 32 independent runs.

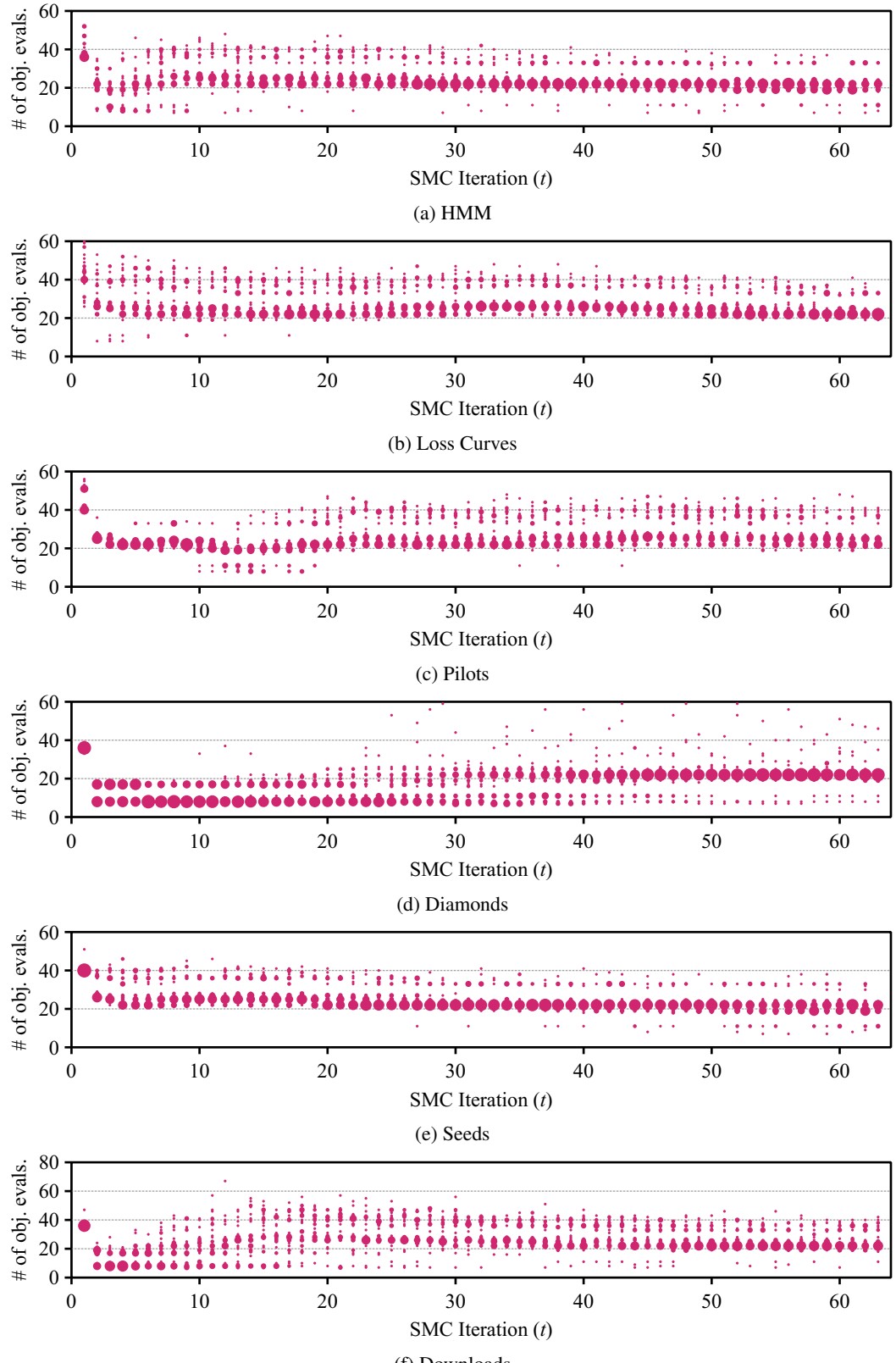

*Figure 23.* **Number of objective evaluations spent during adaptation at each SMC iteration (continued).** The size of the markers represents the proportion of runs that spent each respective number of evaluations among 32 independent runs.

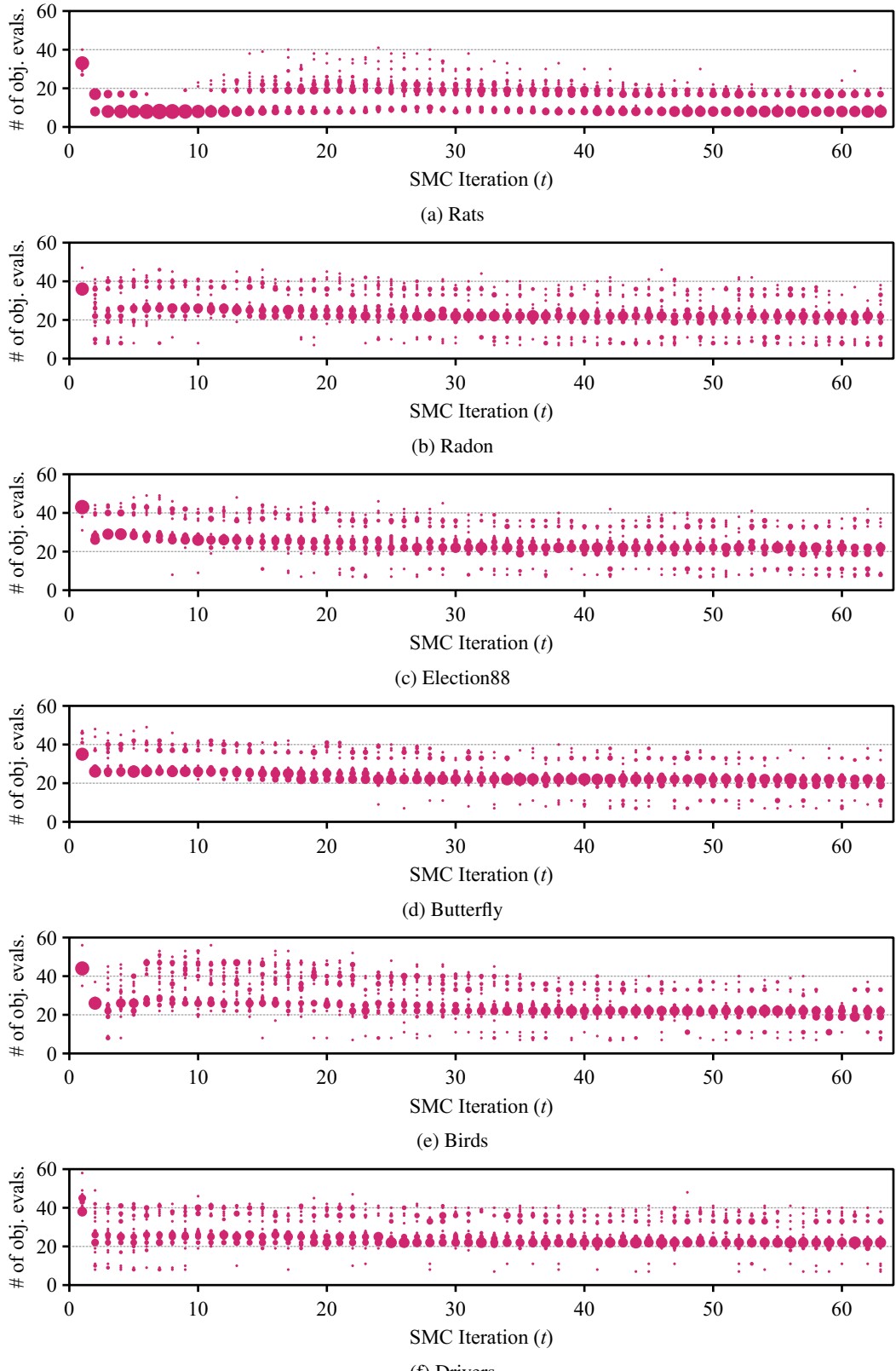

*Figure 24.* **Number of objective evaluations spent during adaptation at each SMC iteration (continued).** The size of the markers represents the proportion of runs that spent each respective number of evaluations among 32 independent runs.

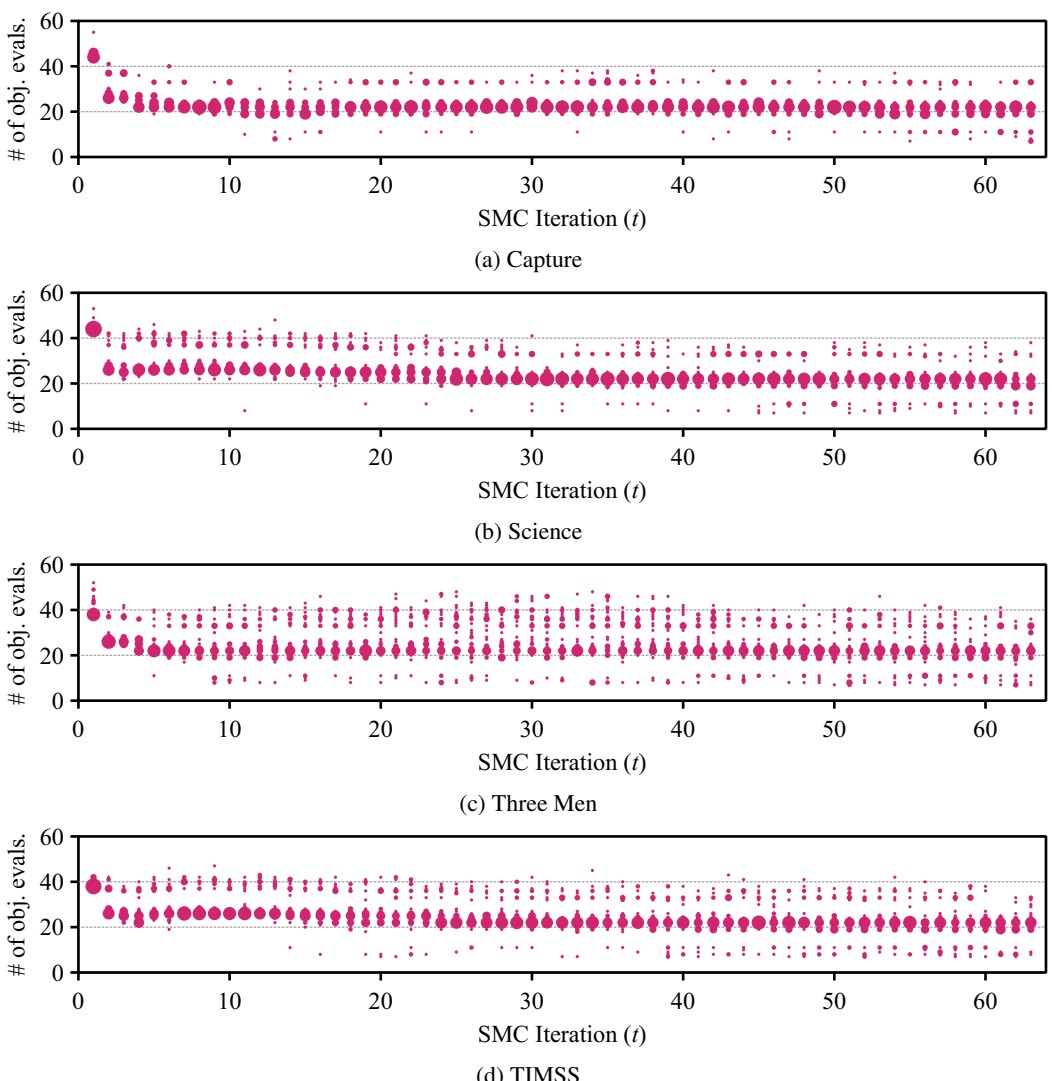

(a) Capture

(b) Science

(c) Three Men

(d) TIMSS

*Figure 25.* **Number of objective evaluations spent during adaptation at each SMC iteration (continued).** The size of the markers represents the proportion of runs that spent each respective number of evaluations among 32 independent runs.

## F.4. Adaptation Results from the Adaptive SMC Samplers

Finally, we will present additional results generated from our adaptive SMC samplers, including the adapted temperature schedule, step size schedule, and normalizing constant estimates. The computational budgets are set as $T_1 = 64$ and $N = 1024$ with $B = 256$.

### F.4.1. SMC-LMC

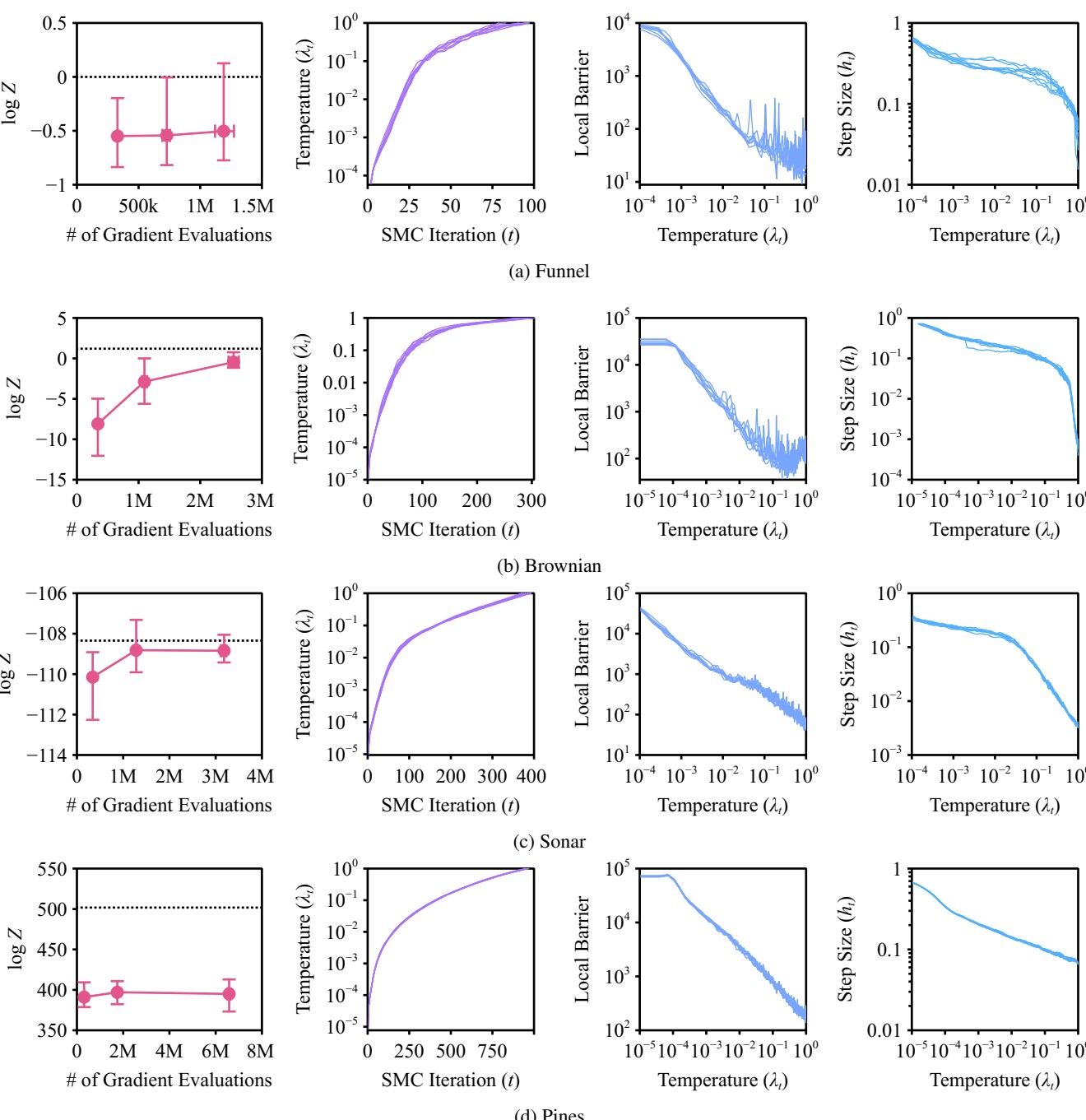

*Figure 26.* **Normalizing constant estimate, temperature schedule, local communication barrier, and step size schedules obtained by running SMC-LMC.** The dotted line is the ground truth value obtained from a large budget run. For the normalizing constant estimate, the confidence intervals in the vertical and horizontal directions are the 80% quantiles obtained from 32 replications. The temperature schedule, local communication barriers, and the step sizes from a subset of 8 runs are shown.

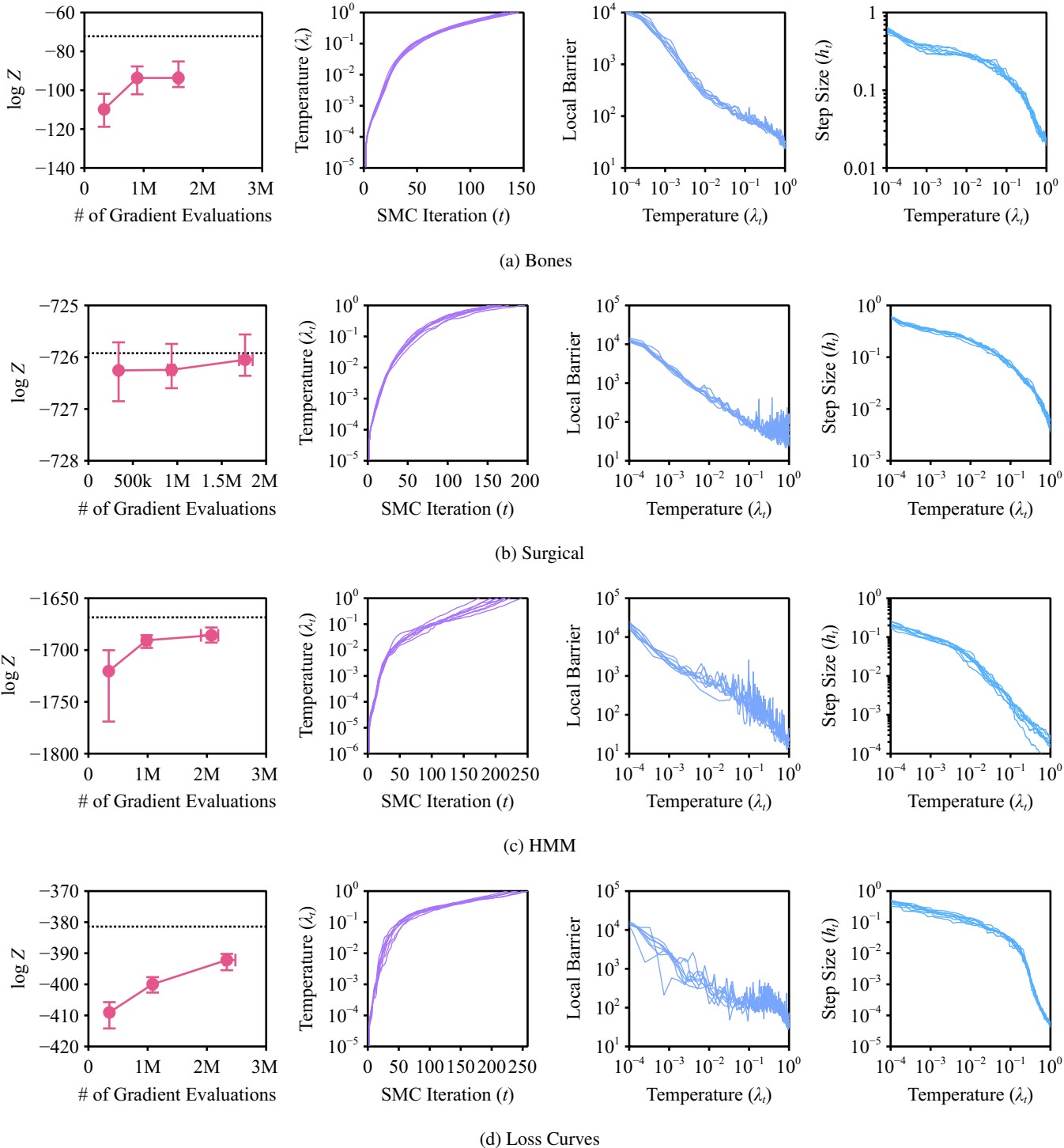

(a) Bones

(b) Surgical

(c) HMM

(d) Loss Curves

*Figure 27.* **Normalizing constant estimate, temperature schedule, local communication barrier, and step size schedules obtained by running SMC-LMC (continued).** The dotted line is the ground truth value obtained from a large budget run. For the normalizing constant estimate, the confidence intervals in the vertical and horizontal directions are the 80% quantiles obtained from 32 replications. The temperature schedule, local communication barriers, and the step sizes from a subset of 8 runs are shown.

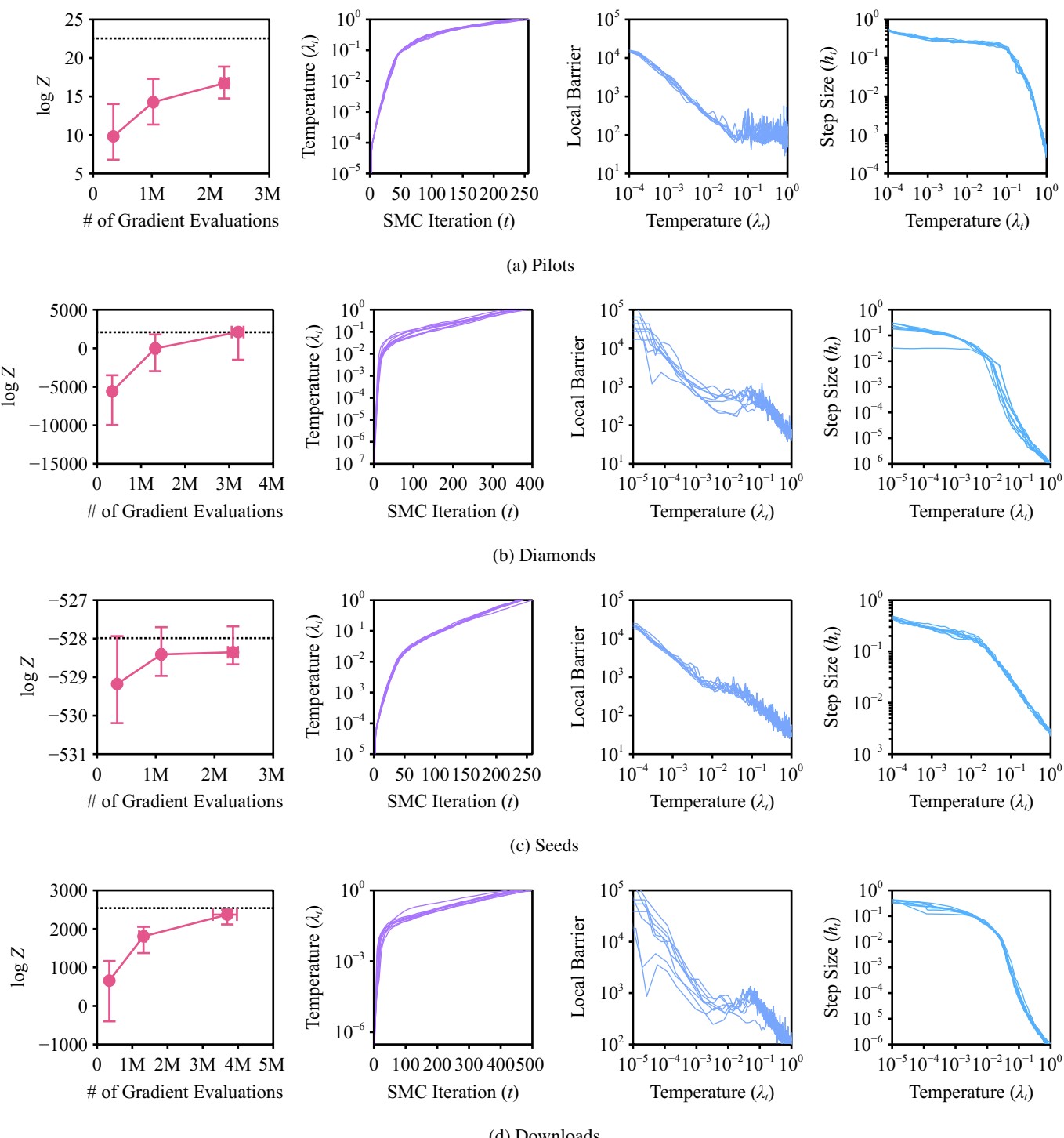

*Figure 28.* **Normalizing constant estimate, temperature schedule, local communication barrier, and step size schedules obtained by running SMC-LMC (continued).** The dotted line is the ground truth value obtained from a large budget run. For the normalizing constant estimate, the confidence intervals in the vertical and horizontal directions are the $80\%$ quantiles obtained from 32 replications. The temperature schedule, local communication barriers, and the step sizes from a subset of 8 runs are shown.

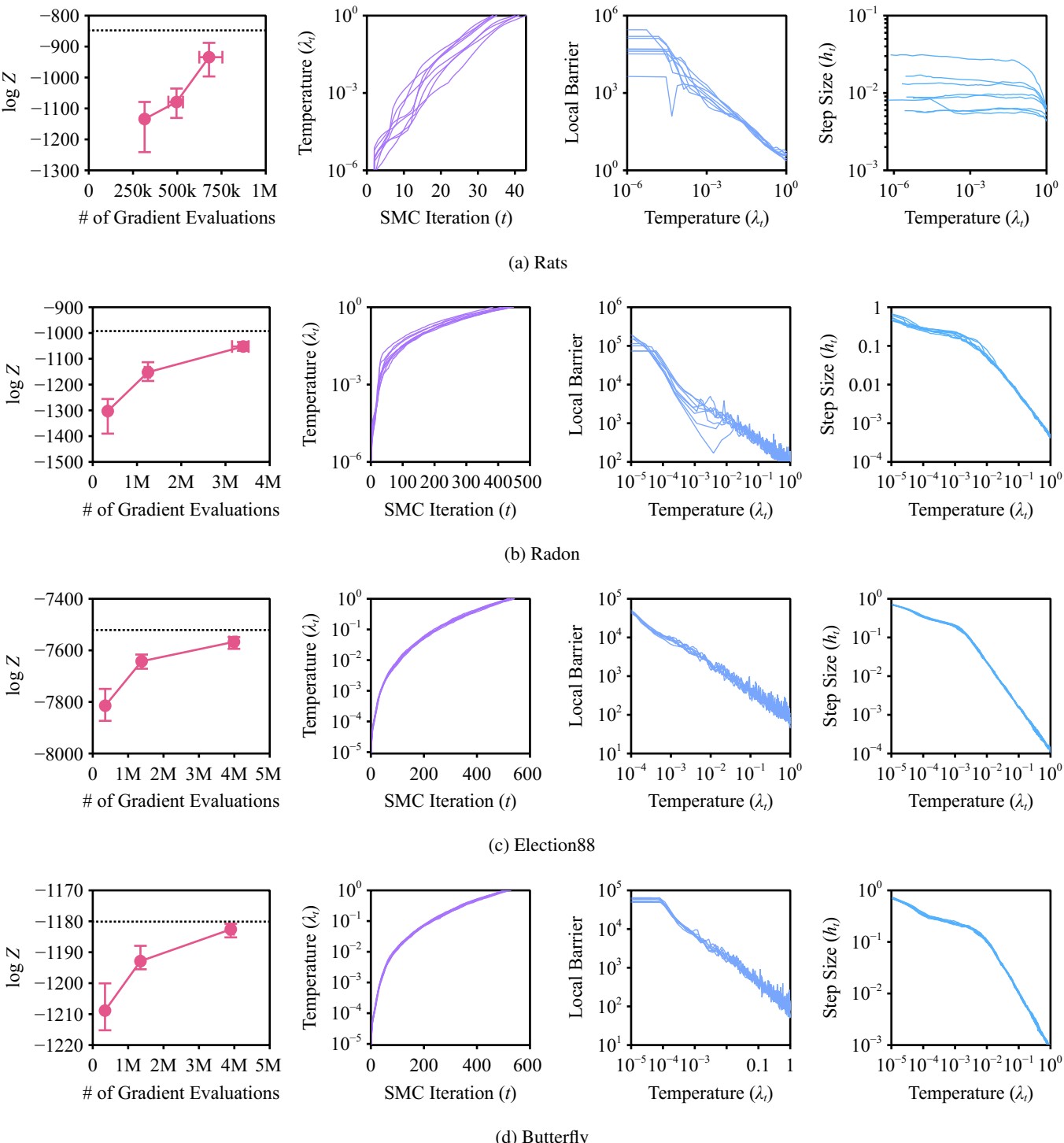

(a) Rats

(b) Radon

(c) Election88

(d) Butterfly

*Figure 29.* **Normalizing constant estimate, temperature schedule, local communication barrier, and step size schedules obtained by running SMC-LMC (continued).** The dotted line is the ground truth value obtained from a large budget run. For the normalizing constant estimate, the confidence intervals in the vertical and horizontal directions are the 80% quantiles obtained from 32 replications. The temperature schedule, local communication barriers, and the step sizes from a subset of 8 runs are shown.

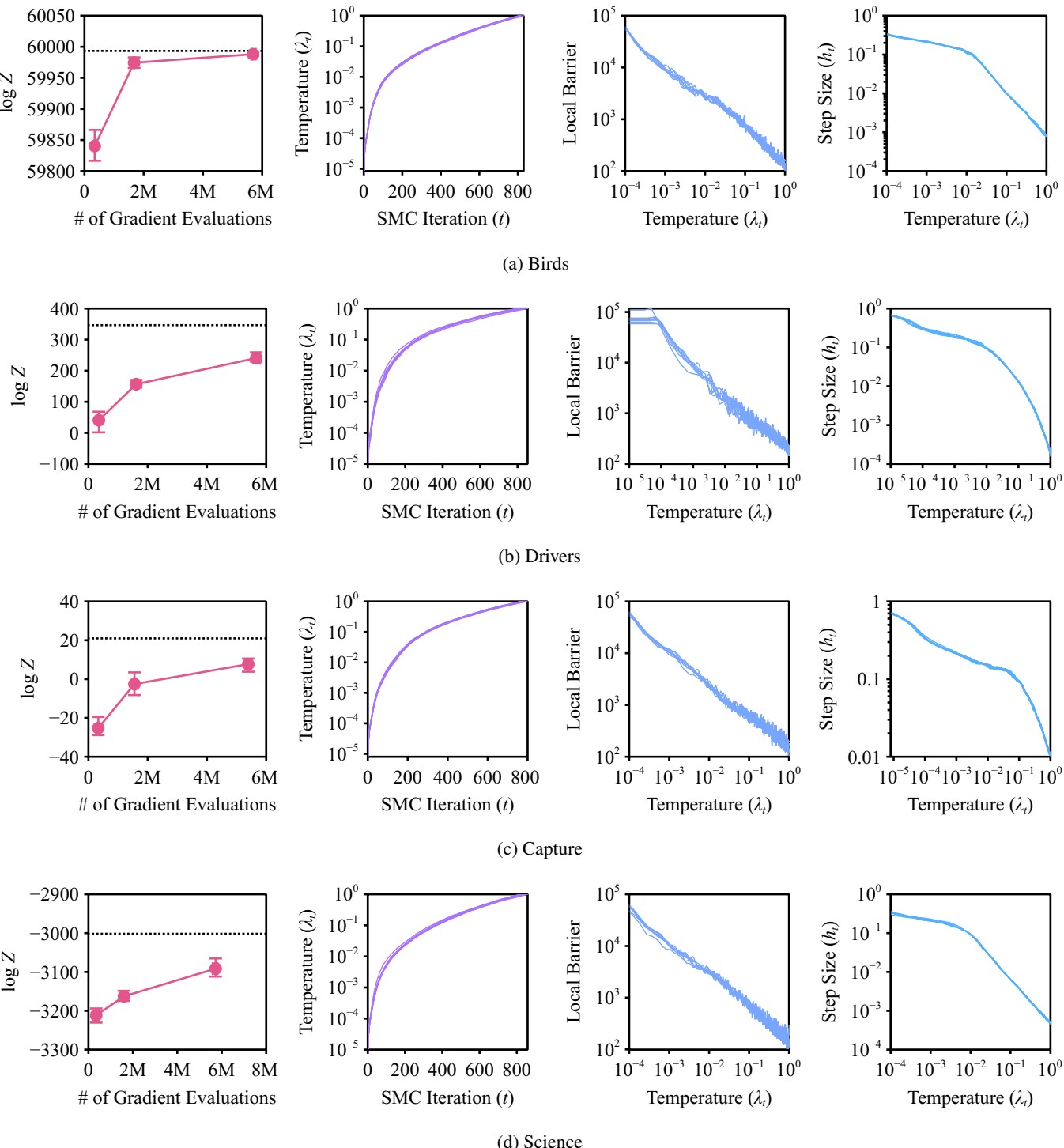

(a) Birds

(b) Drivers

(c) Capture

(d) Science

*Figure 30.* **Normalizing constant estimate, temperature schedule, local communication barrier, and stepsize schedules obtained by running SMC-LMC (continued).** The dotted line is the ground truth value obtained from a large budget run. For the normalizing constant estimate, the confidence intervals in the vertical and horizontal directions are the 80% quantiles obtained from 32 replications. The temperature schedule, local communication barriers, and the step sizes from a subset of 8 runs are shown.

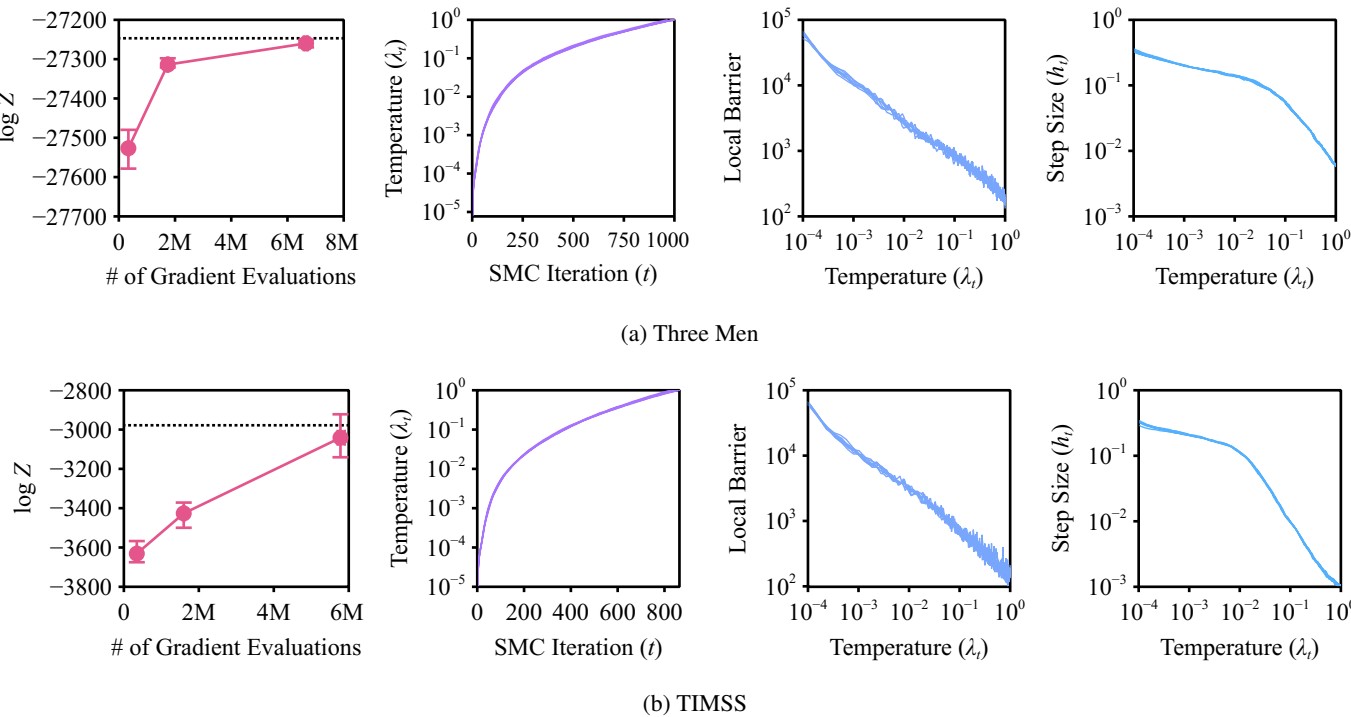

(a) Three Men

(b) TIMSS

*Figure 31.* **Normalizing constant estimate, temperature schedule, local communication barrier, and step size schedules obtained by running SMC-LMC (continued).** The dotted line is the ground truth value obtained from a large budget run. For the normalizing constant estimate, the confidence intervals in the vertical and horizontal directions are the 80% quantiles obtained from 32 replications. The temperature schedule, local communication barriers, and the step sizes from a subset of 8 runs are shown.

## F.4.2. SMC-KLMC

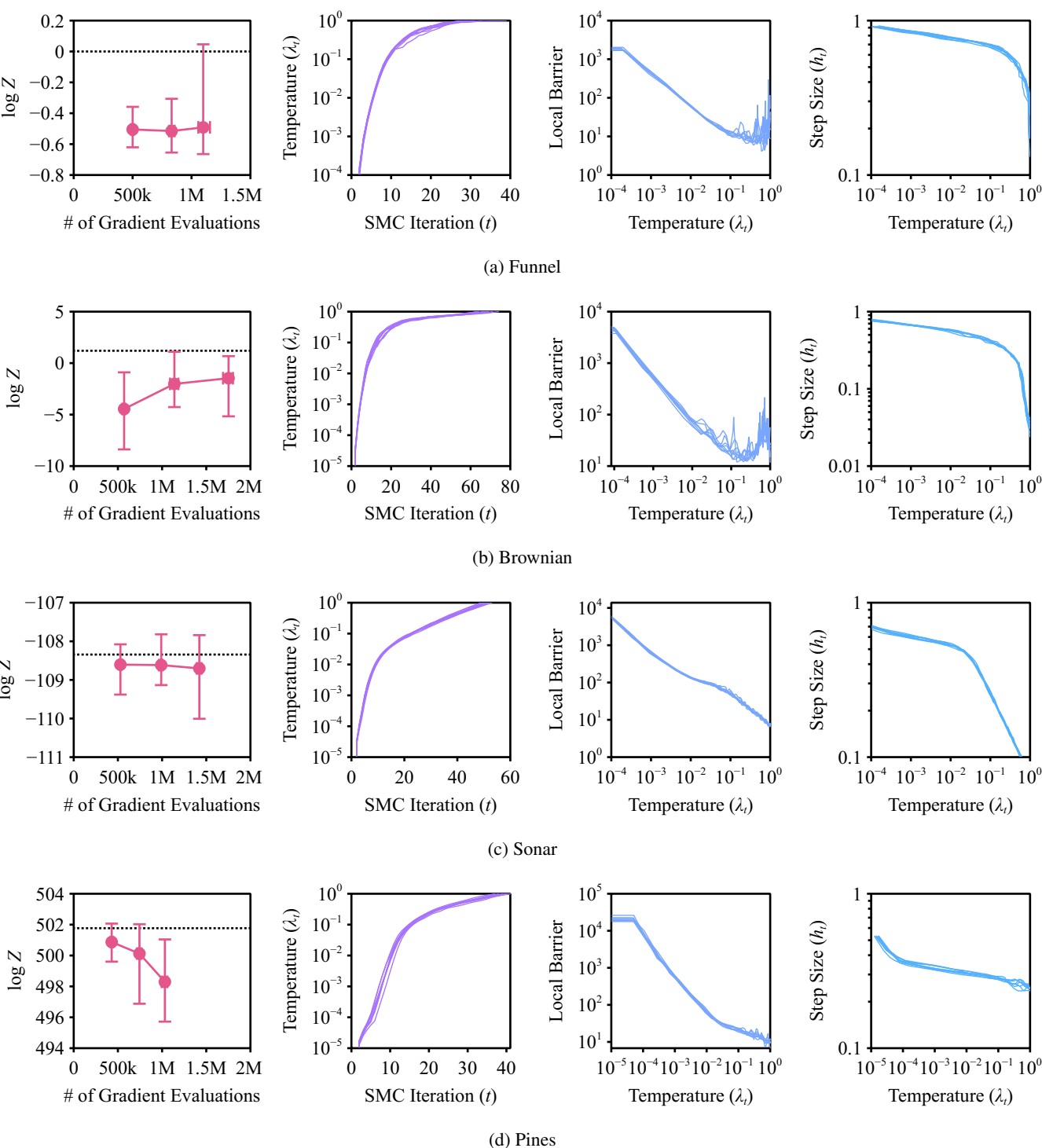

*Figure 32.* **Normalizing constant estimate, temperature schedule, local communication barrier, and step size schedules obtained by running SMC-KLMC.** The dotted line is the ground truth value obtained from a large budget run. For the normalizing constant estimate, the confidence intervals in the vertical and horizontal directions are the $80\%$ quantiles obtained from 32 replications. The temperature schedule, local communication barriers, and the step sizes from a subset of 8 runs are shown.

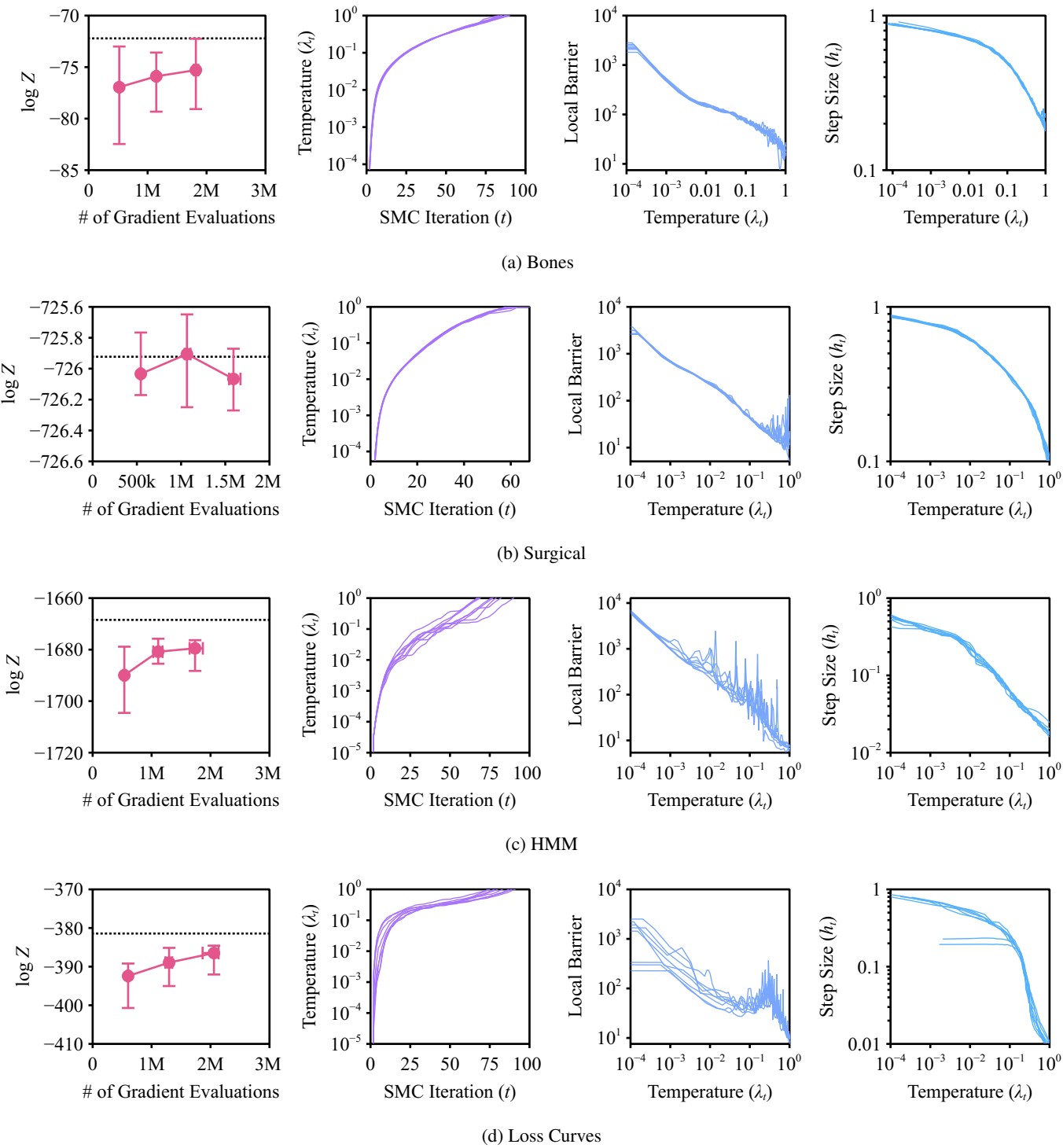

(a) Bones

(b) Surgical

(c) HMM

(d) Loss Curves

*Figure 33.* **Normalizing constant estimate, temperature schedule, local communication barrier, and step size schedules obtained by running SMC-KLMC (continued).** The dotted line is the ground truth value obtained from a large budget run. For the normalizing constant estimate, the confidence intervals in the vertical and horizontal directions are the 80% quantiles obtained from 32 replications. The temperature schedule, local communication barriers, and the step sizes from a subset of 8 runs are shown.

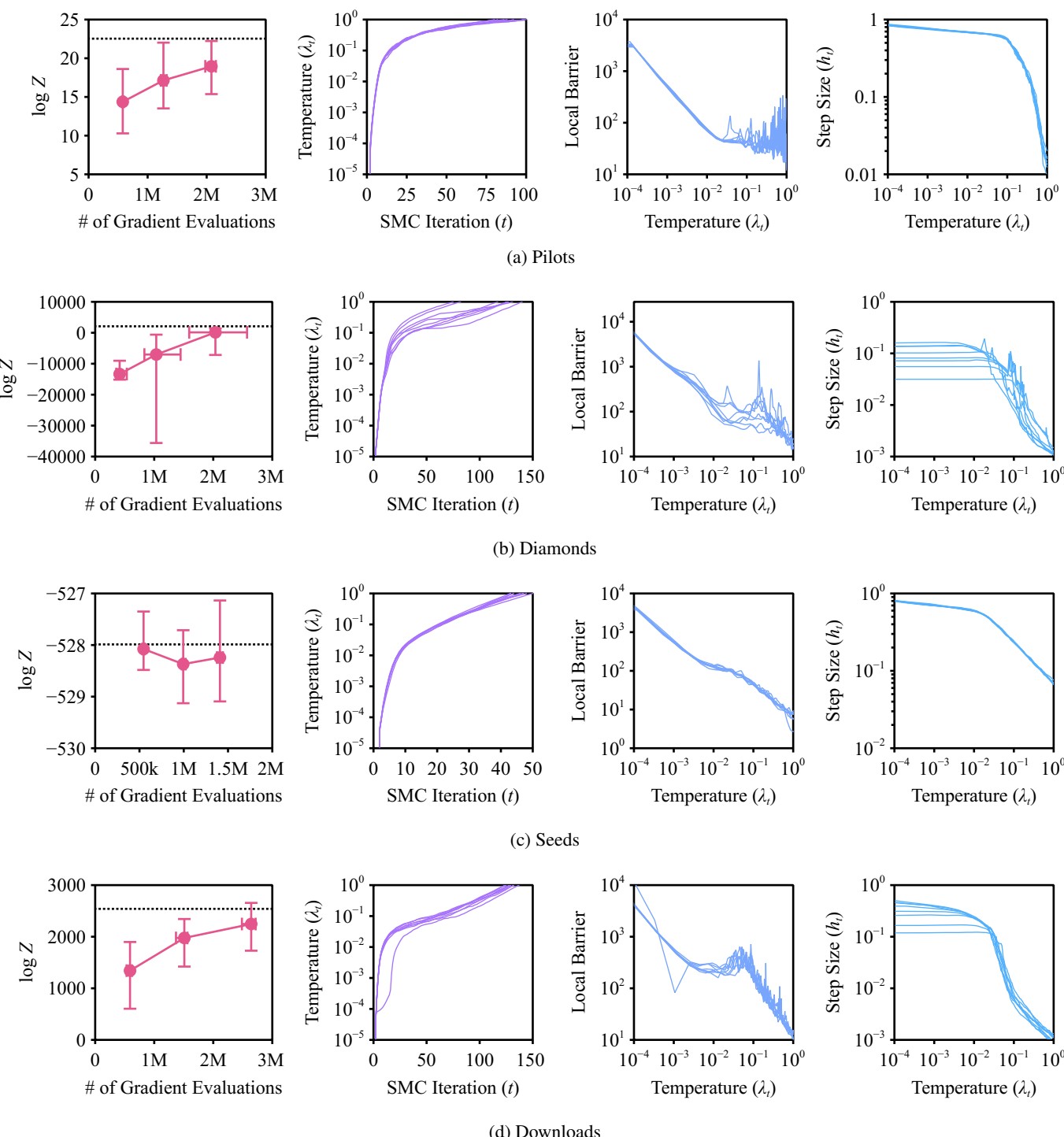

(a) Pilots

(b) Diamonds

(c) Seeds

(d) Downloads

*Figure 34.* **Normalizing constant estimate, temperature schedule, local communication barrier, and step size schedules obtained by running SMC-KLMC (continued).** The dotted line is the ground truth value obtained from a large budget run. For the normalizing constant estimate, the confidence intervals in the vertical and horizontal directions are the 80% quantiles obtained from 32 replications. The temperature schedule, local communication barriers, and the step sizes from a subset of 8 runs are shown.

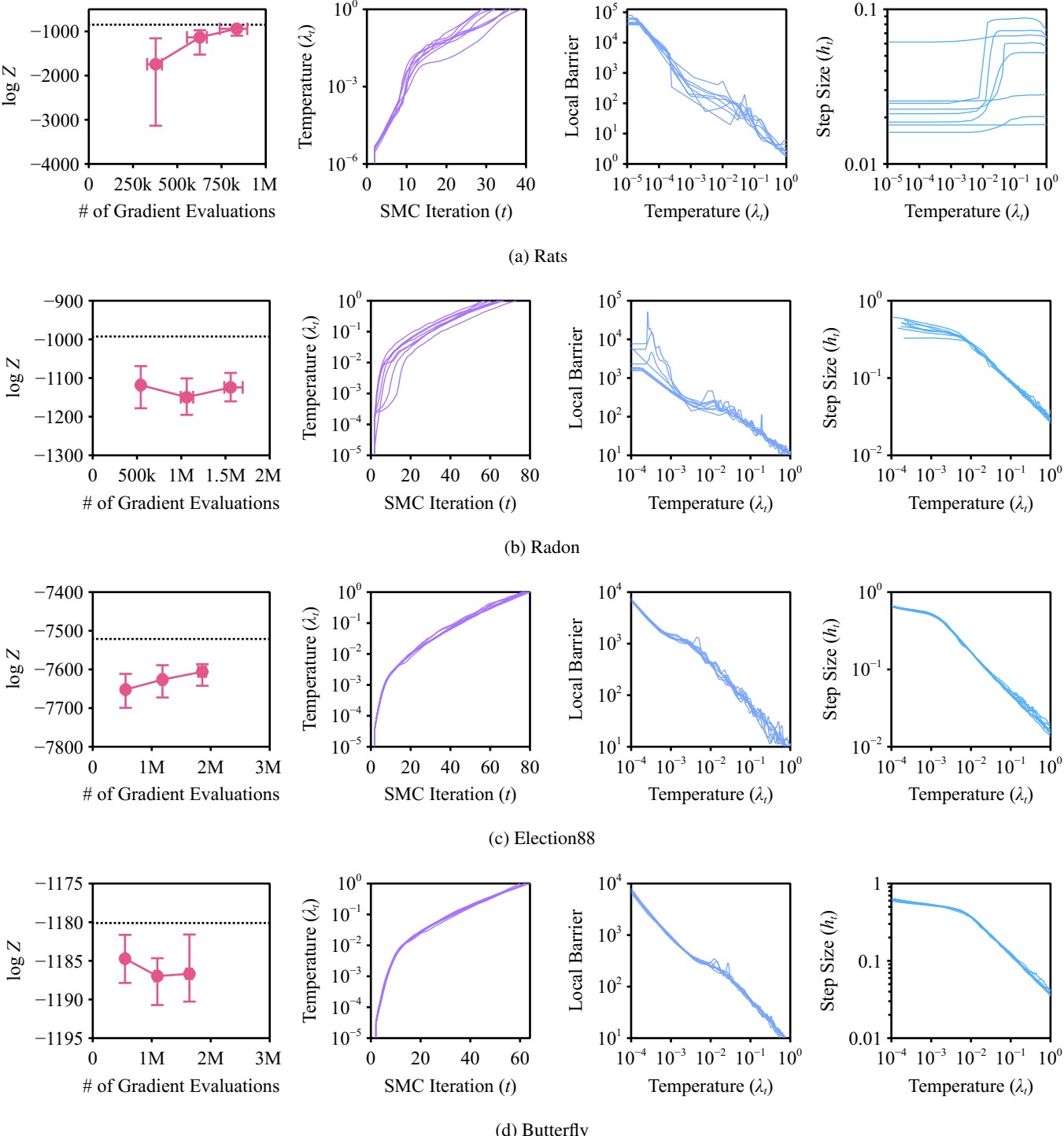

(a) Rats

(b) Radon

(c) Election88

(d) Butterfly

*Figure 35.* **Normalizing constant estimate, temperature schedule, local communication barrier, and step size schedules obtained by running SMC-KLMC (continued).** The dotted line is the ground truth value obtained from a large budget run. For the normalizing constant estimate, the confidence intervals in the vertical and horizontal directions are the 80% quantiles obtained from 32 replications. The temperature schedule, local communication barriers, and the step sizes from a subset of 8 runs are shown.

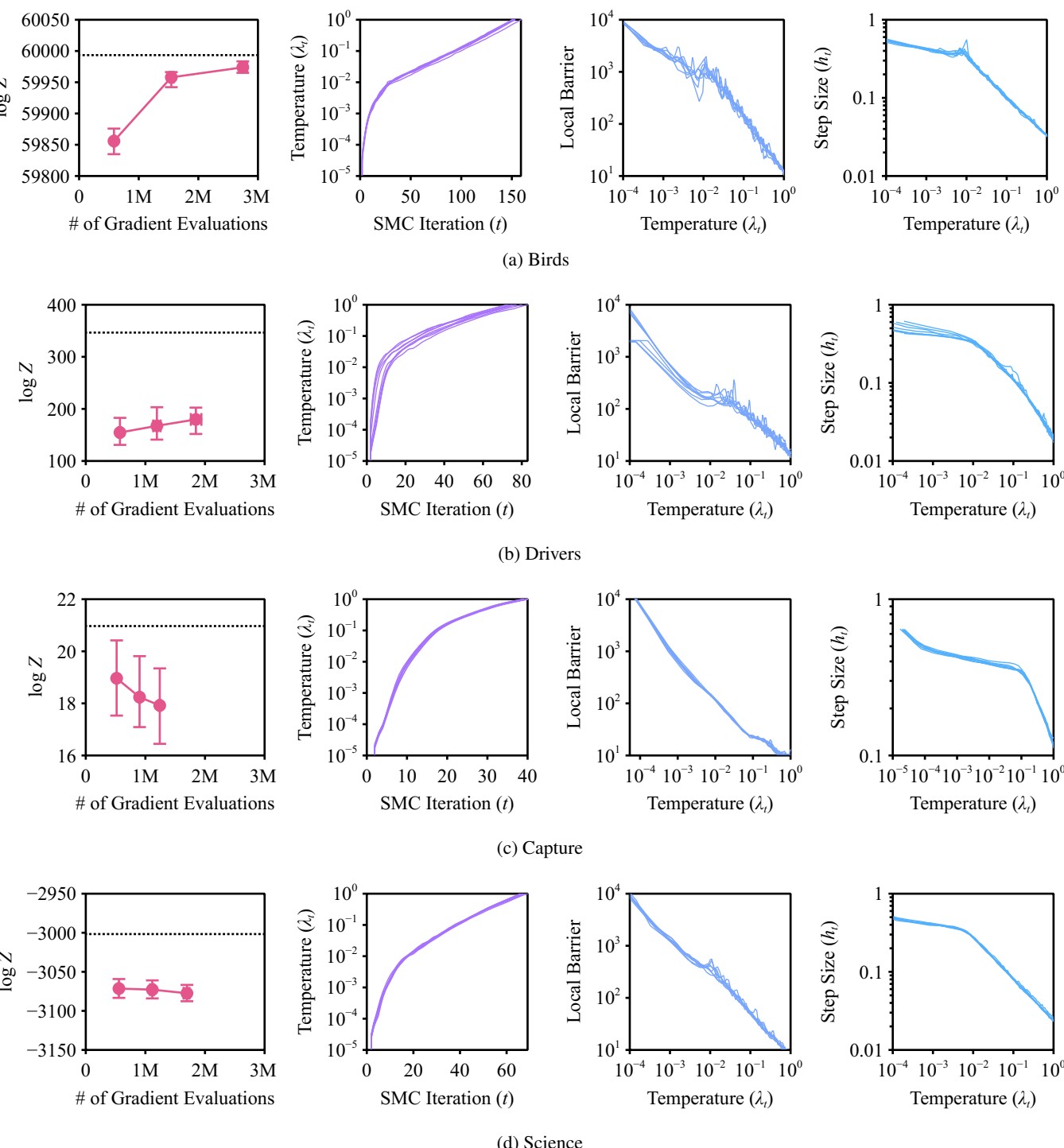

*Figure 36.* **Normalizing constant estimate, temperature schedule, local communication barrier, and step size schedules obtained by running SMC-KLMC (continued).** The dotted line is the ground truth value obtained from a large budget run. For the normalizing constant estimate, the confidence intervals in the vertical and horizontal directions are the 80% quantiles obtained from 32 replications. The temperature schedule, local communication barriers, and the step sizes from a subset of 8 runs are shown.

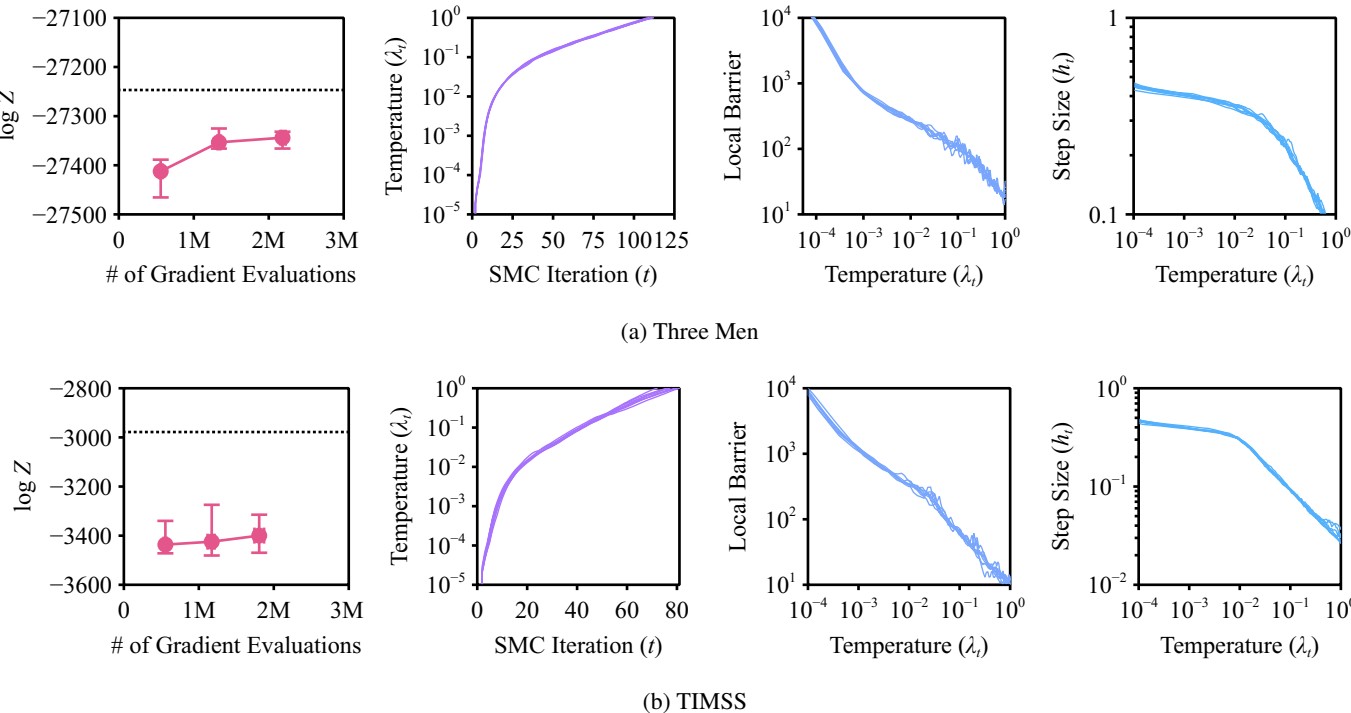

(a) Three Men

(b) TIMSS

*Figure 37.* **Normalizing constant estimate, temperature schedule, local communication barrier, and step size schedules obtained by running SMC-KLMC (continued).** The dotted line is the ground truth value obtained from a large budget run. For the normalizing constant estimate, the confidence intervals in the vertical and horizontal directions are the $80\%$ quantiles obtained from 32 replications. The temperature schedule, local communication barriers, and the step sizes from a subset of 8 runs are shown.

