# OpenReview forum: "Tuning Sequential Monte Carlo Samplers via Greedy Incremental Divergence Minimization"
_ICML.cc/2025/Conference — ICML 2025 poster_

### Official Review · Reviewer_beQ4 · 2025-02-17

**Overall Recommendation:** 4

**Summary:**

The authors propose a method for tuning kernels in SMC by minimising a KL-divergence between proposal paths and target paths. This optimisation is done using a gradient free method. They show that by adapting the step size with their method they get better normalising constant estimates than using fixed step sizes, and that their adaptation scheme has a smaller computation cost compared to setting the step size using gradients.

## update after rebuttal

I maintain my recommendation for acceptance.

**Claims And Evidence:**

From the paper the two key empirical claims are:

1) Our methods achieve lower variance in normalizing
constant estimates compared to the best fixed step sizes
obtained through grid search or SGD-based tuning methods

Figures 2,3,8 and 9 evidence this, with the exception of one or two examples. Some interpretation as to why, eg, the method under-performed on the rats dataset would be welcome.

2) Additionally, the computational cost of our adaptation
scheme is several orders of magnitude lower than that of
SGD-based approaches (Section 5).

Figures 5, 10, 11, 12 and 13 show this. The choices made in the experimental setups for these didn't seem unreasonable so I think these claims were well evidenced.

**Essential References Not Discussed:**

Not that I know of.

**Experimental Designs Or Analyses:**

The experiments seem sound if a bit limited (as I previously mentioned) from what is relayed in the paper. I am curious why 32 replications were used to get the confidence intervals in the adaptive tuning vs fixed experiments but only 5 in the end-to-end optimisation experiments.

**Methods And Evaluation Criteria:**

The datasets used seem quite standard for evaluating SMC methods. One thing I am not sure of is whether just reporting normalising constant is sufficient to show the superiority of this method, it would have been nice to see something like ESS or some assessment of posterior quality like moment estimates.

For comparing against end to end optimisation it wasn't clear to me why a neural network as opposed to any other method would be chosen for learning the step size.

**Other Comments Or Suggestions:**

On line 210 in the second column you refer to h_t before it has been introduced which is potentially confusing. Also I wasn't sure what \theta_t contains h_t meant.

"distribution" is repeated on line 85 in the second column.

**Other Strengths And Weaknesses:**

I found this paper quite clear and this method seems much simpler than previous methods for tuning SMC samplers.

**Questions For Authors:**

Is there any indication that this method under-explored the posterior or suffered more from particle degeneracy than the other compared methods?

**Relation To Broader Scientific Literature:**

This paper aims to tune the hyperparameters of SMC kernels by minimising the divergence between the proposal path and the target path. This idea has similarities to annealed flow transport Monte Carlo and normalising flow.

The issue of adapting the hyperparamters of SMC kernels is a long standing one (P Fearnhead and Benjamin M. Taylor, An Adaptive Sequential Monte Carlo Sampler).

The idea of using the KL divergence to tune parameters of the proposal distribution in SMC also appears in (Shixiang et. al. Neural Adaptive Sequential Monte Carlo).

**Theoretical Claims:**

I didn't check these very closely, but appreciated that some empirical study was done to see how realistic the assumptions were (figure 1.)

---

> ### Author Rebuttal · Authors · 2025-03-29
>
> Thank you for your review.
>
> > The datasets used seem quite standard for evaluating SMC methods. One thing I am not sure of is whether just reporting normalising constant is sufficient to show the superiority of this method, it would have been nice to see something like ESS or some assessment of posterior quality like moment estimates.
>
> Thank you for the suggestion! Theoretically speaking, the accuracy of the normalizing constant and expectations taken over the particles are closely related. That is, the variance of the normalizing constant estimate is the cumulation of the variance of the normalizing constant ratios $Z_t/Z_{t-1}$, which corresponds to setting the integrand of expectation taken over the particles to be the indicator function as $\varphi = \mathbb{1}$. Therefore, if the normalizing constant estimate is accurate, the particle expectations will also be reasonably accurate. Indeed, we ran some experiments on two benchmarks: seeds and radon, where we computed the Wasserstein-2 distance of $N = 1024$ particles from SMC-LMC against ground truth samples from $10^5$ samples from the no-u-turn sampler. The Wasserstein-2 distance is closely related to the accuracy of the first and second moments. Also, it bounds the Wasserstein-1 distance, an integral probability metric, which quantifies the worst-case absolute error for 1-Lipschitz integrands. Therefore, it is a good metric for quantifying the error of particle expectations.
>
> The results are shown in the following link:
>
> https://imgur.com/a/80l62vS
>
> (red line: Wasserstein-2 distance between the samples from vanilla SMC with the corresponding fixed stepsize and NUTS; blue line: Wasserstein-2 distance between the samples from our adaptive SMC sampler and NUTS.)
>
> We can see that our adaptive SMC samplers achieve Wasserstein-2 distances close to the best fixed step size. However, it does not outperform fixed stepsizes as much as for estimating normalizing constants. This suggests that our adaptation scheme could be tailored to expectations taken over the particles for better performance.
>
> > Some interpretation as to why, eg, the method under-performed on the rats dataset would be welcome.
>
> Thank you for pointing this out. With full honesty, it is unclear why this is the case. Upon close inspection, the optimization algorithm seems to correctly find the minimizer of the objective. Therefore, two explanations are possible: Either the sampler is underperforming for the given budget such that the estimate of the incremental KL is inaccurate, or the greediness of the scheme results in a suboptimal solution.
>
> > The experiments seem sound if a bit limited (as I previously mentioned) from what is relayed in the paper. I am curious why 32 replications were used to get the confidence intervals in the adaptive tuning vs fixed experiments but only 5 in the end-to-end optimisation experiments.
>
> We agree with the reviewer that more than 5 evaluations for the end-to-end results would have been better. Therefore, since the submission, we re-ran the experiments using 32 replications for evaluation, which is now the same number as adaptive SMC, and included a more challenging problem (Pines). We observe that on Pines, our method does not outperform end-to-end optimization, which is unsurprising: end-to-end optimization should outperform our greedy scheme on some problems where the gradient noise is negligible. For the grid search experiments, we ran more problems from PosteriorDB, which now totals 21 benchmark problems. Please refer to the response reviewer 3HR9 above!
>
> > The idea of using the KL divergence to tune parameters of the proposal distribution in SMC also appears in (Shixiang et. al. Neural Adaptive Sequential Monte Carlo).
>
> Thank you for pointing this out! We will add this to the list of works performing end-to-end optimization.
>
> > Is there any indication that this method under-explored the posterior or suffered more from particle degeneracy than the other compared methods?
>
> Generally, our method should perform as well as a well-tuned vanilla SMC sampler, which may or may not fully explore the posterior depending on the problem and other configurations. In terms of particle degeneracy, our method should be the least susceptible since we are essentially maximizing the incremental weights. In a sense, our scheme is actively preventing degeneracy. The side effect would be that the biased normalizing constant estimates obtained during adaptive runs will be overestimated.

---

> > ### Comment · Reviewer_beQ4 · 2025-04-02
> >
> > Thank you for addressing my concerns and for the additional experiments. I maintain my recommendation.

---

### Official Review · Reviewer_7uSZ · 2025-03-13

**Overall Recommendation:** 4

**Summary:**

The paper provides a gradient-free, hyperparameter-free tuning algorithm for proposal step sizes and particle refresh/resampling rates in Sequential Monte Carlo pipelines.  Experiments on example graphical models from PosteriorDB demonstrate that the new algorithm not only provides a great advantage in log-evidence estimation relative to a fixed step size, but in fact quickly approaches the true log-evidence after adaptation. In experiments comparing to end-to-end adaptive SMC approaches, the new method performs no worse than end-to-end optimization and, in one experiment, significantly better.

**Claims And Evidence:**

This paper poses a bit of a clarity problem, at least for this reviewer.  The paper takes its starting notation from "Introduction to Sequential Monte Carlo" by Chopin and Papaspiliopoulos, and as a result, is somewhat notationally overloaded.

**Essential References Not Discussed:**

I do not have any missing references to report.

**Experimental Designs Or Analyses:**

PosteriorDB is the gold-standard for benchmarking Bayesian inference and the numerical metrics used here are the valid ones.

**Methods And Evaluation Criteria:**

Yes, in fact PosteriorDB is just the benchmark for Bayesian inference problems that I would recommend.

**Other Comments Or Suggestions:**

There are a few grammar and usage typos, such as "distribution distribution" and "the goal is often to infer... or estimating" (either the infinitive or gerund should be used consistently).

Score has been revised in light of the authors' response to both my review and to others asking for additional experiments.

**Other Strengths And Weaknesses:**

The problem of approximating an idealized annealing path distribution is, to my knowledge, a relatively original one to pose.  I would somewhat like the authors to motivate it more.

**Questions For Authors:**

Solved in author response.

**Relation To Broader Scientific Literature:**

SMC is a back-end workhorse for many scientific computations, and so the paper fits well into the broader literature.  I do not know of a previous paper doing what this paper does.

**Theoretical Claims:**

Proposition 1 more-or-less follows from the definition of a divergence, though a proof summary in the main text would be preferable.

---

> ### Author Rebuttal · Authors · 2025-03-29
>
> Thank you for your review.
>
> > Do the authors want to minimize incremental divergences in order to target the path measure of the annealed importance sampling procedure, or fine-tune an annealed importance sampling procedure in order to later perform SMC on the target density?
>
> We kindly request more details on this question as our tuning procedure does not involve nor is specialized to annealed importance sampling (AIS).
>
> Perhaps the question arose from the fact that the Feynman-Kac model formalism does not explicitly involve resampling, which might have led to the impression that optimizing the target path of the Feynman-Kac model is tuning an AIS procedure, not an SMC procedure. If this is the case, we would like to clarify that while our procedure can absolutely be applied to AIS (after all, SMC and AIS are a few resampling steps away from each other), our focus is purely on SMC samplers. That is, both AIS and SMC are algorithms for simulating the same Feynman-Kac model (it’s only the procedure for simulation that differs.) To be clear, we would like to bring attention to Algorithm 1, which shows the adaptive SMC implementation with our tuning scheme embedded in it. All we do is run Algorithm 1.
>
> If this does not fully address the Reviewer’s question, we would be very happy to further clarify any point of ambiguity.

---

> > ### Comment · Reviewer_7uSZ · 2025-04-04
> >
> > Equation 1 on line 123 shows the geometric annealing path, hence my reference to annealed importance sampling.  In algorithm 1, the potential G is subscripted by a time index, and its definition is a density ratio that helps to weight samples from \pi_{t-1} to target \pi_{t}.
> >
> > My understanding of your response here is that you want to sample from the path measure via SMC; the path of density ratios chosen can be more-or-less arbitrary (not just the geometric annealing path); and that your incremental divergence objective enables incremental adaptation (i.e. the \theta_{t} step inside the outer loop in Algorithm 1) towards that goal.
> >
> > Revising my score in that light.

---

### Official Review · Reviewer_NHUm · 2025-03-14

**Overall Recommendation:** 3

**Summary:**

This paper proposes a novel method for tuning sequential Monte Carlo (SMC) samplers by greedily minimizing the incremental KL divergence between the target and proposal path measures. The authors develop efficient, gradient-free algorithms for tuning key parameters—such as step sizes in unadjusted Langevin Monte Carlo kernels. Experimental results demonstrate that their approach reduces variance in estimates and outperforms both fixed-parameter and gradient-based tuning methods on various benchmark problems.

**Claims And Evidence:**

The submission's claims are largely supported by comprehensive empirical results. The authors verified their method through experiments on multiple benchmarks that their adaptive tuning yields lower variance estimates and comparable or improved performance compared to fixed or gradient-based methods. One potential caveat is that some theoretical guarantees rely on assumptions (e.g., unimodality of the tuning objective) that may limit generalizability, but within the presented contexts the evidence is clear and convincing.

**Essential References Not Discussed:**

No, all the essential related works appear to be appropriately cited and discussed.

**Experimental Designs Or Analyses:**

I briefly reviewed the experiments; however, I have some concerns about whether they accurately reflect real-world scenarios.

**Methods And Evaluation Criteria:**

The proposed methods and evaluation criteria are well-aligned with the problem at hand. The paper’s approach—minimizing incremental KL divergence to tune SMC samplers—is both innovative and practical, addressing the challenges of tuning unadjusted kernels without incurring high computational costs. The evaluation criteria, such as the variance and accuracy of normalizing constant estimates, are standard in the SMC literature. Additionally, the benchmark datasets are used.

**Other Comments Or Suggestions:**

The document is well written and the technical terminology is used correctly.

**Other Strengths And Weaknesses:**

Strengths:

- Novel and efficient approach that combines divergence minimization with adaptive SMC tuning.
- Provides effective empirical results.

**Questions For Authors:**

- How sensitive are the experimental results to changes in key parameters, and does this sensitivity mirror the challenges encountered in practical scenarios?

- Can the proposed method scale efficiently with the increased dimensionality and sample sizes often found in real-world problems?

**Relation To Broader Scientific Literature:**

The paper’s contributions build directly on a rich body of work in sequential Monte Carlo (SMC) methods and adaptive tuning techniques.

**Theoretical Claims:**

I only checked the main body of the theoretical claims and did not find any issues.

---

> ### Author Rebuttal · Authors · 2025-03-29
>
> Thank you for your review.
>
> > One potential caveat is that some theoretical guarantees rely on assumptions (e.g., unimodality of the tuning objective) that may limit generalizability,
>
> We agree that those theoretical assumptions are restrictive. As such, we are happy to mention that we were able to relax those assumptions during the review period. Now, even without unimodality, we are able to guarantee that our algorithm finds a point $\epsilon$-close to a local minimum with similar computational cost. Additionally assuming unimodality strengthens the guarantee so that the returned point is $\epsilon$-close to the global optimum.
>
> The only major change we had to introduce is that we now use a different variant of the golden section search algorithm. In particular, we use the version described in “Numerical Recipes” (Section 10.1 in [1]). We found that this variant is able to guarantee finding a local minimum as long as the initialization satisfies a certain condition implying that the initial search interval contains a local minimum.
>
> > How sensitive are the experimental results to changes in key parameters, and does this sensitivity mirror the challenges encountered in practical scenarios?
>
> Thank you for the great question. Overall, we didn’t spend much effort tuning the parameters, and all of our experiments were run with a single fixed set of parameters for each type of kernel. In more detail though, the only parameters that primarily affect the results are the regularization strength $\tau$ and the optimization accuracy budget $\epsilon$. (The rest of the parameters only change the amount of computation spent obtaining similar solutions.) $\epsilon$, for instance, does not affect the result much as long as it is small enough. On the other hand, the effect of the regularization strength $\tau$ seems to depend on the kernel but not so much on the problem. For instance, the performance of LMC isn’t very sensitive to $\tau$, and we thus use a small amount. On the other hand, KLMC seems to require stronger regularization to obtain good performance. While we suspect this has something with the persistence of the momentum, it is not entirely clear why. Overall, we do not expect the parameters to require much tuning, except when swapping the MCMC kernel.
>
> > Can the proposed method scale efficiently with the increased dimensionality and sample sizes often found in real-world problems?
>
> We extended our experiments with higher dimensional problems since the initial submission. (Please refer to the response to 3HR9 for the new results.) In principle, dimensionality shouldn’t be a problem for our method. As shown in the experiment, our method should perform better or as well as the best-tuned vanilla SMC sampler. Therefore, as long as vanilla SMC can scale, our method should be able to scale as well. In terms of scaling with respect to the sample size $N$, as long as the subsampling size $B$ is smaller than $N$, the added overhead of our method should be negligible. More concretely, if at most $C$ objective evaluations are spent during each adaptation step, the total cost of running SMC with our adaptation is $O(B C +  N)$. Thus, if $BC = o(N)$, our adaptive SMC scheme is just as scalable as vanilla SMC.
>
> 1. Press, William H. Numerical recipes 3rd edition: The art of scientific computing. Cambridge University Press, 2007.

---

### Official Review · Reviewer_3HR9 · 2025-03-15

**Overall Recommendation:** 3

**Summary:**

Main problem and approach: The performance of sequential Monte Carlo (SMC) samplers heavily depends on the tuning of the Markov kernels used in the path proposal. The paper proposes a framework for tuning the Markov kernels in SMC samplers by minimizing the incremental Kullback-Leibler (KL) divergence between the proposal and target paths.
Main Result: the paper show that the approach and implementation are able to obtain a full schedule of tuned parameters at the cost of a few vanilla SMC runs, which is a fraction of gradient-based approaches.

**Claims And Evidence:**

Yes

**Essential References Not Discussed:**

I am not familiar with the domain and did not check thoroughly.

**Experimental Designs Or Analyses:**

The experiment set up is too simple on a collection of toy benchmark datsets. It is not clear how it is applicable to the application domains of SMC, such as steering large language models and conditional generation from diffusion models.

**Methods And Evaluation Criteria:**

The method is sound. The evaluation and benchmark datasets are too simple.

**Other Comments Or Suggestions:**

Provide a practical example to demonstrate the value of this work.

**Other Strengths And Weaknesses:**

I am not familiar with the domain. The paper is well written and the claims seem solid. The experiment setup is too simple, it is hard to assess the practical value of this work.

**Questions For Authors:**

How the proposed tuned SMC samplers can bring benefit to its applications such as steering large language models and conditional generation from diffusion models.

**Relation To Broader Scientific Literature:**

It address an open questions in the broader scientific literature.  Tuning SMC is often a significant challenge.  methods and criteria for tuning the path proposal kernels are relatively scarce. This paper focuses on the setting where a few scalar parameters (e.g., step size) subject to tuning. In this setting, the full generality (and cost) of SGD is not required; it is possible to design a simpler and more efficient method for tuning each
transition kernel sequentially in a single SMC/AIS run.

**Theoretical Claims:**

No fully check the correctness of  theoretical claims

---

> ### Author Rebuttal · Authors · 2025-03-29
>
> Thank you for your review.
>
> > The experiment set up is too simple on a collection of toy benchmark datasets. It is not clear how it is applicable to the application domains of SMC, such as steering large language models and conditional generation from diffusion models.
>
> We agree that more realistic experiments are always a good thing. As such, we added new benchmarks from both PosteriorDB and Inference Gym, including a high-dimensional benchmark problem (Pines) from Inference Gym, which is 1600 dimensional and multiple problems from PosteriorDB, including TIMSS and Three Men, both of which have more than 500 dimensions. Here is the full list of problems with their corresponding dimensionality:
>
> | Name | dims. | Source |
> | ----- | --- | ---- |
> | Funnel | 10 | Inference Gym |
> | Brownian | 32 | Inference Gym |
> | Sonar | 61 | Inference Gym |
> | Pines | 1600 | Inference Gym |
> | Bones | 13 | PosteriorDB |
> | Surgical | 14 | PosteriorDB |
> | HMM | 14 | PosteriorDB |
> | Loss Curves | 15 | PosteriorDB |
> | Pilots | 18 | PosteriorDB |
> | Diamonds | 26 | PosteriorDB |
> | Seeds | 26 | PosteriorDB |
> | Rats | 65 | PosteriorDB |
> | Radon | 90 | PosteriorDB |
> | Election88 | 90 | PosteriorDB |
> | Butterfly | 106 | PosteriorDB |
> | Birds | 237 | PosteriorDB |
> | Drivers | 389 | PosteriorDB |
> | Capture | 388 | PosteriorDB |
> | Science | 408 | PosteriorDB |
> | Three Men | 505 | PosteriorDB |
> | TIMSS | 530 | PosteriorDB |
>
>
>
> The new experimental results can be found in the anonymized links below.
>
> Comparison against end-to-end optimization:
>
> https://imgur.com/pdk3Bza
>
> (red dot: normalizing constant estimate obtained by our adaptive SMC sampler; blue line: normalizing constant estimate obtained by end-to-end optimization; dotted line: ground truth)
>
> Here, we can see the comparison against end-to-end optimization results with the new Pines benchmark problem. Furthermore, unlike the experiments included in the original submission, we use the official code provided by Geffner and Domke [1], which we find to be better tuned. In particular, on Pines, we observe that end-to-end optimization performs slightly better than our adaptive SMC procedure, while on other problems, we obtain more accurate results.
>
> Comparison against grid search for SMC with Langevin Monte Carlo kernels:
>
> https://imgur.com/9JjJEiK
>
> Comparison against grid search for SMC with kinetic Langevin Monte Carlo kernels:
>
> https://imgur.com/mUeWbuz
>
> (red line: the normalizing constant obtained by our adaptive SMC sampler; purple line: the normalizing constant obtained by vanilla SMC with a fixed stepsize; dotted line: ground truth estimate.)
>
> Overall, for the comparisons against grid search, we observe a similar trend where our tuning procedure finds better or comparable results to the best-fixed step size. The only exception appears to be Rat, as observed in the original submission.
>
> We would also like to point out that PosteriorDB is a state-of-the-art benchmark (with the corresponding paper published this year at AISTATS’25 [2]) that contains problems that have actually been used for practical applications or closely resemble such problems. As such, we believe our experiments adequately represent problems to which SMC is expected to be applied in practice. While evaluating our method on applications such as LLM steering and conditional generation from diffusion models would definitely be interesting, they will probably need their own investigation with more specialized solutions.
>
> 1. Geffner, Tomas, and Justin Domke. "Langevin diffusion variational inference." International Conference on Artificial Intelligence and Statistics. PMLR, 2023.
> 2. Magnusson, Måns, et al. "posteriordb: Testing, benchmarking and developing Bayesian inference algorithms." AISTATS’25, to be presented.

---

### Decision · Program_Chairs · 2025-05-01

**Decision:**

Accept (poster)

**Comment:**

The paper introduces a novel approach for tuning SMC proposal kernels; all reviewers agree that this is a clear method, praise the fact it does not require expensive or difficult end-to-end gradient-based training, and the paper itself is well-written. Additional experiments were added during the rebuttal period to strengthen the experimental validation of the approach. All reviewers agreed the paper should be accepted.